# URBANVERSE: SCALING URBAN SIMULATION BY WATCHING CITY-TOUR VIDEOS

**Mingxuan Liu**[1,2,*,†]  **Honglin He**[1,*]  **Elisa Ricci**[2,3]  **Wayne Wu**[1]  **Bolei Zhou**[1]
[1]University of California, Los Angeles  [2]University of Trento  [3]Fondazione Bruno Kessler

https://urbanverseproject.github.io/

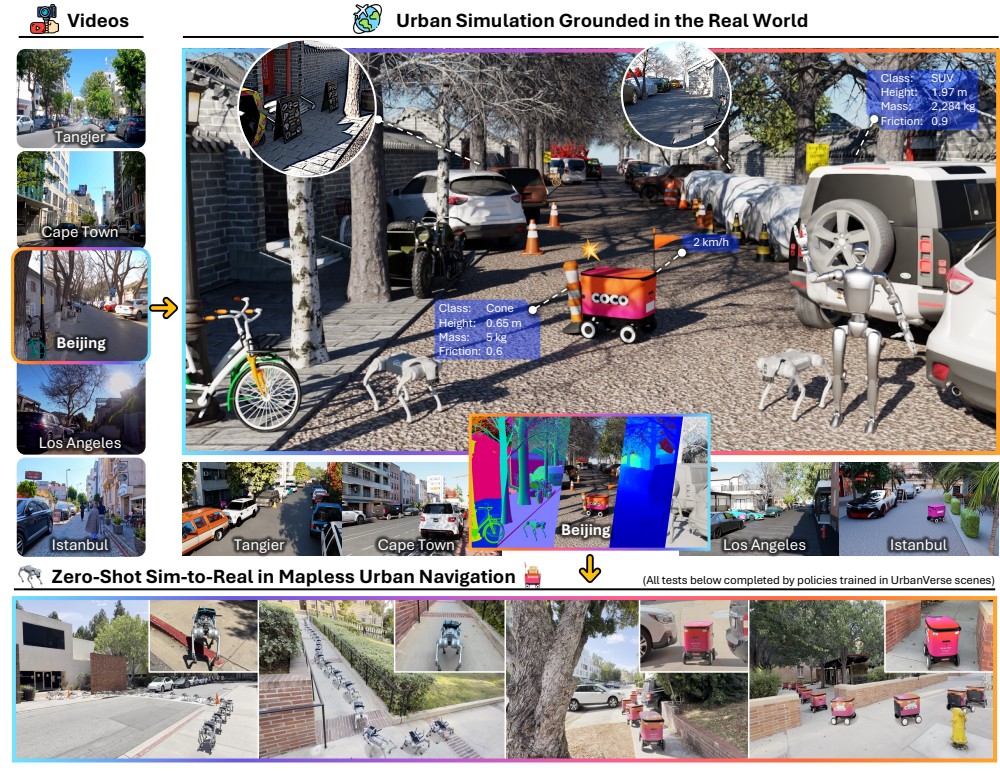

Figure 1: **UrbanVerse** system converts real-world urban scenes from city-tour videos into physics-aware, interactive simulation environments, enabling scalable robot learning in urban spaces with real-world generalization.

## ABSTRACT

Urban embodied AI agents, ranging from delivery robots to quadrupeds, are increasingly populating our cities, navigating chaotic streets to provide last-mile connectivity. Training such agents requires diverse, high-fidelity urban environments to scale, yet existing human-crafted or procedurally generated simulation scenes either lack scalability or fail to capture real-world complexity. We introduce **UrbanVerse**, a data-driven real-to-sim system that converts crowd-sourced city-tour videos into physics-aware, interactive simulation scenes. UrbanVerse consists of: *(i) UrbanVerse-100K*, a repository of 100k+ annotated urban 3D assets with semantic and physical attributes, and *(ii) UrbanVerse-Gen*, an automatic pipeline that extracts scene layouts from video and instantiates metric-scale 3D simulations using retrieved assets. Running in IsaacSim, UrbanVerse offers 160 high-quality constructed scenes from 24 countries, along with a curated benchmark of 10 artist-designed test scenes. Experiments show that UrbanVerse scenes preserve real-world semantics and layouts, achieving human-evaluated realism comparable to manually crafted scenes. In urban navigation, policies trained in UrbanVerse exhibit scaling power laws and strong generalization, improving success by +6.3% in simulation and +30.1% in zero-shot sim-to-real transfer comparing to prior methods, accomplishing a 300 m real-world mission with only two interventions.

*Equal contribution. †Work conducted while Mingxuan Liu was a Visiting Researcher at UCLA.

# 1 INTRODUCTION

Today's urban spaces have emerged as key arenas for the rise of micromobility systems (Oeschger et al., 2020). Small, lightweight Embodied AI (E-AI) agents (Zhang et al., 2024), such as wheeled delivery robots, quadrupeds, and humanoids, are increasingly navigating city streets to provide last-mile transportation and urban services. These mobile robots are reshaping the dynamics of city streets by improving logistical efficiency and reducing carbon emissions. Yet, the environments they operate in are often highly cluttered, with street infrastructure, sidewalks frequently blocked by parked cars, and narrow passageways, all of which pose significant challenges for E-AI agents to generalize across diverse real-world settings.

In efforts to improve generalizability, data scaling has been validated in both vision (Radford et al., 2021) and language (Kaplan et al., 2020), where large and diverse web corpora consistently lead to better generalization. In contrast, current urban navigation datasets remain limited in both *scale* and *quality*. Collecting high-quality data through human demonstrations in public spaces (*e.g.*, sidewalk traversing) is unsafe, labor-intensive, and often impractical, and thus cannot scale (Shah et al., 2023). Passive real-world data, such as city-tour videos shared daily on social media like YouTube, captures diverse environments but lacks associated action labels. Moreover, both types of data are *non-interactive* and lack *causal action–effect* dynamics, which are crucial for robust real-world decision-making. Thus, urban simulators (Wu et al., 2025a) have emerged as a compelling alternative, offering interactive and virtually unlimited scenes for E-AI training. Yet, current simulators either rely on hand-crafted scenes (Dosovitskiy et al., 2017) or procedurally generated layouts defined by hard-coded rules (Wu et al., 2025a;b). The former lacks scalability, while the latter produces rigid, template-driven scenes that deviate from real-world distributions, such as randomly parked scooters or cars. However, simply increasing its quantity will lead to sustained generalization if the data does not faithfully reflect real-world distributions. This issue prompts the central question of this work: *Can we build realistic, interactive urban scenes from real-world videos for scalable robot navigation?*

To address this question, we propose **UrbanVerse**, a data-driven real-to-sim system that bridges the gap between synthetic simulators and the complex, real-world streets. As shown in Fig. 1 (top), UrbanVerse reconstructs interactive urban scenes with real-world distributional fidelity from worldwide city-tour videos (walking or driving), enabling the training of "street-smart" urban agents. At its core, UrbanVerse reconstructs *digital cousin* (Dai et al., 2024) scenes, by mapping a 2D scene to a 3D simulation-ready virtual world, with layouts, semantics, and physics aligned to real-world statistics. *This combines the diversity of real data with the interactivity of simulation*, enabling unlimited scene generation while preserving street-level distributions. UrbanVerse builds on two complementary pillars. The first is **UrbanVerse-100K** (Fig. 2), a curated repository of 102,444 high-quality metric-scale urban object assets, 306 skyboxes, and 288 ground materials for roads and sidewalks. Each asset is organized within a three-level urban ontology and annotated with 33 semantic, affordance, and physical attributes (*e.g.*, mass, friction), forming the foundation for open-world, physics-ready scene construction. The second is **UrbanVerse-Gen** (Fig. 4), an automatic pipeline that accurately distills semantics, layouts, ground appearance, and sky from videos into an urban scene graph representation, retrieves matched assets from UrbanVerse-100K, and instantiates multiple digital cousin simulation scenes in IsaacSim (NVIDIA, 2025). Using YouTube city-tour footage spanning 24 countries across six continents, UrbanVerse produces a library of 160 simulation scenes with real-world street distributional fidelity ready for E-AI training. Together, we also introduce a 3D artist-designed set of 10 realistic urban scenes as test-only environments for closed-loop evaluation.

UrbanVerse is evaluated through both scene generation quality assessment and its applicability to urban E-AI policy learning. Validated on 45 video sequences from KITTI-360 (Liao et al., 2022) , our approach achieves high-fidelity scene reconstruction, recovering object semantics with 93.0% accuracy and localizing objects within 1.4 m error over 198.7 m scene horizon. Building on this real-world distributional fidelity, we train mapless urban navigation policies using a 160-scene simulation library generated by UrbanVerse. We show that data scaling *power-law* emerges when training on our high-fidelity scenes, enabling simple policies to generalize to diverse, unseen urban spaces. Deployed across 16 real-world streets on two embodiments (a wheeled robot and a quadruped) in a zero-shot manner, policies trained on UrbanVerse scenes consistently outperform state-of-the-art navigation foundation models, reaching up to 89.7% success in sim-to-real transfer. Finally, our policy completes a 337 m long-horizon mission in public spaces with only two human interventions. All assets, scenes, and code of UrbanVerse will be *open-sourced* to accelerate embodied AI research.

| Simulator | Scene Creation | Layout Realism | Scene Diversity | Asset Physics | # Object Classes | # Object Assets | # Sky Maps | # Ground Materials | # Robot Types | Training Paradigms | Embodied Tasks |
|---|---|---|---|---|---|---|---|---|---|---|---|
| CARLA | Hand Crafted | Realistic | 15 Scenes | ✓ | 106 | 935 | 5 | 30 | 1 | RL, IL VLA | Navigation, VQA |
| MetaUrban | Procedural Generation | Unrealistic | 7 Templates | ✗ | 39 | 10,000 | 1 | 5 | 1 | RL, IL | Navigation |
| UrbanSim | Procedural Generation | Unrealistic | 6 Templates | ✗ | 39 | 15,000 | 1 | 8 | 10 | RL, IL | Navigation |
| **UrbanVerse** | **Auto Real2sim Data-driven** | **Realistic** | **+∞** | **✓** | **659** | **102,444** | **306** | **288** | **20** | **RL, IL VLA** | **Navigation, VQA Mobile Manipulation** |

Table 1: Comparison of UrbanVerse with existing urban embodied-AI simulators.

## 2 RELATED WORK

**Urban Navigation.** Deep reinforcement learning (Mirowski et al., 2016) has demonstrated strong potential for goal-based mapless urban navigation by removing the dependency on pre-built maps (Chaplot et al., 2020). Recent advances have introduced vision-based navigation foundation models (Shah et al., 2023; Sridhar et al., 2024; Hirose et al., 2025), which leverage cross-sensor capabilities and large-scale offline vision data to improve generalization across robot platforms and camera setups. A major limitation of these approaches, however, is the lack of environmental interaction in the training data. As we demonstrate in Sec. 4.3, this results in poor obstacle avoidance (Liu et al., 2025). S2E (He et al., 2025) tackles this issue by training in interactive simulation scenes, rather than relying on passive data for path-following, and achieves real-world obstacle avoidance. In this work, we extend both coverage and realism by training vision-based position-goal navigation policies in our real-world grounded UrbanVerse scenes, leading to improved robustness and generalization.

**Urban Embodied AI Simulators.** Mainstream simulators focus either on *indoor* environments (Kolve et al., 2017; Li et al., 2023a) or on *driving* domains centered on roads and highways (Kothari et al., 2021; Li et al., 2022), with only CARLA (Dosovitskiy et al., 2017) extending to city neighborhoods but relying on non-scalable, hand-crafted scenes (15 in total). The rise of *micromobility* (Abduljabbar et al., 2021) agents, such as e-scooter or delivery robots, has motivated the development of *urban* simulators like MetaUrban (Wu et al., 2025a) and UrbanSim (Wu et al., 2025b), which share our goal of modeling richer city spaces. However, these simulators face three key limitations: *i) layout realism:* procedurally generated scenes deviate from real-world distributions; *ii) asset diversity:* limited object categories; *iii) physics annotation:* objects lack physical properties and remain static props. We provide a functional comparison between UrbanVerse and existing urban simulators in Tab. 1. Building upon UrbanSim simulation platform, we address these gaps with our real-world grounded scene creation pipeline and a semantically-rich, physics-annotated asset library.

**Simulation Scene Creation.** Automating scene creation for E-AI learning has traditionally relied on either enhancing procedural generation rules (Deitke et al., 2022; Li et al., 2023b; Yang et al., 2024; Wu et al., 2025a;b) or using high-precision 3D scans to replicate real-world environments (Deitke et al., 2023a; Huang et al., 2025; Yu et al., 2025). More recently, 3D Gaussian Splatting (3DGS) methods like OmniRe (Chen et al., 2024) and Vid2Sim (Xie et al., 2025) reconstruct 3DGS digital twin environments from RGB videos for coarse simulation. However, current 3DGS-based digital twins are designed for one-to-one reconstruction and produce a single fused radiance field *without* complete geometry, object instances, semantics, or physical attributes, making them unsuitable for editing, interaction, or physics-based simulation. Our work shares conceptual similarities with digital cousin approaches (Dai et al., 2024; Maddukuri et al., 2025; Melnik et al., 2025), which generate multiple virtual *indoor* scenes from a *calibrated RGB image* while preserving key semantics, geometry, and layout to improve manipulation policies. In contrast, UrbanVerse constructs large-scale, street-level urban digital cousin scenes from *uncalibrated RGB videos*.

## 3 METHODOLOGY

To convert worldwide city-tour videos into physics-aware simulations, two indispensable elements are required: *i)* a large-scale 3D asset database with physical annotations that match the magnitude of real-world semantic richness and appearance diversity, and *ii)* an automated open-vocabulary pipeline that extracts semantic and spatial layouts from any uncalibrated videos to generate simulation scenes. To this end, in Sec. 3.1, we first describe the data collection and semi-automatic annotation pipeline in UrbanVerse for building the **UrbanVerse-100K** database. Next, in Sec. 3.2, we present **UrbanVerse-Gen**, our automated pipeline that extracts detailed scene representations from videos

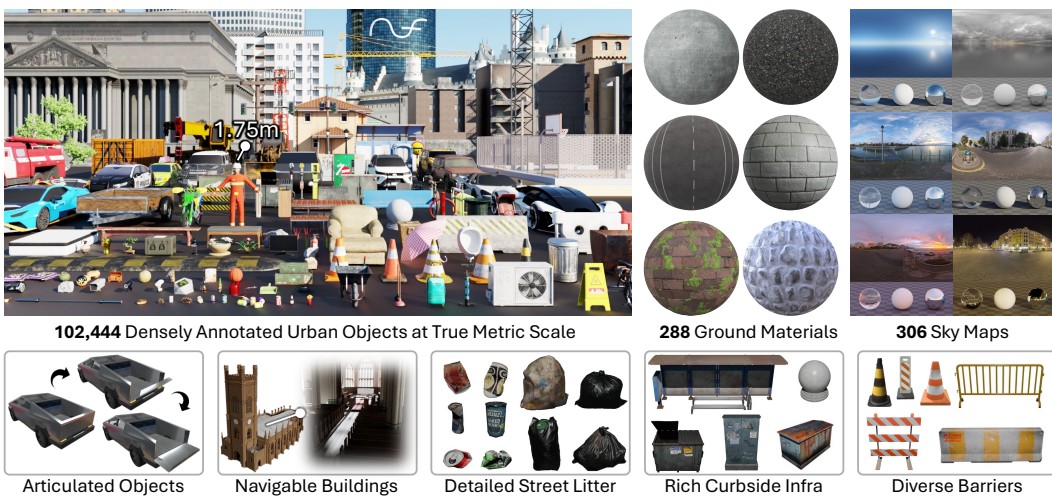

Figure 2: Example instances from our large-scale urban asset database **UrbanVerse-100K**. Assets range from a 0.03 m crushed can to a 200 m skyscraper, all annotated to metric scale; *note the realistic relative scales between objects*, with road and sidewalk materials and sky maps ensuring realistic ground appearance and illumination.

to ground simulated scene generation. Finally, in Sec. 3.3, we show how UrbanVerse leverages crowd-sourced videos to construct a large-scale scene library for policy learning and testing.

## 3.1 URBANVERSE-100K ASSET DATABASE

**UrbanVerse-100K** is a large-scale, high-quality 3D asset database designed for urban simulation and beyond. To comprehensively model real-world scenes, we require not only a vast collection of on-ground 3D object assets but also diverse materials to render ground surfaces (*e.g.*, cobblestone sidewalks or snowy roads) and realistic lighting conditions across different times of day. Each of these significantly influences robot perception and interactivity. For this, as shown in Fig. 2, UrbanVerse-100K comprises three collections: *(i) Object:* 102,444 GLB objects spanning 659 categories, each annotated with 33 semantic, physical, and affordance attributes in true metric scale; *(ii) Ground:* 288 photorealistic PBR materials (98 road, 190 sidewalk) for ground plane texturing; and *(iii) Sky:* 306 HDRI sky maps for realistic global illumination and immersive 360° backgrounds. We next outline the collection and annotation strategies for UrbanVerse-100K, with additional details in App. C.

**Object Collection.** The recent development of large-scale 3D object repositories (Deitke et al., 2023b), such as Objaverse (Deitke et al., 2023c), provides valuable resources for constructing simulation scenes. However, due to their web-crawled nature, these repositories suffer from several critical issues: *(i)* most assets are *unrelated* to urban environments; *(ii)* many assets are *corrupted* (*e.g.*, missing textures, incomplete 3DGS reconstructions, or paper-thin geometry); (iii) assets often have *non-metric* scales, with examples

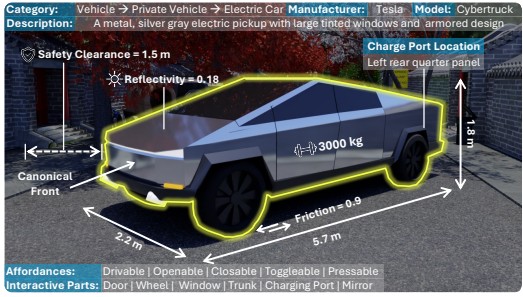

Figure 3: Example of annotated object attributes.

such as a cucumber being as large as a car; and (iv) assets lack semantic and physical attribute annotations. Fig. 25 and Fig. 26 in the appendix illustrate these issues. To this end, we curate a high-quality urban subset from the 800K noisy 3D assets in the Objaverse dataset (Deitke et al., 2023c) through a three-stage semi-automatic pipeline. First, we build a Three.js–based asset viewer interface (Fig. 22) and employ human annotators to efficiently filter out corrupted or low-quality assets. After filtering, we retain 158K high-quality assets. Next, we build a *three-level* urban ontology derived from the OpenStreetMap tag structure (Bennett, 2010) and expanded with categories from driving and scene understanding datasets (Zhou et al., 2017; Cordts et al., 2016; Caesar et al., 2020; Gupta et al., 2019; Kuznetsova et al., 2020), resulting in 659 leaf-level categories. Next, each asset is classified into a leaf category using CLIP (Radford et al., 2021) on its thumbnail, followed by manual verification to remove non-urban items (*e.g.*, game weapons, spaceships) and correct misclassifications, yielding our final curated set of objects. Lastly, we leverage the world knowledge of GPT-4.1 (OpenAI, 2025) to

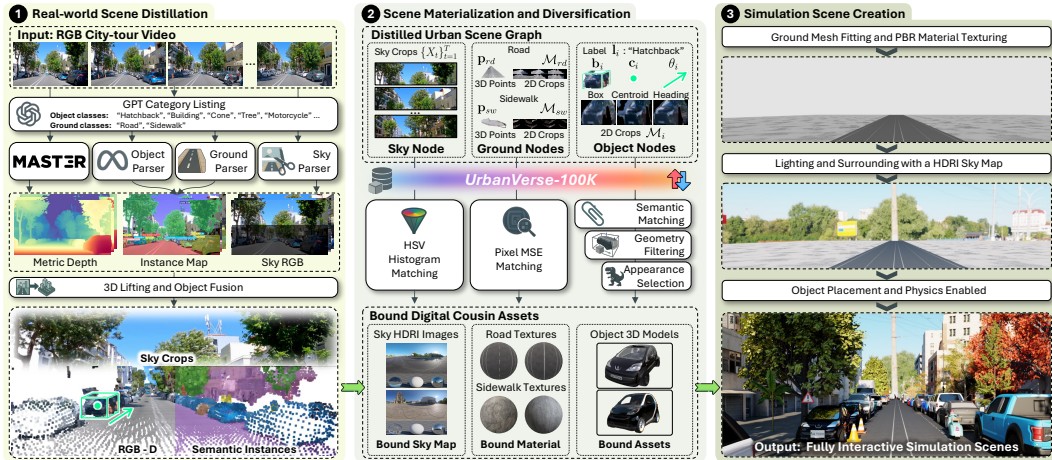

Figure 4: Overview of the UrbanVerse-Gen pipeline. Figure better seen at magnification.

annotate each asset with semantic, affordance, and physical attributes (*e.g.*, size, mass). We prompt it with the object thumbnail and four rotated snapshots. Fig. 3 shows an example of few annotated attributes. Using the annotated size and front-view, as displayed in Fig. 2 (left), we standardize all assets in our database to metric scale and a consistent orientation.

**Ground and Sky Collections.** To provide the simulated ground plane with realistic and diverse appearances, we collect 4K-quality PBR road and sidewalk materials from free-licensed repositories (FreePBR, 2025; AmbientCG, 2025), each with a thumbnail and spanning a wide range of conditions, as shown in Fig. 2 (middle). For lighting, we gather artist-designed HDRI sky maps (Fig. 2 (right)) for urban settings from PolyHaven (2025), each accompanied by a rendered thumbnail and description. These maps enable image-based lighting, providing realistic global illumination with precise control over color, intensity, and direction, while also supplying immersive background.

## 3.2 URBANVERSE-GEN SCENE CONSTRUCTION PIPELINE

With our richly annotated asset database, we design **UrbanVerse-Gen**, an automatic open-vocabulary pipeline capable of extracting real-world 3D scene layouts from uncalibrated RGB city-tour videos and generating fully interactive simulations. To structure a scene, we first introduce a unified 3D urban scene graph $\mathcal{V} = \langle \mathcal{O}, \mathcal{G}, \mathcal{S} \rangle$ that encodes: *(i)* $\mathcal{O}$ - **object nodes** (*e.g.*, cars, buildings) with category, location, orientation, and appearance; *(ii)* $\mathcal{G}$ - **ground nodes** (road/sidewalk) with spatial extent and appearance; and *(iii)* $\mathcal{S}$ - a **sky node** capturing illumination and distant background. Distilled from real-world videos, this graph serves as a compact *blueprint* for guiding simulation scene generation.

**Overview.** As shown in Fig. 4, UrbanVerse-Gen operates in three stages: (1) *distillation*, where object semantics and 3D layout, ground composition (road/sidewalk) and appearance, lighting, and distant background are extracted from the input video into a unified scene graph representation; (2) *materialization and diversification*, where multiple digital cousin assets from UrbanVerse-100K are matched and bound to each graph instance; and (3) *generation*, where object, ground, and sky instances are assembled with their extracted spatial information into physically plausible scenes. We detail each stage below. Further technical specifics are in App. E.

**Real-World Scene Distillation.** To accurately parse semantics and estimate 3D layouts from uncalibrated open-world videos, we design a distillation module that integrates 2D open-vocabulary foundation models with SfM through 3D lifting. Given an RGB city-tour video $\mathcal{I} = \{I_t^{\text{rgb}}\}_{t=1}^{T}$, we sample every third frame and query GPT-4.1 (OpenAI, 2025) to enumerate visible categories and form a candidate vocabulary. The video is lifted to metric 3D by estimating depth, intrinsics, and SE(3) poses using MASt3R (Leroy et al., 2024). With these semantic and geometric estimates, we assemble an open-vocabulary object parser using YoloWorld (Cheng et al., 2024) and SAM 2 (Ravi et al., 2024)) to obtain per-frame instance masks, which are lifted to metric 3D. Identical detections are fused across frames by semantic similarity and point-cloud overlap, yielding $N$ persistent object nodes $\mathcal{O} = \{\mathbf{o}_i = \langle \mathbf{l}_i, \mathbf{c}_i, \mathbf{b}_i, \theta_i, \mathcal{M}_i \rangle\}_{i=1}^{N}$, where $\mathbf{l}_i$ is the category, $\mathbf{c}_i$ the centroid, $\mathbf{b}_i$ the oriented 3D box, $\theta_i$ the yaw, and $\mathcal{M}_i = \{\mathbf{m}_{i,j}\}_j$ the 2D object crops. In parallel, we segment

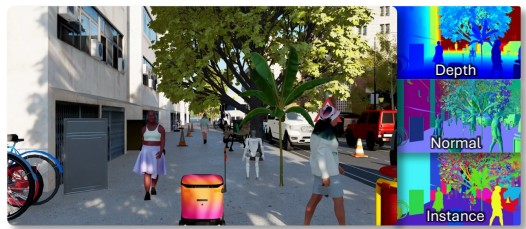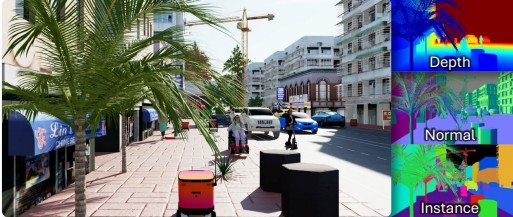

Figure 5: Diverse dynamic agents in UrbanVerse scenes. Two example scenes populated with pedestrians, cars, wheelchair users, and scooter riders, shown across multiple observation modalities.

road and sidewalk using a panoptic Mask2Former ground parser (Cheng et al., 2022) trained on Cityscapes (Cordts et al., 2016), lift and fuse results into metric point clouds $\mathbf{p}$, and define ground nodes $\mathcal{G} = \{\langle \text{road}, \mathbf{p}_{rd}, \mathcal{M}_{rd} \rangle, \langle \text{sidewalk}, \mathbf{p}_{sw}, \mathcal{M}_{sw} \rangle\}$, where $\mathcal{M}$ preserves ground masks. Finally, the sky parser captures global illumination and distant background by cropping the upper half of each frame, stored in the sky node as $\mathcal{S} = \{X_t\}_{t=1}^{T}$.

**Scene Materialization and Diversification.** With the distilled scene graph, our goal is to materialize each instance using *digital-cousin* assets from UrbanVerse-100K that are semantically aligned, geometrically consistent, and visually faithful yet diverse, enabling varied appearances for stronger policy generalization. To meet these requirements, we retrieve $k_{\text{cousin}}$ assets for each object node $\mathbf{o}_i \in \mathcal{O}$ through three corresponding steps: *(i) semantic matching* selects the best-matched asset category via CLIP similarity (Radford et al., 2021) with its label $\mathbf{l}_i$; *(ii) geometry filtering* ranks candidates within that category by minimal Bounding Box Distortion (mBBD) between $\mathbf{b}_i$ and candidate boxes, retaining the top 1,000; *(iii) appearance selection* re-ranks these candidates using DINOv2 similarity (Oquab et al., 2023) between $\mathcal{M}_i$ and asset thumbnails, keeping the top-$k_{\text{cousin}}$ as final matches. For ground nodes, we retrieve $k_{\text{cousin}}$ PBR materials by comparing pixel-wise MSE between road/sidewalk crops $\mathcal{M}_{rd}/\mathcal{M}_{sw}$ and rendered thumbnails. For the sky node, we select $k_{\text{cousin}}$ HDRIs by matching HSV histograms of sky crops $\{X_t\}$ to HDRI thumbnails, reproducing illumination and distant background.

**Simulation Scene Creation.** Finally, we instantiate the materialized graph into $k_{\text{cousin}}$ simulated scenes in UrbanSim (IsaacSim backend) by: *(i) ground fitting and texturing*, where road and sidewalk planes are fitted from $\mathbf{p}_{rd}/\mathbf{p}_{sw}$, sidewalks are elevated by $15\,\text{cm}$, and surfaces are textured with the selected PBR materials; *(ii) lighting and surroundings*, where a matched HDRI sky map is used as both a dome light source and a spherical environment to provide realistic skies and distant context; *(iii) object placement*, where each object is positioned at its centroid $\mathbf{c}_i$, aligned to the canonical front and heading $\theta_i$, and adjusted to avoid penetrations. We then assign annotated physical parameters (*e.g.*, mass, friction) and enable rigid-body dynamics, producing stable, interactive scenes. With metrically scaled assets from UrbanVerse-100K aligned to extracted layouts, the resulting scenes are true-to-scale, grounded in real-world layouts, and ready for E-AI agents to explore.

**Interactive Dynamic Agent Population.** UrbanVerse also supports dynamic agents such as pedestrians, cars, wheelchair users, and scooter riders. Following UrbanSim (Wu et al., 2025b), we use a GPU-accelerated ORCA-based planner (Van Den Berg et al., 2011) to populate scenes with agents that move realistically and interact with the robot. For each scene, we build a 2D occupancy map, sample start–goal pairs, and compute collision-free paths; agents then continuously adjust their velocities during simulation for smooth, collision-aware motion. As shown in Fig. 5, this scene-agnostic mechanism enables diverse dynamic agents across all UrbanVerse environments.

### 3.3 URBANVERSE SCENE LIBRARY

**UrbanVerse Scene Library Construction.** Using UrbanVerse, we build a *training* library of 160 simulation scenes grounded in real-world distribution. We have collected 32 city-tour videos from YouTube under Creative Commons License, spanning 7 continents, 24 countries, and 27 cities. Each 3-min clip is distilled into a layout-grounded scene using our UrbanVerse-Gen and expanded into $k_{\text{cousin}} = 5$ digital cousin variants, yielding $5 \times 32 = 160$ scenes. See construction details in App. D.1.

**UrbanVerse Benchmark.** Further, to enable closed-loop evaluation, we construct a benchmark that comprises: *(i) AutoBench*, 10 scenes automatically generated from hold-out city-tour videos using

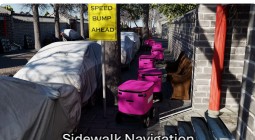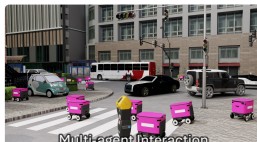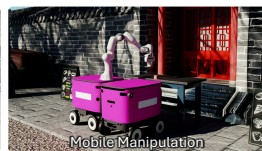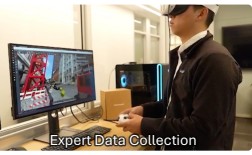

Figure 6: Urban embodied-AI tasks supported by the UrbanVerse simulation platform.

UrbanVerse-Gen; and *(ii) CraftBench*, 10 artist-designed scenes reserved for test-only evaluation. As shown in Fig. 32 in appendix, CraftBench spans diverse scenarios, layouts, cultural context, and safety-critical edge cases. To avoid bias, designers had no access to our assets and scenes.

**UrbanVerse Tasks.** In this work, we focus on *urban navigation* as the primary case study in our experiments because it most clearly demonstrates the benefits of UrbanVerse's realistic layouts and asset distributions. However, all assets in UrbanVerse include semantic labels, physical parameters, and affordance tags, enabling a wide range of urban embodied tasks. As the few examples illustrated in Fig. 6, UrbanVerse naturally supports multi-agent interaction, mobile manipulation, and expert data collection for imitation learning, opening broader research opportunities.

# 4 EXPERIMENTS

We evaluate UrbanVerse on three aspects: *(i)* **scene construction capability**, assessing fidelity and quality of scenes constructed from real-world videos (Sec. 4.1); *(ii)* **scaling capability**, examining whether training on UrbanVerse scenes follows scaling laws that improve policy generalization (Sec. 4.2); and *(iii)* **sim-to-real transfer capability**, testing whether policies trained in UrbanVerse enable robust and stable transfer to real-world environments (Sec. 4.3).

**Policy learning for mapless urban navigation.** In our study, following Wu et al. (2025b), we focus on RL to exploit scene interactivity, and study the task of *position-goal urban navigation*: the agent starts from a known ground-plane pose, receives a goal and waypoints sampled every 5 m from GPS projected to a local metric frame, and must learn a policy to reach the goal within a distance tolerance while avoiding collisions and staying on traversable surfaces. Using the 160-scene UrbanVerse library, we train vision-only navigation policies using PPO (Schulman et al., 2017) to show the effectiveness of our real-world grounded simulation scenes. During training, we load 16 different scenes at a time and repeat each scene 4–6 times depending on the available GPU memory. The set of scenes is changed every 100 RL episodes to expose the policy to diverse environments. In both training and testing, the agent is provided with *only RGB* observation, its relative position to the goal, *without* access to the global map. See model architecture and training details in App. L.3.

## 4.1 SCENE CONSTRUCTION FIDELITY AND QUALITY

**Scene fidelity evaluation.** We first evaluate whether UrbanVerse can faithfully recover scene semantics and layouts from video. Using 45 KITTI-360 (Liao et al., 2022) sequences (average length 198.7 m) of residential and city streets, we generate digital cousin scenes with UrbanVerse-Gen. Scene reconstruction fidelity is measured by comparing the nearest digital-cousin scene, built with top-1 matched assets, against ground-truth annotations: semantic fidelity by the proportion of correctly preserved categories; layout fidelity by per-object pose errors ($\mathcal{L}_2$ distance and orientation difference); geometric fidelity by bounding-box volume difference; and overall recovery by 3D detection mAP25 (Kumar et al., 2024). Appearance fidelity cannot be directly measured since real-world asset replicas are unavailable. Instead, we input walkthrough videos of ten simulated CraftBench scenes into UrbanVerse-Gen and report the proportion of objects whose 3D models are exactly retrieved. For comparison, we evaluate two SfM models (VGGT (Wang et al., 2025) and our default MASt3R) and two open-vocabulary parsers (GroundedSAM2 (Ren et al., 2024) and our default YoWorldSAM2, combining YoloWorld with SAM2). Lastly, we present side-by-side comparisons of city-tour videos and their reconstructed scenes in Fig. 7.

Quantitatively, Tab. 2 shows that the MASt3R with our YoWorldSAM2 parser yields the best overall results, which we adopt as UrbanVerse's default. With this setup, 93.1% of object categories are correctly preserved; reconstructed objects deviate by only 1.4 m in position, 19.8° in orientation, and 0.8 m³ in volume. Asset retrieval achieves 75.1% accuracy, indicating effective matching of

| SfM | Scene Parser | Cat. (%)↑ | Dist. (m)↓ | Ori. (°)↓ | Scale (m³)↓ | mAP25↑ | Ast. (%)↑ |
|---|---|---|---|---|---|---|---|
| VGGT | GroundedSAM2 | 88.2 | 2.4 | 21.5 | 1.5 | 7.5 | 67.5 |
| | YoWorldSAM2 | 91.5 | 2.1 | 20.1 | 1.3 | 9.4 | 70.6 |
| MASt3R | GroundedSAM2 | 86.1 | 2.1 | 19.9 | 1.1 | 24.3 | 68.5 |
| | **YoWorldSAM2** | **93.1** | **1.4** | **19.8** | **0.8** | **28.2** | **75.1** |

Table 2: Scene reconstruction fidelity evaluation of nearest digital-cousin generation. We report average results on KITTI-360, including: Category recovery (Cat.), the fraction of correctly categorized objects; Distance (Dist.), Orientation (Ori.), and Scale, the mean differences in centroid position, heading, and volume between recovered and ground-truth bounding boxes; and 3D detection mAP. Asset recovery (Ast.), the fraction of correctly retrieved objects evaluated from CraftBench scene videos, is also reported.

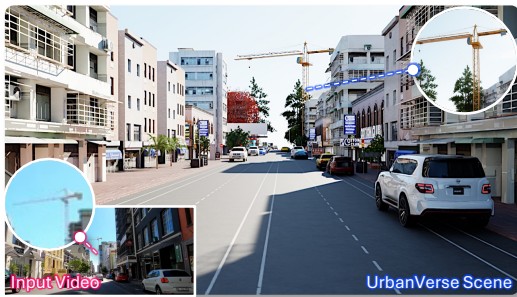
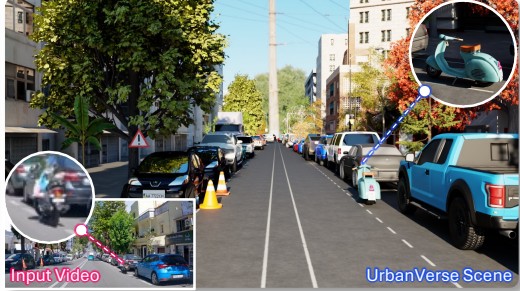

Figure 7: Qualitative scene generation results of UrbanVerse. Scenes generated from Cape Town (left) and Morocco (right) city-tour videos in our library. Highlighted details are shown in the circled areas. See Fig. 28 in the Appendix for additional qualitative results.

visually similar 3D assets from our database. Qualitatively, Fig. 7 shows that UrbanVerse produces physically plausible scenes that preserve fine object placement details from the original videos, such as cranes located in the distance or motorcycles parked along the roadside. Given the long horizon length of the evaluated street scenes, these results demonstrate that UrbanVerse can faithfully capture real-world semantics and layout distributions from casual city-tour videos, enabling the generation of high-fidelity simulated scenes that reflect real-world street distribution.

**Human Evaluation of Scene Quality**. We evaluate whether UrbanVerse scenes align with human impressions of everyday streets through two user studies with 32 undergraduates. In the first comparative study, 100 UrbanVerse scenes sampled from our library are paired with 100 UrbanSim's procedurally generated (PG) scenes generated from the same UrbanVerse-100K assets. Participants view shuffled overview images and choose which scene is better (or equally good) in terms of object diversity, layout coherence, and overall realism. In the second study, participants rate the overall realism of 30 scenes, 10 from UrbanVerse, 10 from PG, and 10 artist-designed oracles from CraftBench, by watching 360° walkthrough videos and scoring them on a 1–5 scale. Further details are provided in App. M.

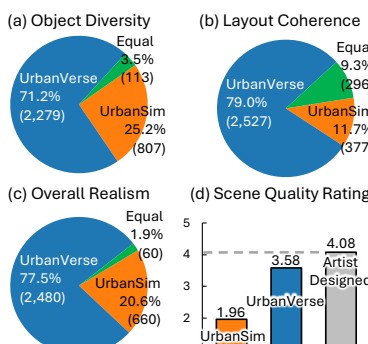

Figure 8: Human evaluation results.

As shown in Fig. 8 (a–c), participants consistently preferred UrbanVerse over PG, with more than 70% favoring it across Object Diversity, Layout Coherence, and Overall Realism. In the second study, the human ratings of artist-designed scenes (4.08/5.00) in Fig. 8 (d) suggest that even hand-crafted scenes are imperfect, underscoring the challenge of creating highly realistic outdoor environments. Given this difficulty, the 3.58/5.00 score of UrbanVerse scenes is satisfactory for an automated approach.

## 4.2 SCALING URBANVERSE FOR POLICY GENERALIZATION

We next examine the scaling properties of using the UrbanVerse 160-scene library for policy generalization in mapless urban navigation on a wheeled robot. We first study how the number of training layouts influences generalization, then analyze the effect of expanding each layout with more digital cousins. Policy performance is measured by success rate (SR), route completion (RC), and collision times (CT), with all evaluations conducted in *unseen* environments from AutoBench and CraftBench. We provide detailed experimental setups and metric definitions in App. L.

Figure 9: Generalization by scaling training layouts and digital cousins per scene.

**Scaling with more unique layouts.** We fix the number of cousins per layout and vary the number of unique layouts. From the 160-scene UrbanVerse library, we select 32 unique layouts (each from a distinct city-tour video), each expanded with five cousins. Training sets are constructed with $N \in \{1, 8, 16, 32\}$ layouts, corresponding to 5, 40, 80, and 160 scenes. For comparison, we train policies on matched PG scenes in UrbanSim with identical layout counts and per-layout variants. Policies are then evaluated on the 10 CraftBench scenes with five runs per scene. As shown in Fig. 9 (a), increasing the number of UrbanVerse layouts consistently improves generalization, with success rates rising sharply as coverage expands. The flat slope at small $N$ highlights how limited layout diversity constrains generalization. In contrast, policies trained on PG layouts show very limited improvement, indicating that hard-coded templates lack the diversity needed to scale.

**Scaling with more digital cousins.** We fix the number of unique layouts and vary the number of digital cousins per layout. Using 32 layouts, we select $m \in \{1, 2, 3, 4, 5\}$ top-ranked cousins, yielding training sets of 32–160 scenes. Policies trained on these sets are evaluated on 10 AutoBench scenes and 10 CraftBench scenes. As shown in Fig. 9 (b), increasing the number of cousins consistently further boosts success rate, confirming that intra-layout diversity complements inter-layout diversity to reinforce generalization. Performance is overall higher on AutoBench, reflecting distributional familiarity with scenes generated by the same pipeline, while the lower scores on CraftBench highlight the difficulty of artist-designed environments. Notably, the narrowing gap between the two testbeds indicates that greater per-layout diversity improves robustness to distribution shift.

**Validating scaling power laws.** We next validate whether performance gains follow a power-law scaling relationship (Lin et al., 2025b). Defining test error as $E = 1 - \text{SR}$ and training scale as $N$ (number of layouts or cousins), a power law holds if $E = \beta N^{-\alpha}$, which becomes linear after log transform: $\log E = -\alpha \log N + \log \beta$. As shown in Fig. 9 (c, d), linear fits in log–log space confirm clear power-law behavior for both layout and cousin scaling, with strong Pearson correlations.

**Benchmarking overall generalization.** We compare our strongest policy (PPO-UrbanVerse), trained on all 160 UrbanVerse scenes, against foundation models MBRA (Hirose et al., 2025), CityWalker (Liu et al., 2025), S2E (He et al., 2025), and a PPO policy trained on 160 UrbanSim PG scenes, all evaluated on CraftBench. As an overfitting reference, we also train policies directly on each test scene. As shown in Tab. 3, PPO-UrbanVerse, despite its simple architecture, consistently outperforms all baselines, achieving a **+6.3%** SR gain over the second-best model

| Method | SR ↑ | CT ↓ | RC ↑ |
|---|---|---|---|
| MBRA | 35.6 | **25.6** | 52.9 |
| CityWalker | 29.2 | 38.2 | 48.6 |
| S2E | 33.1 | 27.7 | 55.7 |
| PPO-UrbanSim | 9.1 | 31.5 | 19.4 |
| **PPO-UrbanVerse** | **41.9** | 35.5 | **62.4** |
| Overfitting | 26.5 | 32.2 | 40.6 |

Table 3: Results on CraftBench.

MBRA. Policies trained directly on test scenes perform poorly on altered routes, revealing strong overfitting and underscoring the need for diverse training scenes to enable true generalization.

## 4.3 ZERO-SHOT SIM-TO-REAL POLICY TRANSFER

**Zero-shot transfer across urban spaces and embodiments.** We evaluate our strongest policy, trained on all UrbanVerse scenes, in 16 unseen real-world urban scenarios averaging 24.6 m per route (see Fig. 35 for examples). The same policy is deployed on two embodiments: the Coco wheeled delivery robot (Coco Robotics, 2024) and the Unitree Go2 quadruped (see Fig. 36 for robot configuration). We benchmark against navigation foundation models NoMad (Sridhar et al., 2024), S2E, and a PPO policy trained on PG scenes. Each evaluation is repeated three times, with distance-to-goal (DTG) also reported. As shown in Tab. 4, PPO-UrbanVerse significantly outperforms

| 🛒 Wheeled | SR ↑ | CT ↓ | RC ↑ | DTG ↓ |
|---|---|---|---|---|
| NoMad | 33.3 | 66.7 | 57.4 | 9.8 |
| CityWalker | 25.0 | 75.0 | 42.7 | 12.0 |
| S2E | 47.9 | 54.2 | 59.6 | 8.2 |
| PPO-UrbanSim | 18.8 | 81.3 | 34.6 | 11.8 |
| **PPO-UrbanVerse** | **77.1** | **22.9** | **83.4** | **3.6** |
| 🐾 Quadruped | SR ↑ | CT ↓ | RC ↑ | DTG ↓ |
| NoMad | 37.5 | 62.5 | 54.4 | 12.5 |
| CityWalker | 31.3 | 66.8 | 42.6 | 13.4 |
| S2E | 58.6 | 41.7 | 71.4 | 6.3 |
| PPO-UrbanSim | 18.8 | 81.3 | 29.1 | 15.4 |
| **PPO-UrbanVerse** | **89.7** | **10.4** | **86.4** | **2.5** |

Table 4: Real-world results.

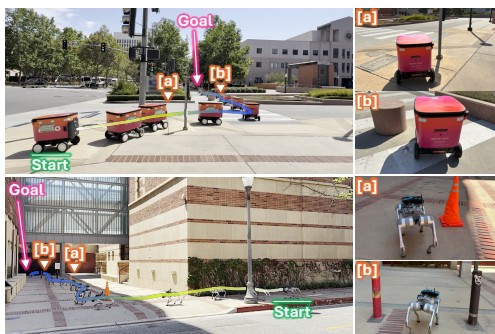

Figure 10: Visualization of real-world results.

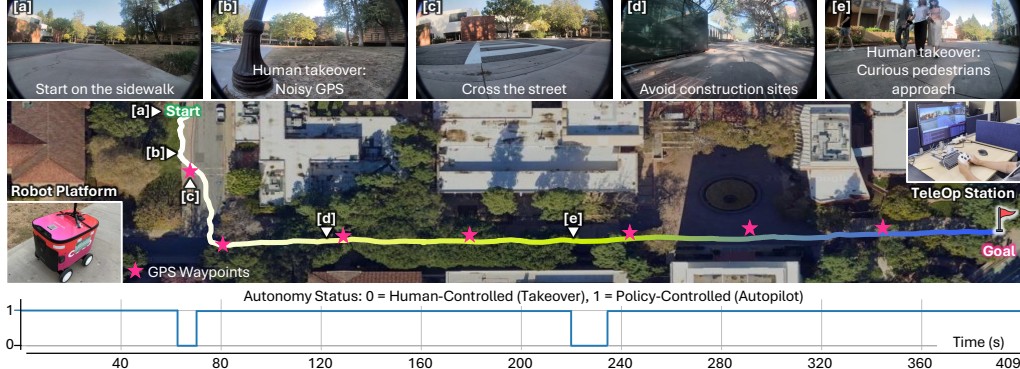

Figure 11: Long-horizon mapless urban navigation. The PPO-UrbanVerse policy pilots the Coco robot across 337 m of public urban space, reaching the goal with only two human interventions and no collisions.

all baselines, despite its simple architecture, it surpasses the second-best model (S2E) by **+29.2%** SR on Coco and **+31.1%** SR on Go2. Foundation models like CityWalker, trained on large-scale but non-interactive data, succeed mainly in obstacle-free cases and fail when obstacles appear. In contrast, PPO-UrbanVerse consistently demonstrates robust obstacle avoidance (*e.g.*, bypassing bollards after turns or while crossing streets). These results show that interactive capabilities learned via RL in UrbanVerse transfer effectively and reliably to real-world settings in a zero-shot manner.

**Long-horizon mapless urban navigation.** To stress-test policy stability, we deploy PPO-UrbanVerse on the Coco robot for a 337 m mission in real urban spaces. The robot follows GPS waypoints at 10 m intervals. For safety in public streets we implement a human–AI shared autonomy TeleOp system that allows real-time human intervention when needed. As the route recording shown in Fig. 11, it successfully completes the task with only *two* interventions Completing such a challenging task in public streets demonstrates the robustness of policies trained on UrbanVerse scenes, a stability we attribute in part to its long training routes (≈200 m) that encourage generalizability for long-horizon tasks. This highlight UrbanVerse's potential for training versatile, practical urban agents.

## 5 CONCLUSION

We introduce UrbanVerse, a data-driven real-to-sim system that brings our daily messy streets into interactive simulation environments. Leveraging the curated UrbanVerse-100K and the automated UrbanVerse-Gen pipeline, UrbanVerse can mass-produce simulated scenes that faithfully capture real-world distribution, enabling effective policy scaling and more generalizable urban AI embodiments.

**Limitation.** UrbanVerse is currently tailored to street-level urban environments; extending it to parks, campuses, or indoor-outdoor transitions would require additional terrain modeling and access structures, which we consider a promising direction for future work. In addition, our UrbanVerse-Gen real-to-sim pipeline can be affected by rare challenging video conditions, such as low light, fast motion, or heavy occlusion—that may introduce depth and pose drift or imperfect object placement.

## ACKNOWLEDGMENTS

This project was supported by NSF Grants CNS-2235012, IIS-2339769, CCF-2344955, TI-2346267, and ONR grant N000142512166. Mingxuan Liu and Elisa Ricci were partially supported by the EU Horizon projects ELIAS (No. 101120237) and ELLIOT (No. 101214398). Honglin He was supported by the Amazon Trainium Fellowship. We gratefully acknowledge Coco Robotics for their generous equipment donation. We thank Yukai Ma for his support with the robotic infrastructure and Jack He for his assistance with the VR demo. We also acknowledge CINECA and the ISCRA initiative for providing high-performance computing resources.

## ETHICS STATEMENT

All city-tour videos used in this work are sourced from YouTube platforms that provide free Creative Commons licenses. Prior to use, we apply automated and manual filtering to remove any frames containing human faces, license plates, or other identifiable information, ensuring that no personally sensitive data is retained. Our focus is solely on urban layouts, object distributions, and physical attributes. While our system enables scalable simulation for embodied AI, potential misuse (*e.g.*, surveillance applications) must be acknowledged. We therefore emphasize responsible use of our released assets, code, and data strictly for research purposes in urban simulation and embodied AI.

## REPRODUCIBILITY STATEMENT

To ensure reproducibility, we will release the full UrbanVerse-100K (annotated 3D assets, ground materials, and skyboxes), the UrbanVerse-Gen implementation code, and the 160-scene UrbanVerse library constructed from city-tour videos. All experiments are documented with dataset splits, training details, and hyperparameters. Scripts for preprocessing, scene generation, and policy training will be included, along with instructions for reproducing results on KITTI-360 and real-world deployment tests. By open-sourcing our resources, we aim to support and accelerate embodied AI research in urban environments

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

APPENDIX

This appendix provides additional demonstrations, visualizations, statistics, experiments, and implementation details that complement the main paper. App. A summarizes the supplementary demonstration videos and interactive visualizations. App. B presents the complete user-side pipeline for generating and using UrbanVerse simulation scenes. App. C provides extended statistics and annotation details of the UrbanVerse-100K database, together with a discussion of scale and quality issues in existing datasets and a quantitative validation of our physical attribute annotations. App. D offers further construction details for the 160-scene UrbanVerse library and the artist-designed CraftBench benchmark, along with additional qualitative examples. App. E describes implementation details of the UrbanVerse-Gen pipeline. App. F analyzes typical failure cases and reconstruction challenges. App. G visualizes the real-world testing scenes and outlines their selection strategies. App. H provides additional studies on the horizon length of training scenes. App. I further reports results on policy learning and transfer in customized real-world scenes, while App. J provides expanded real-world results for each test scene. App. K presents computational and scalability analyses of UrbanVerse. App. L reports the evaluation metrics, training and evaluation setup, policy learning details, and robot configurations in both simulation and the real world. Finally, App. M describes the human evaluation protocols and interface.

## LLM Usage Declaration

In our system, GPT-4.1 was employed as a tool for category listing from input videos (with all identity information masked) and for annotating semantic, affordance, and physical attributes of virtual 3D assets. For writing, GPT-4o was used to check grammar mistakes.

## A    Demonstration Video and Interactive Visualization

We encourage readers to explore the videos and interactive demonstrations available on our **project page here**, which showcase detailed examples of the scene distillation process, generated simulation scenes, the proposed asset database, and real-world navigation policy performance.

## B    User-side Pipeline

In this section, we outline the full user-side pipeline for using the UrbanVerse simulation platform. As illustrated in Fig. 12, UrbanVerse supports both automatic generation of new simulation scenes from raw video inputs and direct use of built-in scene repositories, enabling a broad range of embodied AI applications.

**Generating Custom Simulation Scenes.** Users can create their own simulation environments directly from raw inputs. UrbanVerse-Gen accepts diverse sources such as YouTube city-tour videos, mobile-recorded walk-through clips, or folders of RGB frames. After providing the input, users simply use the UrbanVerse-Gen API to automatically convert the video into a fully interactive, physics-ready simulation scene. The pipeline handles all steps internally—from video normalization to layout extraction and scene materialization—requiring no manual editing. UrbanVerse-Gen supports generating multiple "digital cousins" from one video.

**Using Built-in Scene Repositories.** UrbanVerse also provides two ready-to-use simulation libraries. **UrbanVerse-160** contains 160 automatically generated real-to-sim scenes extracted from city-tour videos across the world. **CraftBench** provides 10 high-fidelity artist-designed scenes for benchmarking robustness, generalization, and navigation difficulty. Both repositories can be loaded directly without running UrbanVerse-Gen.

**Downstream Tasks Supported by UrbanVerse** With either custom-generated scenes or the built-in repositories, users can seamlessly train and evaluate embodied agents. UrbanVerse supports reinforcement learning (e.g., PPO) for navigation and interaction, as well as imitation learning through expert demonstration collection via keyboard, joystick, gamepad, or VR teleoperation. The simulator also facilitates large-scale multimodal dataset collection (RGB, depth, normals, segmentation, LiDAR, and poses) for perception training. Finally, trained policies can be evaluated in closed-loop within UrbanVerse scenes and deployed to real robots for zero-shot sim-to-real transfer.

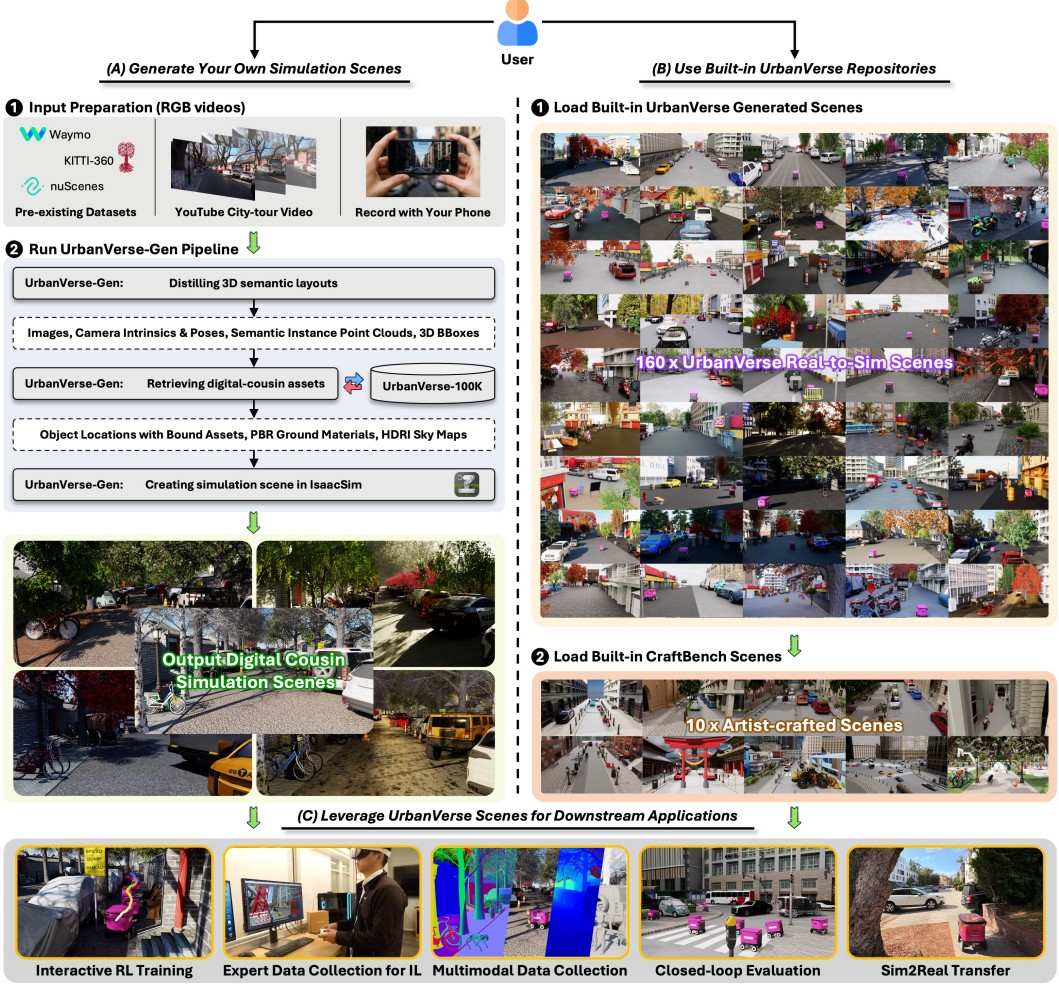

Figure 12: **User-side pipeline of the UrbanVerse simulation platform.**

Overall, UrbanVerse provides a practical, flexible, and complete user-side pipeline that spans real-to-sim scene generation, large-scale simulation assets, embodied task execution, and real-world deployment.

## C  DETAILS OF URBANVERSE-100K ASSET DATABASE

### C.1  URBANVERSE-100K STATISTICS

To provide an overview of the rich semantic coverage of UrbanVerse-100K, we visualize the category distribution of its curated 102,444 object assets under our introduced three-level urban ontology, along with a few examples sampled from the database.

Fig. 13 shows the hierarchical breakdown of categories across all the three levels, spanning broad groups such as *buildings*, *vehicles*, *street users*, *barriers*, *amenities*, and *urban objects*, with finer-grained divisions into 659 leaf categories. Complementing this, Fig. 14 presents a word cloud of all 659 leaf-level categories, where font size reflects category frequency.

Finally, we showcase a few 3D urban object assets sampled from the object collection of UrbanVerse-100K in Fig. 15, Fig. 16, Fig. 17, and Fig. 18; several ground PBR materials sampled from the ground collection in Fig. 19 (road) and Fig. 20 (sidewalk); and a few sky HDRI maps sampled from the sky collection in Fig. 21. These examples illustrate the quality and diversity of the assets that constitute the foundation of our urban scene generation pipeline.

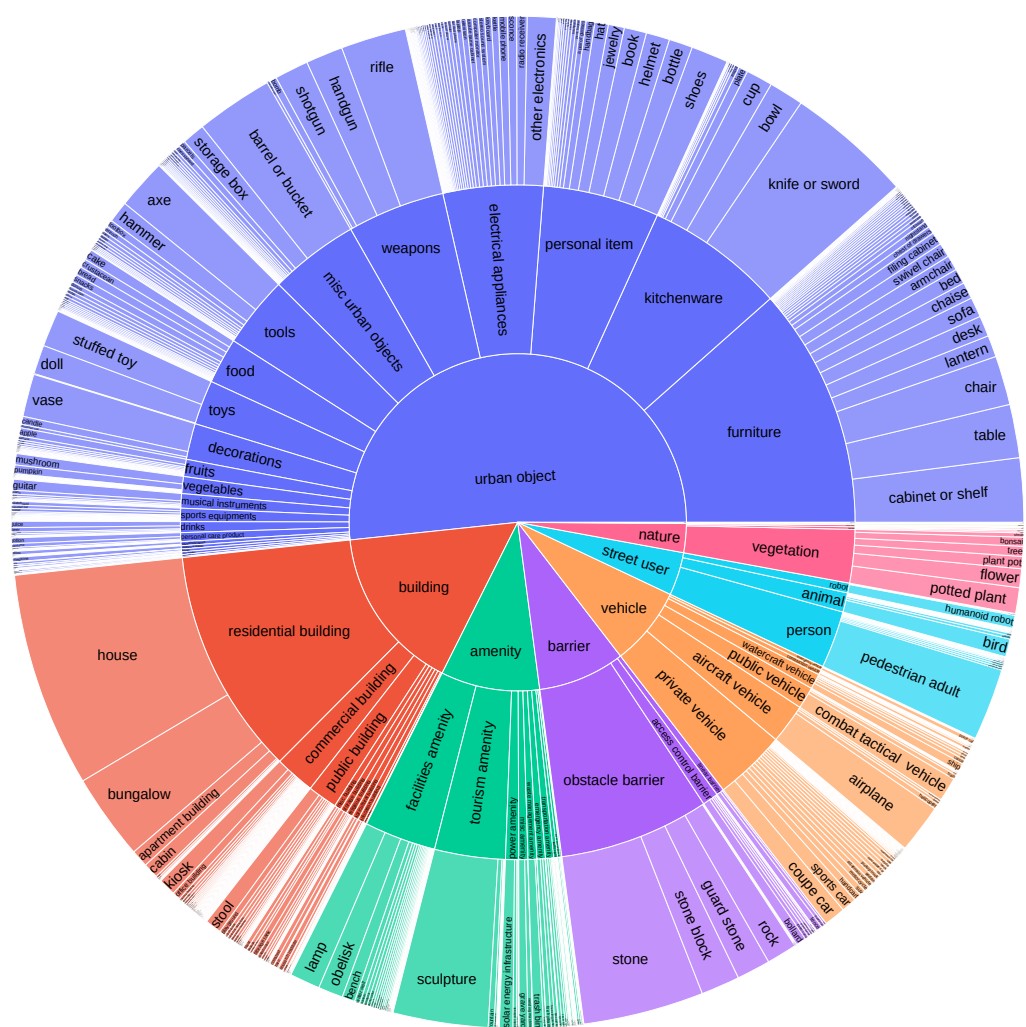

Figure 13: **Hierarchical category distribution of UrbanVerse-100K within our curated three-level urban .**

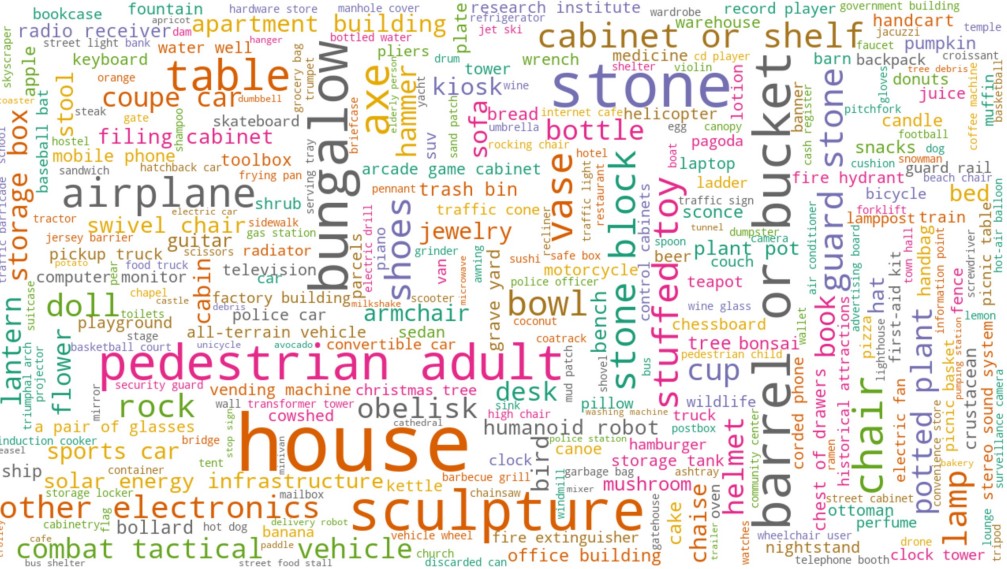

Figure 14: **Word cloud of UrbanVerse-100K category distribution over the 659 leaf-level categories.**

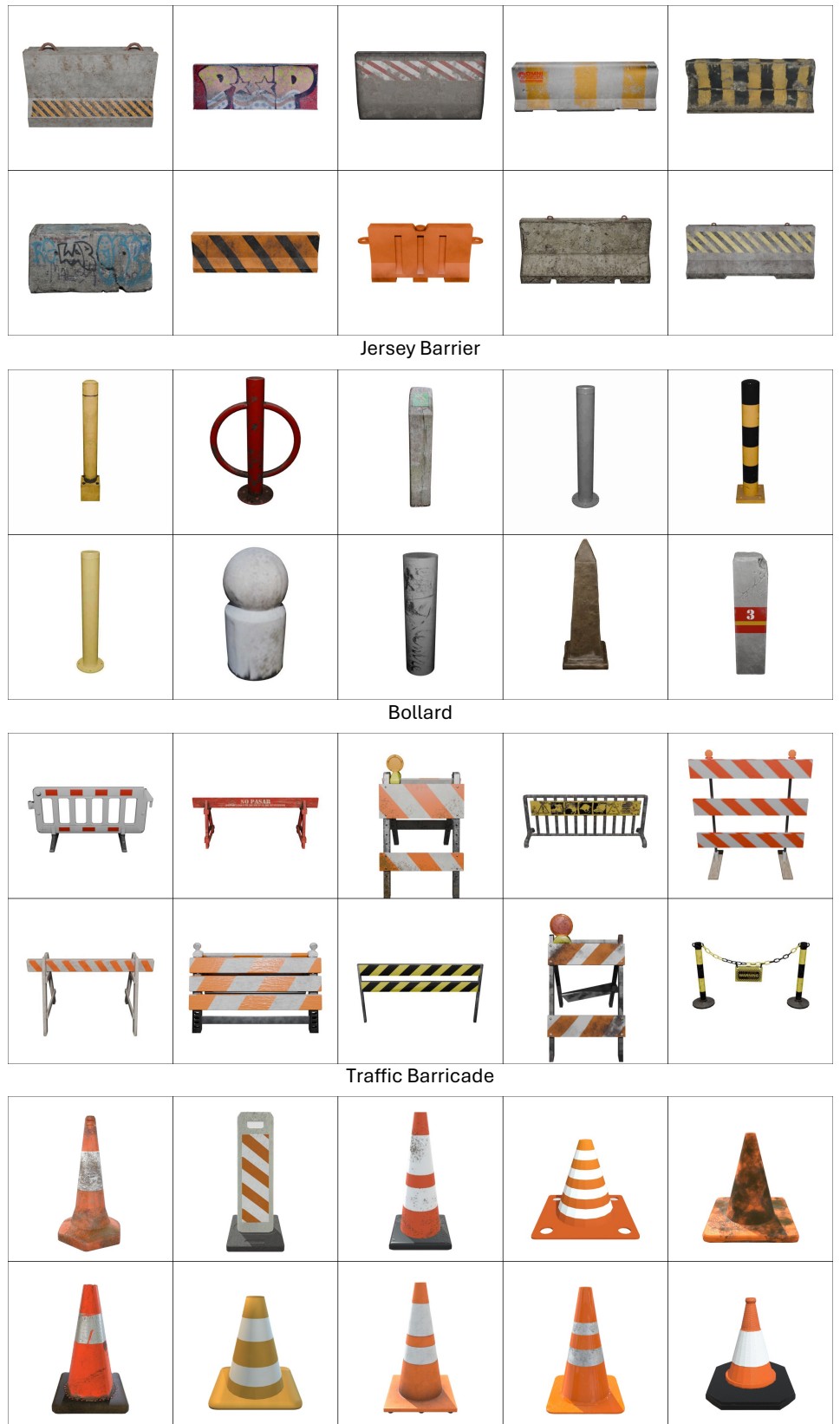

Jersey Barrier

Bollard

Traffic Barricade

Traffic Cones

Figure 15: **A few examples of UrbanVerse-100K objects — Urban Barriers.**

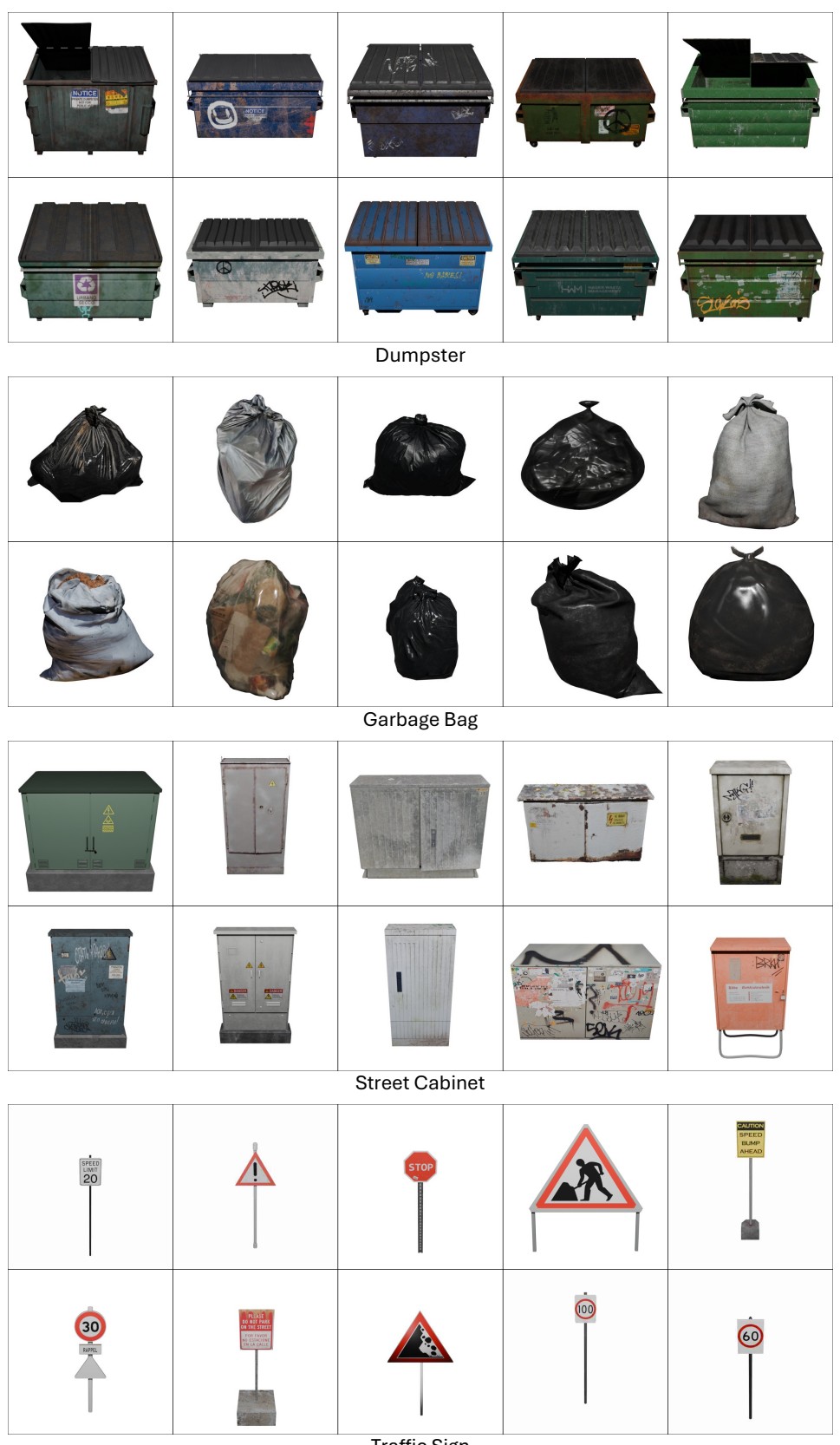

Figure 16: **A few examples of UrbanVerse-100K objects — Urban Street Objects.**

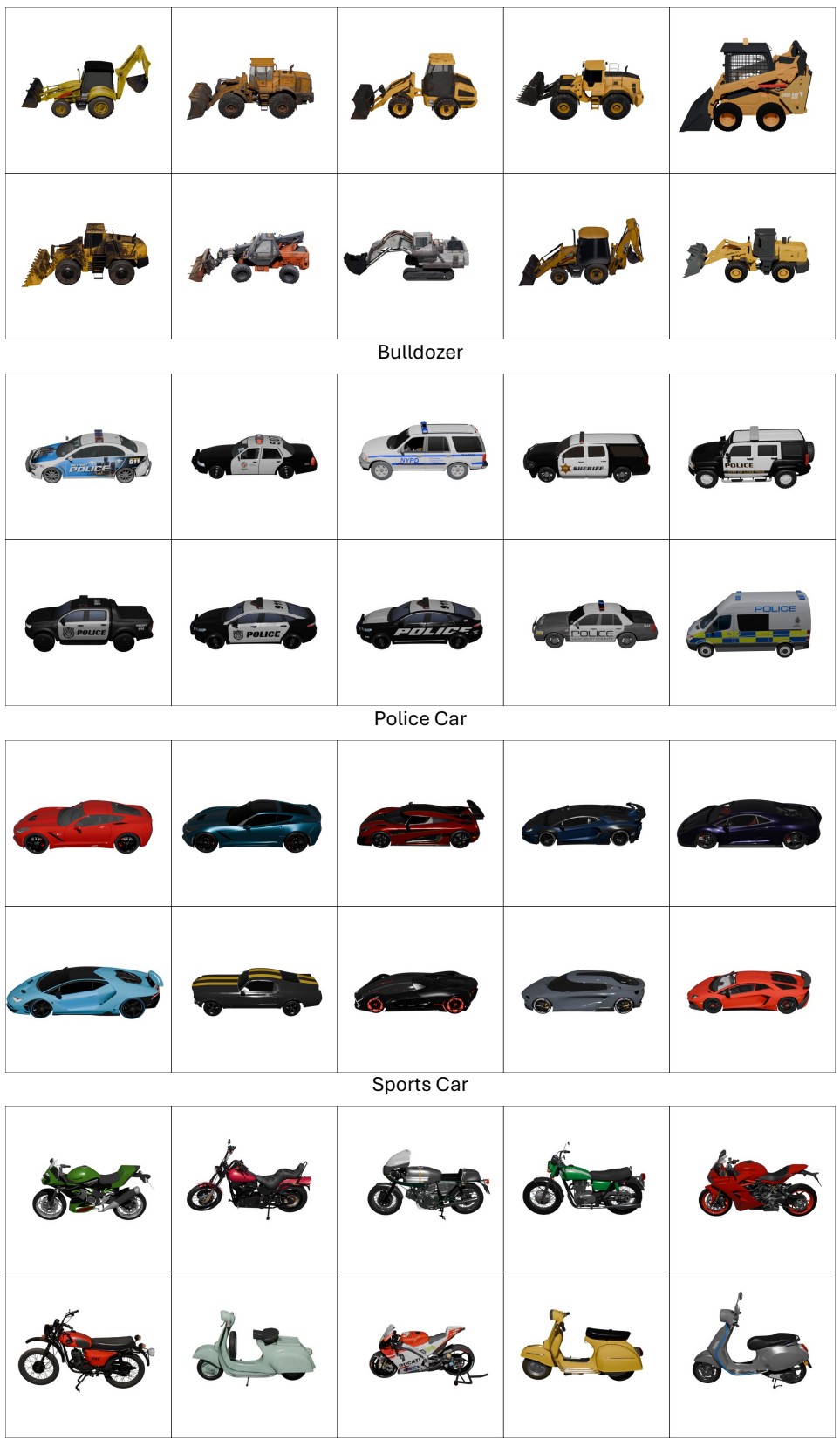

Bulldozer

Police Car

Sports Car

Motorcycle

Figure 17: **A few examples of UrbanVerse-100K objects — Urban Vehicles.**

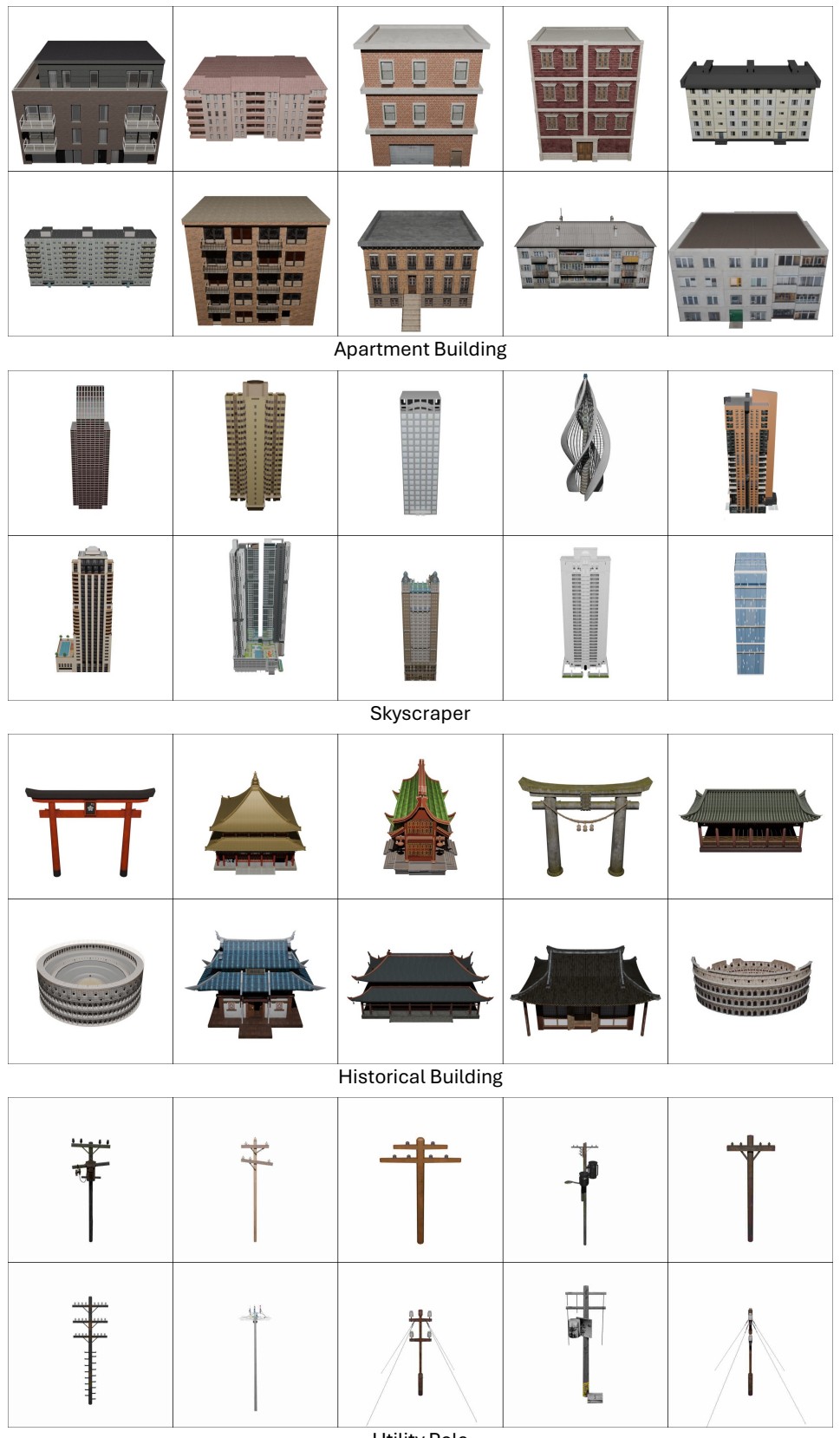

Figure 18: **A few examples of UrbanVerse-100K objects — Urban Structures.**

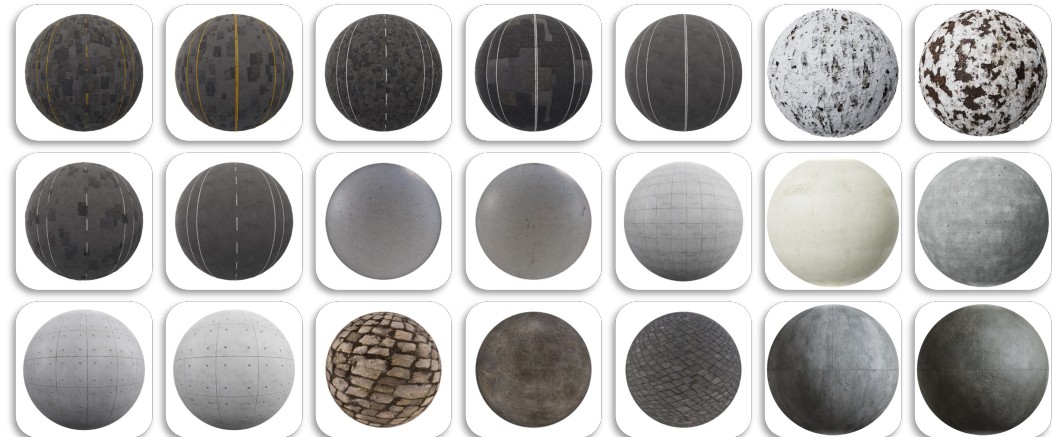

Figure 19: **A few examples of UrbanVerse-100K ground PBR materials — Road.**

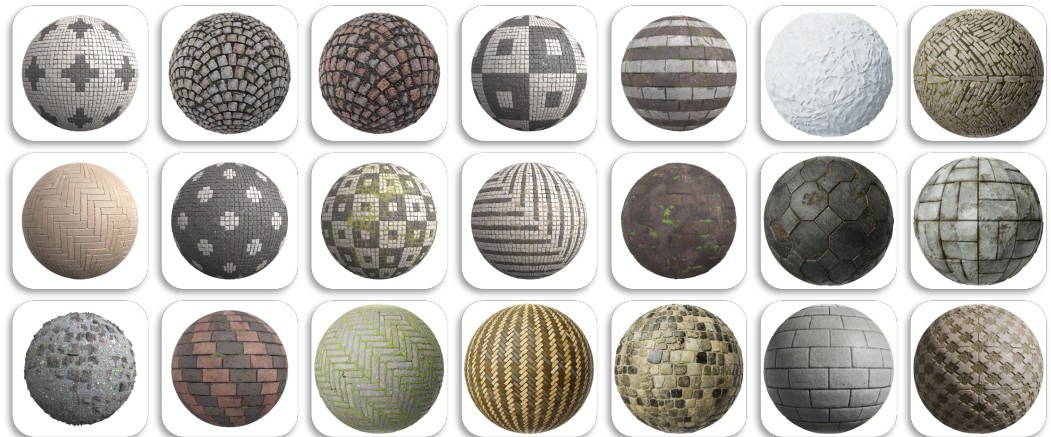

Figure 20: **A few examples of UrbanVerse-100K ground PBR materials — Sidewalk.**

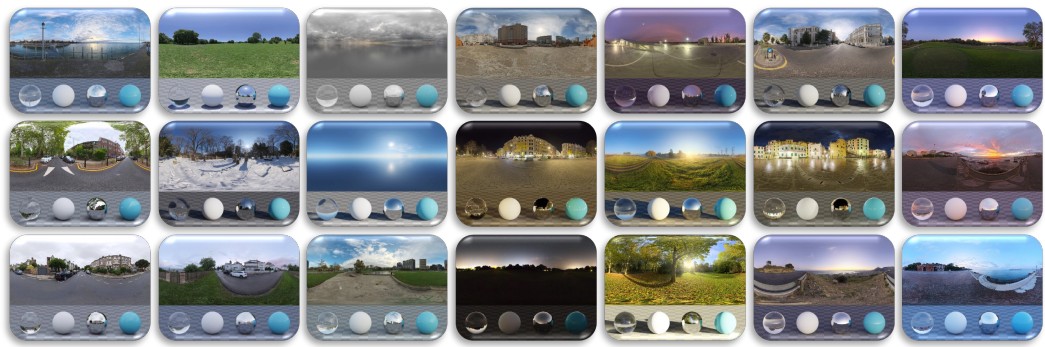

Figure 21: **A few examples of UrbanVerse-100K HDRI sky maps.**

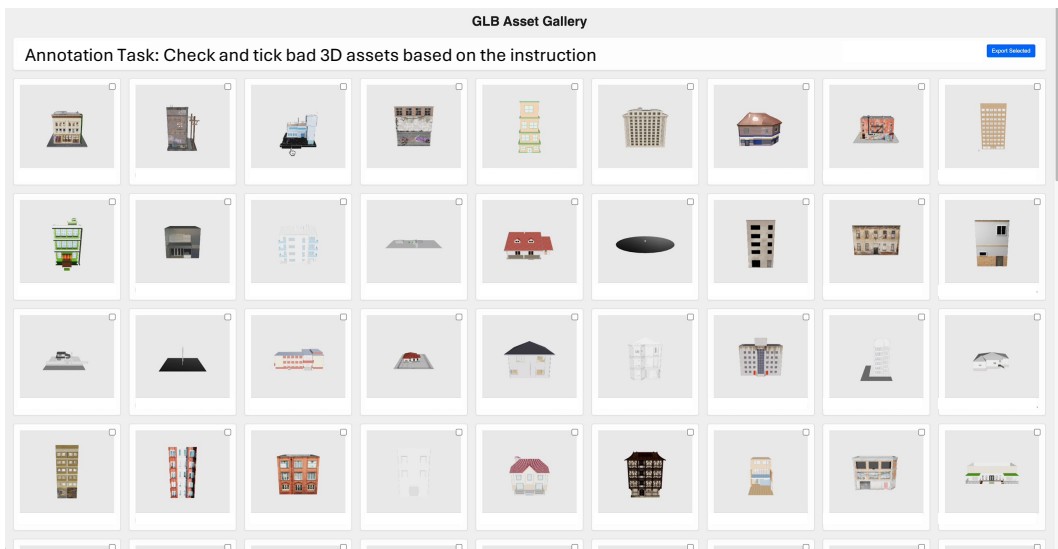

Figure 22: **Efficient Three.js–based annotation interface for asset quality filtering.**

## C.2 DETAILS OF URBANVERSE-100K ANNOTATION

Our goal is to curate a high-quality 3D urban asset database with accurate and semantically rich annotations from the 800K assets of Objaverse (Deitke et al., 2023c), thereby addressing the quality and scale issues discussed in Sec. C.3. Concretely, as described in Sec. 3.1, we design an efficient semi-automatic annotation pipeline consisting of three steps: (1) *Non-simulatable asset filtering*: in-house human annotators manually remove low-quality assets that could corrupt simulation; (2) *Urban ontology and categorization*: we construct an ontology of common urban semantics and categorize assets accordingly; and (3) *Attribute annotation*: we employ GPT-4.1 (OpenAI, 2025) to automatically annotate semantic, affordance, and physical attributes of the curated assets. We now describe each step in detail.

**(1) Non-simulatable asset filtering.** We first eliminate assets that are likely to fail in simulation by filtering out eight common corruption types. To support this process, we built a lightweight Three.js–based GLB gallery interface that allows annotators to quickly inspect assets and perform binary quality tagging. Ten in-house annotators worked for three weeks, resulting in a curated set of 158k simulatable 3D objects. As shown in Fig. 22, our interface enables rapid quality inspection and efficient removal of unusable assets. Prior to annotation, annotators underwent training to ensure consistent identification of low-quality assets.

**(2) Urban ontology and categorization.** We next construct a three-level urban ontology seeded from the OpenStreetMap (OSM) tag structure (Bennett, 2010). OSM is a collaborative, open-source project that provides freely accessible maps of the world, created by volunteers using GPS devices, aerial imagery, street-level photos, and local knowledge. Its tag structure includes common urban amenities, objects, and landmarks. Building on this structure, we expand the leaf level with categories drawn from ADE20K (Zhou et al., 2017), Cityscapes (Cordts et al., 2016), nuScenes (Caesar et al., 2020), LVIS (Gupta et al., 2019), and OpenImagesV7 (Kuznetsova et al., 2020). After deduplication and refinement, the resulting ontology contains 8 top-level, 59 mid-level, and 659 leaf categories. Assets are automatically classified into leaf classes using CLIP (Radford et al., 2021) applied to thumbnails, followed by human verification to prune non-urban objects (*e.g.*, weapons, spaceships) and correct misclassifications. This process yields 102,444 assets organized under our ontology.

**(3) Semantic, affordance, and physical attribute annotation.** Finally, guided by the question "How would a robot interact with this object?", we annotate each asset with 36 attributes spanning semantic, affordance, and physical properties in metric units, enabling physically plausible interactions and richer semantics. For each asset, we provide GPT-4.1 (OpenAI, 2025) with its thumbnail and four yaw snapshots at $0°/90°/180°/270°$, prompting it to produce attribute values. This step was completed

---

**Object Attributes Annotation Prompt**

Annotate this 3D asset assuming it can be found in an urban scenario (i.e., a street, a road, a sidewalk, a neighborhood, a garage, an park, etc), with the following values:

{
"annotations": {
  "**description_long**": a very detailed visual description of this [__category_l3__] object that is no more than 6 sentences. Don't use the term "3D asset" or similar here and don't comment on the object's orientation. Do use proper nouns when appropriate.,
  "**description**": a 1-2 summary of description_long, keep the description rich and visual,
  "**description_view_**": a short description of this [__category_l3__] object from view i (highlight/compare features that are different from other views),
  "**category**": [__category_l3__],
  "**height**": approximate height of this [__category_l3__] object in meters (m). Report the height for the object's orientation as shown in the images. For a standing human male this could be "1.75",
  "**max_dimension**": approximate maximum dimension of this [__category_l3__] object in meters (m). This is the longest dimension of this [__category_l3__] object, regardless of orientation. This should always be greater or equal to the height,
  "**materials**": a Python list of the materials that this [__category_l3__] object appears to be made of, taking into account the visible exterior and also likely interior (roughly in order of most used material to least used; include "air" if the object interior doesn't seem completely solid),
  "**materials_composition**": a Python list with the apparent volume mixture of the materials above (make the list sum to 1),
  "**mass**": approximate mass of this [__category_l3__] object in kilogram (kg) considering typical densities for the materials. For a human being this could be "72",
  "**receptacle**": a boolean indicating whether or not this [__category_l3__] object is a receptacle (e.g. a bowl, a cup, a vase, a box, a bag, etc). Return true or false with no explanations,
  "**frontView**": integer index of the view that represents the front of this [__category_l3__] object. This is typically the view from which you would approach the object to interact with it,
  "**quality**": a number, 0-10, indicating the quality of this [__category_l3__] object. 0 is very low quality (amateurish, confusing, missing textures, a 3D scan with many holes, etc), 10 is very high quality (professional, detailed, etc),
  "**movable**": a boolean indicating whether this [__category_l3__] object is movable or fixed/static in the environment. Return true or false with no explanations,
  "**required_force**": an approximate force in Newtons (N) required to move or push this [__category_l3__] object if it is movable. Base your estimate on the object's size, shape, and apparent material. Return 0 if the object is clearly immovable (e.g., a building or embedded structure),
  "**walkable**": a boolean indicating whether agents (e.g., humans, robots) can walk or move on this [__category_l3__] object (e.g., flat roofs, wide benches). Return true or false with no explanations,
  "**enterable**": a boolean indicating whether agents can physically enter this [__category_l3__] object (e.g., vehicles, doorways, booths, houses or buildings). Return true or false with no explanations,
  "**affordances**": a Python list of high-level functional affordances that this [__category_l3__] object provides (e.g., "sittable", "openable", "closable", "pressable", "**toggleable**", "drivable", "rotatable", "pushable", "pullable", "liftable"). Return only the most relevant affordances. Exclude general affordances such as "**enterable**", "walkable", and "movable", as they are annotated elsewhere. Return only the list, without explanation or additional text.,
  "**support_surface**": a boolean indicating whether this [__category_l3__] object can physically support other objects placed on it (e.g., tables, platforms, roofs). Return true or false with no explanations,
  "**interactive_parts**": a Python list of distinct functional parts or components of this [__category_l3__] object that can be interacted with (e.g., "handle", "drawer", "wheel", "door", "button"). Only include parts visible or implied from geometry. Return only the list, no explanation or other words,
  "**traversability**": a string label describing how an agent might traverse this [__category_l3__] object. Choose one of: "pass_through", "push_through", or "obstacle",
  "**traversable_by**": a Python list of agent types that can traverse or pass through this [__category_l3__] object (e.g., "person", "wheeled_robot", "drone"). Only include agents for which traversal is physically feasible. If the object is not traversable by any agent, return an empty list (i.e., []).,
  "**colors**": a Python list of the visible colors of this [__category_l3__] object (e.g., ["white", "gray", "blue"]). Focus on dominant and distinct colors visible from the exterior.,
  "**colors_composition**": a Python list of floats representing the approximate volume composition of the colors listed above. Ensure the list sums to 1.0 and corresponds to the order of "colors",
  "**surface_hardness**": a string describing the tactile hardness of the surface of this [__category_l3__] object. Choose one of: "soft", "semi-soft", or "hard",
  "**surface_roughness**": a float in the range [0, 1] indicating the micro-texture roughness of the surface of this [__category_l3__] object. 0 means perfectly smooth (e.g., polished glass), 1 means extremely rough (e.g., coarse stone),
  "**surface_finish**": a string describing this [__category_l3__] object's surface tactile/visual quality. Choose one of: "rough", "matte", "smooth", "glossy", "sleek", or "grippy",
  "**reflectivity**": a float in the range [0, 1] that controls how much light is reflected by the surface of this [__category_l3__] object. 0 means no visible reflection, 1 means mirror-like reflection,
  "**index_of_refraction**": a float representing the surface's optical index of refraction (IOR) of this [__category_l3__] object. Typical values range from 1.0 (air) to ~2.5 (diamond). Use realistic values based on material type. Higher values increase reflection and refraction at oblique angles,
  "**youngs_modulus**": approximate material stiffness of this [__category_l3__] object in Megapascals (MPa). Use realistic values inferred from material types (e.g., 1e7 for rubber, 2e11 for steel),
  "**friction_coefficient**": a positive float representing the estimated friction coefficient, based on this this [__category_l3__] object's material and surface finish (e.g., polished ice ≈ 0.01, plastic ≈ 0.3, wood ≈ 0.5, rubber ≈ 0.9, dry concrete ≈ 1.2),
  "**bounciness**": a float in the range [0, 1] representing the expected elasticity of the [__category_l3__] object upon impact. Higher values indicate more bounce (e.g., rubber ball ≈ 0.9), while lower values indicate minimal or no bounce (e.g., stone ≈ 0.0),
  "**recommended_clearance**": approximate safe buffer distance (in meters) that should surround this [__category_l3__] object when placed in a scene. This helps avoid collision or interference with agents,
  "**asset_composition_type**": a string describing the structural nature of this 3D asset. Choose one of: "single" (a standalone atomic object), "group" (a small collection of related objects), or "scene" (a full composite scene with layout),
}

Figure 23: **Prompt for object attribute annotation.**

at a total API cost of $1,334. The exact annotation prompt is shown in Fig. 23, and the full set of annotated attributes is presented in Fig. 24.

## C.3 DISCUSSION ON ASSET SCALE AND QUALITY ISSUES OF EXISTING DATABASES

**Asset Quality Issues.** High-quality 3D assets are essential for constructing realistic and physically accurate simulation environments. The recent proliferation of large-scale 3D repositories has made it possible to efficiently assemble datasets for diverse scene construction. Objaverse (Deitke et al., 2023c), for instance, provides over 800K 3D objects sourced from the Internet, and its extension Objaverse-XL (Deitke et al., 2023b) scales this collection to 10.2M unique objects from sources such as GitHub. However, because these assets are primarily scraped from the web, their quality is highly inconsistent and largely uncontrolled.

In the early stage of our project, we systematically examined Objaverse's 800K assets and identified nine recurring forms of corruption, which affect more than half of the collection. As illustrated in Fig. 25, these issues include:

1. **Bad mesh:** incompletely reconstructed assets (often from 3D Gaussian Splatting (Kerbl et al., 2023)), resulting in noisy, broken geometry;

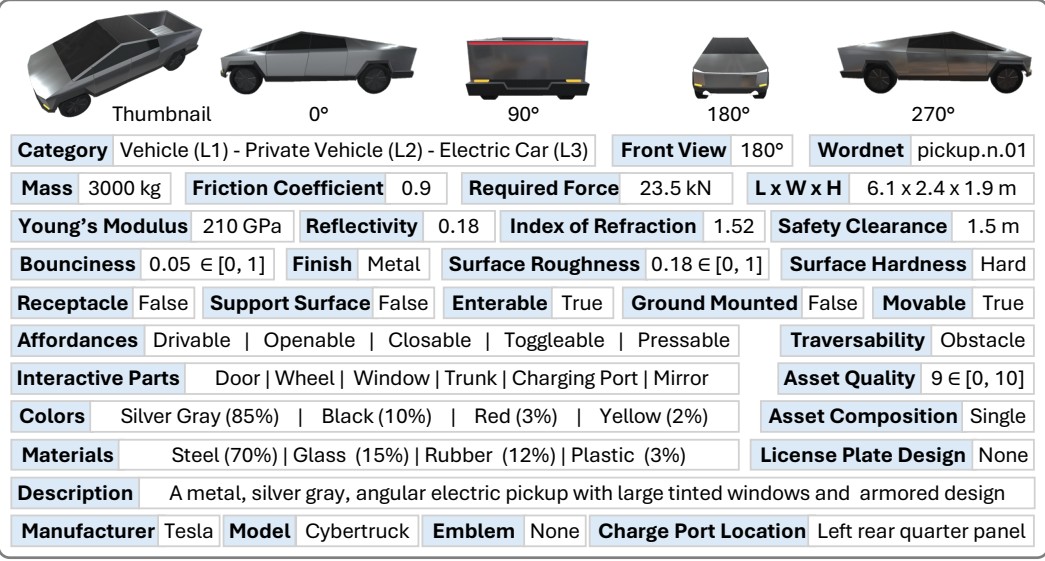

| Category | Vehicle (L1) - Private Vehicle (L2) - Electric Car (L3) | Front View | 180° | Wordnet | pickup.n.01 |

Figure 24: **Example of full annotated attributes for each object asset in UrbanVerse-100K**

2. **No texture:** pure meshes without surface textures, lacking visual realism;

3. **Paper-like:** thin, hand-authored background props with negligible mesh depth, unsuitable for physical simulation;

4. **With base:** assets embedded in oversized base meshes, producing inaccurate occupancy and collisions;

5. **Terrain maps:** large terrain-like assets that cannot be meaningfully used in urban embodied AI simulation;

6. **Inconsistent names:** category names directly inherited from web tags, often written in multiple languages, idiosyncratic codes, or designer-specific terms;

7. **CAD-like:** CAD models lacking textures and physical realism, unsuitable for direct use in interactive simulation;

8. **Non-single objects:** assets that are entire scenes or contain multiple unrelated objects rather than a single entity;

9. **Non-uniform scales:** assets not in metric scale, which makes them unusable for physically grounded simulation.

The non-uniform scale issue is particularly problematic: as shown in Fig. 26, direct import of such assets can lead to absurd scenarios (e.g., a fire hydrant larger than a building). Similar issues have also been noted in Objaverse++ (Lin et al., 2025a). To address these problems at their root, we curated a new repository, UrbanVerse-100K, by employing human annotators to carefully filter low-quality assets. This manual quality control ensures that only simulation-ready objects are included. Our curated database will be open-sourced to facilitate reliable and reproducible embodied AI research.

**Asset Scale Issues.** Due to these quality issues, existing simulators—such as MetaUrban (Wu et al., 2025a), UrbanSim (Wu et al., 2025b), indoor simulators (Deitke et al., 2022; Gan et al., 2020; Deitke et al., 2020; Szot et al., 2021; Kolve et al., 2017; Li et al., 2023a), and driving simulators (Dosovitskiy et al., 2017; Martinez et al., 2017; Kothari et al., 2021; Caesar et al., 2021)—typically rely on relatively small, manually curated asset repositories. Human annotators must painstakingly adjust object scales and orientations one by one, which is not scalable. As summarized in Tab. 5, current urban simulators rarely exceed 15K curated assets, limiting the diversity and richness achievable in simulation environments.

In contrast, our work introduces a hybrid annotation pipeline that combines efficient human filtering with large language model (LLM)-based automatic annotation. Human annotators perform rapid

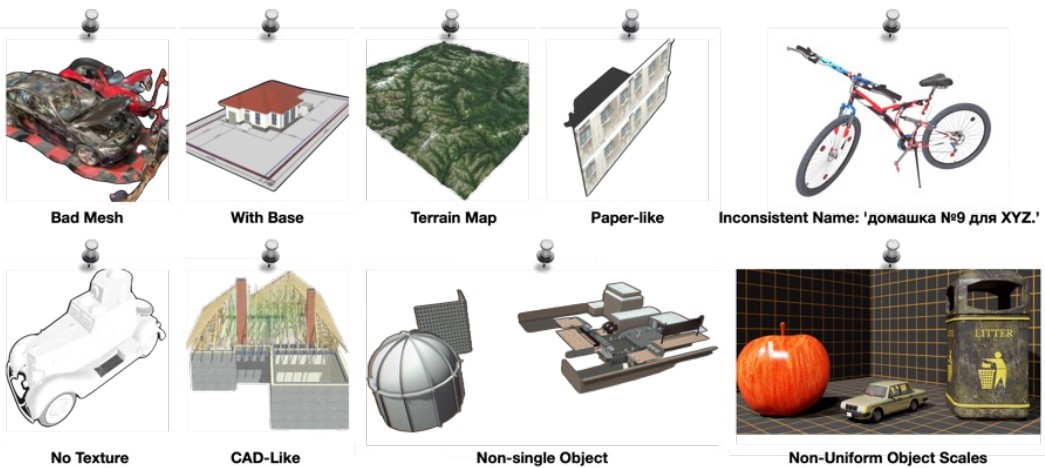

Figure 25: **Typical quality issues in existing 3D asset databases.**

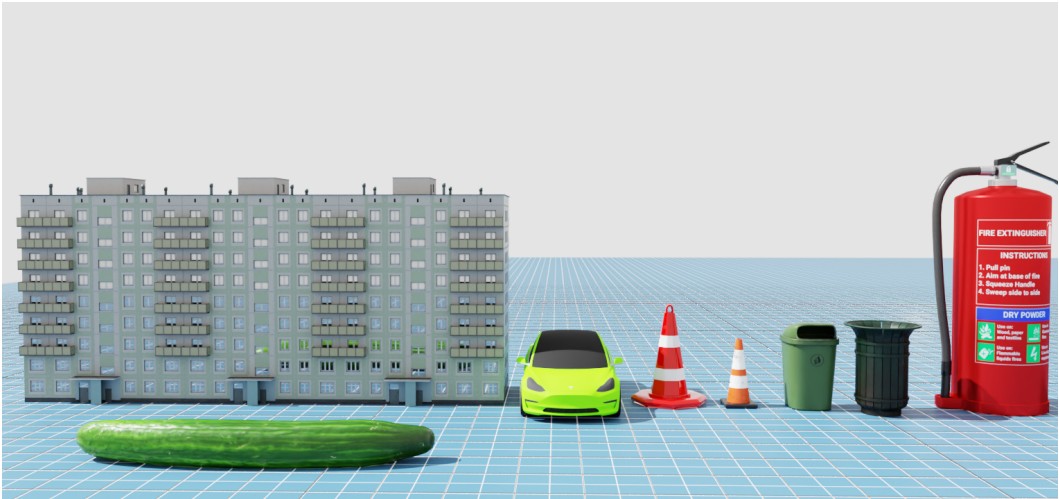

Figure 26: **Non-metric scale problems in existing 3D asset databases.**

binary tagging to eliminate poor-quality assets, while LLMs contribute common-sense knowledge to automatically annotate metric scales, canonical front views, semantic categories, and physical attributes. This hybrid process enables us to construct a significantly larger, physically grounded urban asset database at scale, bridging the gap between raw Internet collections and high-quality simulation-ready repositories. In the next section, we detail our hybrid annotation pipeline.

## C.4 VALIDATION OF PHYSICAL ATTRIBUTE ANNOTATIONS

Validating the physical attributes annotation in UrbanVerse-100K is essential, yet direct human annotation of true object dimensions or mass is generally infeasible without expert knowledge. Prior simulators (*e.g.*, MetaUrban, UrbanSim) typically rely on *anchor-based visual calibration*, where objects are manually resized based on appearance and relative proportions. Following this practice, our main paper presents large-scale qualitative validation by placing hundreds of assets side-by-side to demonstrate consistent, physically plausible relative scales in Fig. 2.

To complement these qualitative checks, in this section, we conduct a quantitative evaluation on 17 categories comprising 1,335 objects for which reliable specifications or commonly agreed real-world dimensions are publicly available (*e.g.*, Tesla Cybertruck, vending machines, traffic cones, laptops).

| Scenario | Simulator | Scene Creation | Physics Engine | Parallel Training | # of Scenes | # of Categories | # of Assets | Asset Physics | # of Skyboxes | # of Ground Materials |
|---|---|---|---|---|---|---|---|---|---|---|
| Indoor | AI2-THOR | Manual | Unity | ✗ | 120 | – | 3,578 | ✓ | ✗ | ✗ |
| | ProcTHOR | PG | Unity | ✗ | +∞ | 108 | 1,633 | ✓ | ✗ | ✗ |
| | Habitat 3.0 | Manual | Bullet | ✗ | 211 | – | 18,656 | ✗ | ✗ | ✗ |
| | Holodeck | Manual | Unity | ✗ | +∞ | 108 | 1,633 | ✗ | ✗ | ✗ |
| | Behavior | Manual | PhysX | ✓ | 50 | 1,900 | 9,000 | ✓ | ✗ | ✗ |
| Driving | GAT-V | Manual | Unity | ✗ | – | – | ✗ | ✗ | ✗ | ✗ |
| | CARLA | Manual | Unreal5 | ✗ | 15 | 106 | 935 | ✗ | 1 | 10 |
| | MetaDrive | PG | Panda3D | ✗ | +∞ | 5 | 5 | ✗ | 1 | 3 |
| Urban | MetaUrban | PG | Panda3D | ✗ | +∞ | 39 | 10,000 | ✗ | 1 | 5 |
| | Urban-Sim | PG | IsaacSim | ✓ | +∞ | 39 | 15,000 | ✗ | 1 | 8 |
| | **UrbanVerse** | **Real2sim** | **IsaacSim** | **✓** | **+∞** | **659** | **102,444** | **✓** | **306** | **288** |

Table 5: **A systematic comparison of urban Embodied AI simulators.**

| Category | # Objects | MAPE H (%) | MAPE L (%) | MAPE W (%) | MAPE M (%) |
|---|---|---|---|---|---|
| **Average** | – | **5.88 ± 3.85** | **6.14 ± 3.86** | **7.00 ± 4.11** | **19.58 ± 12.11** |
| Lamborghini Huracan STO | 12 | 0.49 ± 0.30 | 0.66 ± 0.40 | 0.51 ± 0.30 | 2.02 ± 1.30 |
| McLaren 600LT Spider | 4 | 0.59 ± 0.40 | 0.78 ± 0.50 | 0.62 ± 0.40 | 2.47 ± 1.50 |
| Tesla Cybertruck | 7 | 1.97 ± 1.40 | 2.50 ± 1.70 | 2.02 ± 1.40 | 9.05 ± 6.50 |
| Land Rover Defender | 2 | 2.99 ± 2.00 | 3.72 ± 2.80 | 1.99 ± 1.40 | 12.56 ± 7.50 |
| Electric Scooter | 68 | 5.97 ± 4.00 | 9.38 ± 7.00 | 12.32 ± 6.00 | 29.96 ± 19.00 |
| Bicycle | 118 | 7.03 ± 5.00 | 5.05 ± 3.00 | 7.75 ± 5.00 | 17.78 ± 13.00 |
| Vending Machine | 153 | 7.94 ± 5.00 | 9.00 ± 5.00 | 7.18 ± 5.00 | 19.60 ± 10.00 |
| Street Cabinet | 43 | 20.00 ± 13.00 | 23.00 ± 13.00 | 22.00 ± 12.00 | 14.86 ± 10.00 |
| Parking Meter | 16 | 7.94 ± 5.00 | 10.71 ± 7.00 | 6.56 ± 5.00 | 22.94 ± 14.00 |
| Fire Hydrant | 160 | 1.47 ± 1.00 | 1.53 ± 1.00 | 1.53 ± 1.00 | 7.14 ± 4.00 |
| Traffic Cone | 126 | 12.57 ± 9.00 | 20.40 ± 13.00 | 20.40 ± 13.00 | 51.07 ± 25.00 |
| Jersey Barrier | 95 | 5.06 ± 3.00 | 7.07 ± 5.00 | 25.83 ± 13.00 | 75.00 ± 50.00 |
| Egg | 188 | 1.09 ± 0.70 | 1.03 ± 0.70 | 1.03 ± 0.70 | 12.45 ± 8.00 |
| Cigarette | 28 | 1.00 ± 0.60 | 1.00 ± 0.60 | 1.00 ± 0.60 | 15.10 ± 9.00 |
| Laptop | 171 | 20.46 ± 13.00 | 5.06 ± 3.00 | 4.84 ± 3.00 | 25.02 ± 17.00 |
| Football | 99 | 1.50 ± 1.00 | 1.50 ± 1.00 | 1.50 ± 1.00 | 6.83 ± 4.00 |
| Basketball | 45 | 1.92 ± 1.00 | 1.92 ± 1.00 | 1.92 ± 1.00 | 8.97 ± 6.00 |

Table 6: **Evaluation of annotated physical attributes.** We report the MAPE and standard deviation against ground-truth attribute values across 17 object categories for Height (H), Length (L), Width (W), and Mass (M).

For each category, we compute the Mean Absolute Percentage Error (MAPE),

$$\text{MAPE} = \frac{100\%}{N} \sum_{i=1}^{N} \left| \frac{\text{Annotation}_i - \text{GT}_i}{\text{GT}_i} \right|,$$

over height, length, width, and mass.

As summarized in Tab. 6, geometric attributes are highly accurate—typically within 1–8% MAPE and often 1–3% for rigid objects such as cars, hydrants, and balls. Mass values exhibit larger variation due to material uncertainty but remain within a reasonable error range (mean 19.58%). Overall, these results demonstrate that our automatic annotation pipeline yields reliable physical attributes at scale, enabling high-fidelity, physics-aware simulation without manual labeling.

# D  DETAILS OF URBANVERSE SCENES

## D.1  DETAILS OF URBANVERSE SCENE LIBRARY CONSTRUCTION

**City-tour video collection.** To construct a scene library that captures diverse and realistic layouts reflective of real-world urban settings, we collect 32 city-tour videos from YouTube released under Creative Commons licenses. These videos span 7 continents, 24 countries, and 27 cities, providing geographically and culturally comprehensive coverage for our city-tour video inputs. We present a few examples of the collected city-tour videos in Fig. 27. The distribution is as follows:

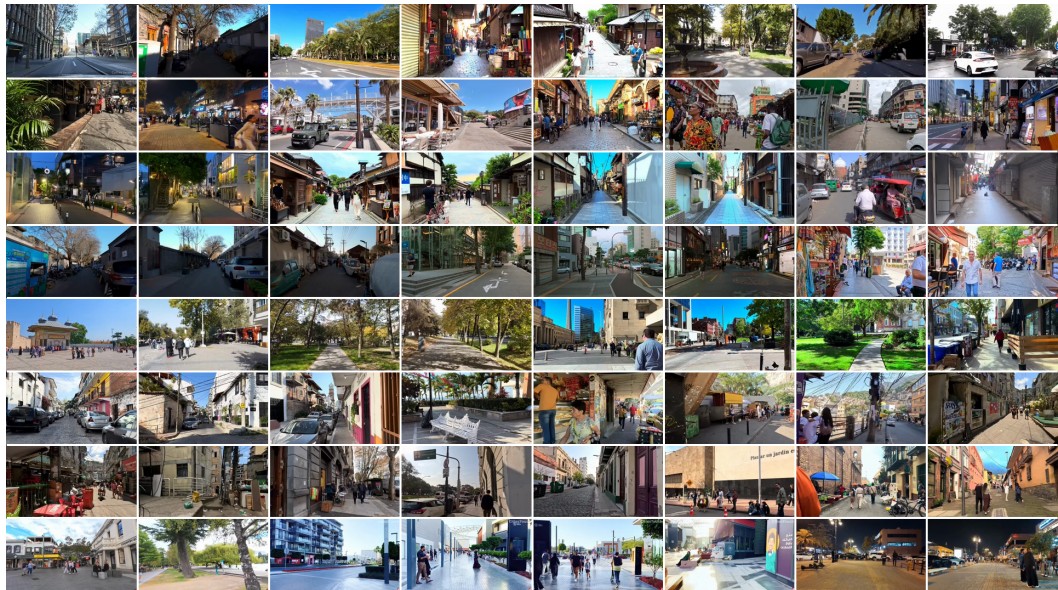

Figure 27: **A thumbnail montage showcasing a subset of the city-tour videos collected worldwide for grounding scene generation.**

- **Continents:** Africa, Asia, Europe, Middle East, North America, Oceania, South America
- **Countries:** Egypt, Kenya, Morocco, Nigeria, South Africa, China, India, Japan, Kazakhstan, Singapore, South Korea, Vietnam, France, Iceland, Italy, Netherlands, Spain, Sweden, Saudi Arabia, United Arab Emirates, Canada, Mexico, United States, Australia, New Zealand, Argentina, Brazil, Colombia
- **Cities:** Cairo, Nairobi, Tangier, Rabat, Lagos, Cape Town, Beijing, Shijiazhuang, New Delhi, Tokyo, Kyoto, Almaty, Singapore, Seoul, Ho Chi Minh City, Paris, Reykjavik, Naples, Amsterdam, Barcelona, Stockholm, Riyadh, Dubai, Toronto, Puerto Vallarta, Los Angeles, Sydney, Auckland, Buenos Aires, São Paulo, Rio de Janeiro, Bogotá

**UrbanVerse simulation scene library.** By applying our UrbanVerse-Gen pipeline to the collected city-tour videos, we generate a library of 160 urban simulation scenes for embodied AI training. Here, we present a few examples from the generated scene library in Fig. 28.

**UrbanVerse scene diversity.** The diversity of UrbanVerse scenes emerges from the wide range of retrieved object assets, ground materials, and sky maps integrated by our automatic real-to-sim UrbanVerse-Gen pipeline. From the collected city-tour videos, UrbanVerse-Gen generates multiple *digital cousin* scenes that share the same underlying layout but differ in visual and physical composition, while preserving the key geometry and affordances. Fig. 29 showcases variations in lighting conditions, distant backgrounds, and ground appearances within the simulator. By retrieving multiple digital cousin assets from UrbanVerse-100K for each object instance, UrbanVerse-Gen can produce numerous diverse urban scenes from a single city-tour video, varying in illumination, ground materials, object geometry, and texture. We present the top-5 digital cousin scenes generated from city-tour videos filmed in Beijing, China, and Tangier, Morocco, in Fig. 30 and Fig. 31, respectively.

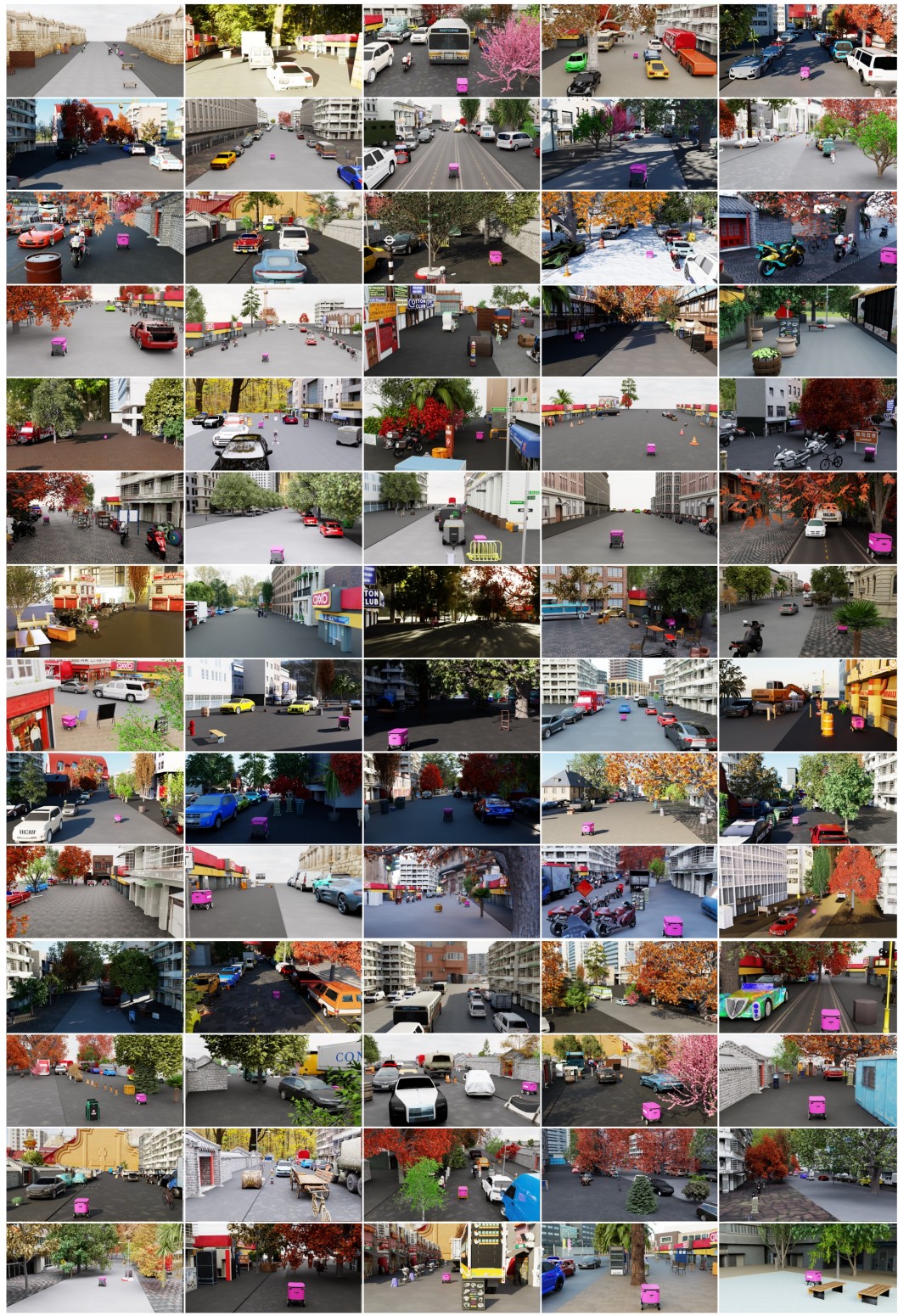

Figure 28: **A thumbnail montage showcasing a subset of the generated UrbanVerse simulation scenes.**

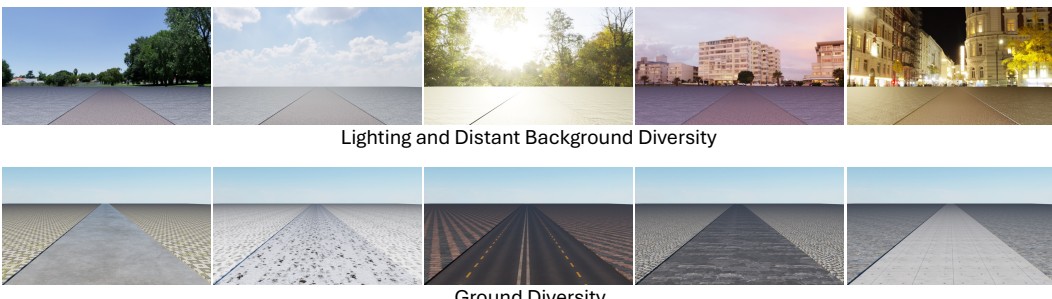

Figure 29: **Examples illustrating the diversity of lighting conditions, distant backgrounds, and ground materials in UrbanVerse scenes.**

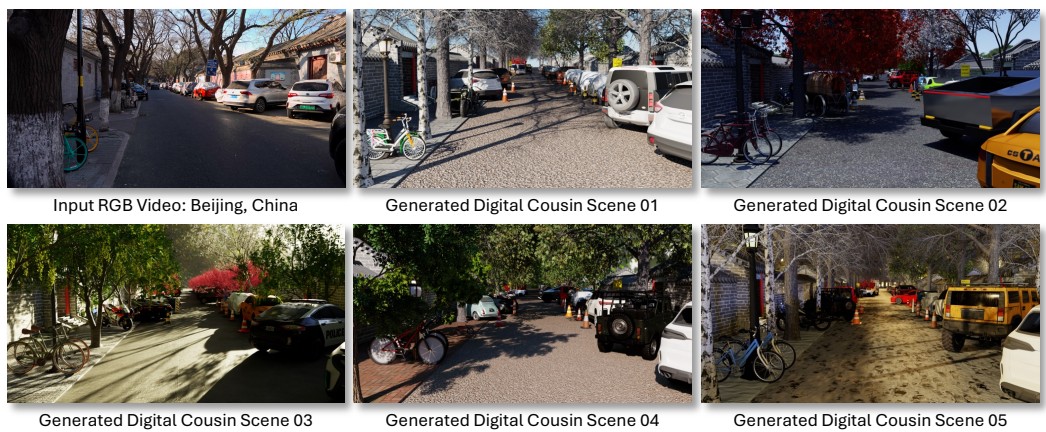

Figure 30: **Qualitative results of the top-5 digital cousin scenes** generated by UrbanVerse-Gen from a walking city-tour video of Beijing, China.

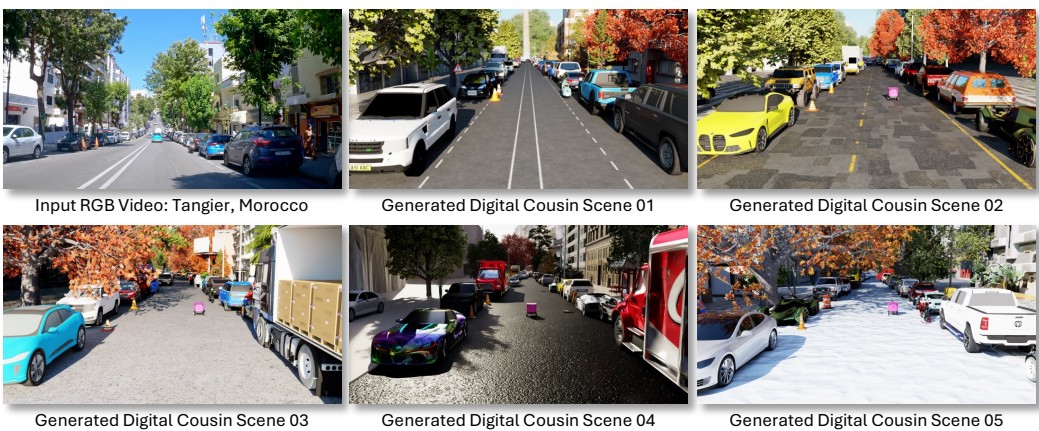

Figure 31: **Qualitative results of the top-5 digital cousin scenes** generated by UrbanVerse-Gen from a driving city-tour video of Tangier, Morocco.

## D.2  DETAILS OF CRAFTBENCH SCENE CREATION

To complement automatically generated scenes, we commission professional 3D artists to design a suite of high-fidelity environments that serve as the CraftBench benchmark for closed-loop evaluation. These scenes are carefully crafted to balance realism and diversity, capturing not only the everyday orderliness of urban streets but also the messy, safety-critical edge cases that real-world agents

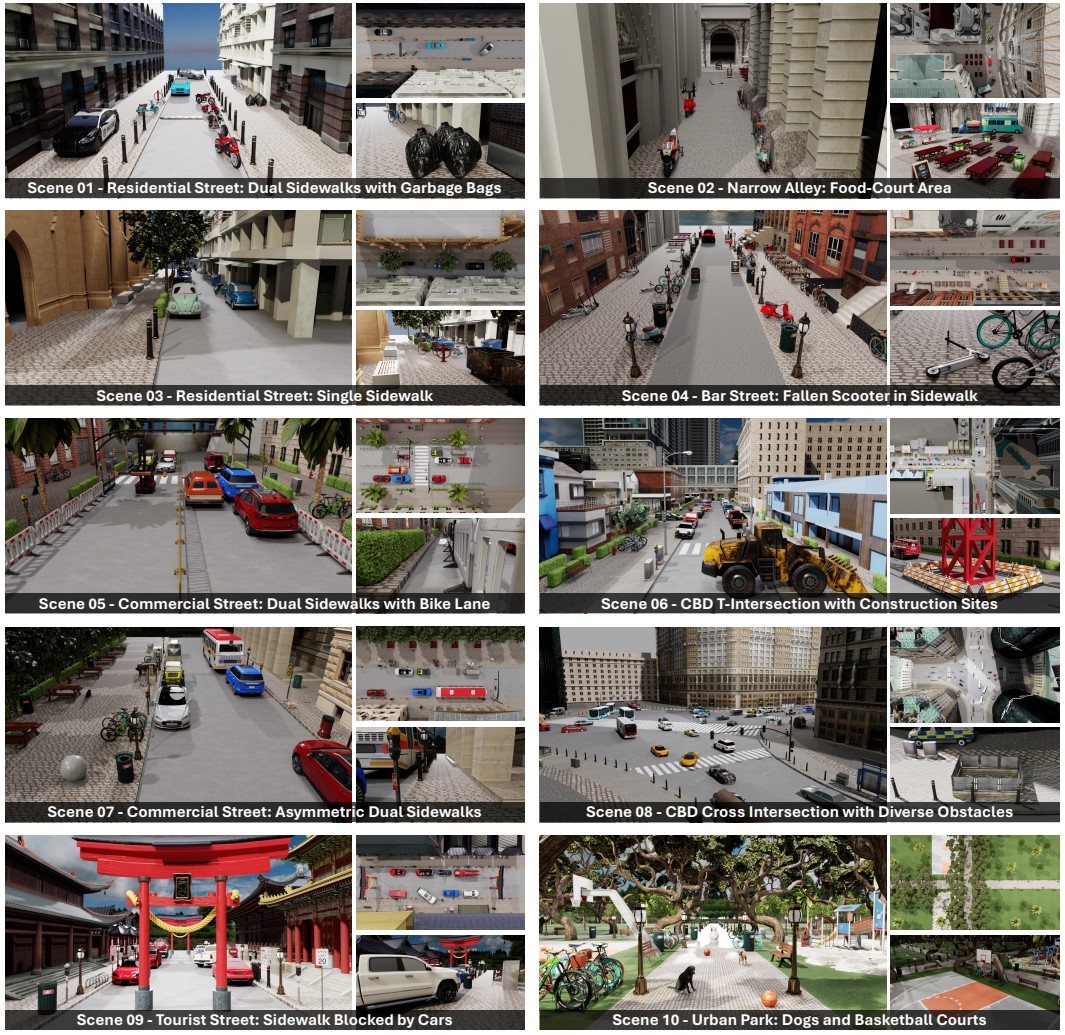

Figure 32: **Examples of all CraftBench test scenes.**

frequently encounter. As shown in Fig. 32, the benchmark includes diverse urban layouts such as residential streets with garbage bags on sidewalks, narrow alleys lined with food courts, bar streets with fallen scooters obstructing walkways, commercial districts with bike lanes, and CBD intersections under active construction. The scenes also depict edge cases such as asymmetric sidewalks, sidewalks blocked by illegally parked cars, and public parks populated with both dogs and basketball courts. By combining realistic details with safety-critical anomalies, these artist-created scenes provide challenging yet authentic environments for evaluating embodied urban navigation.

# E IMPLEMENTATION DETAILS OF URBANVERSE-GEN PIPELINE

In this section, we provide implementation details of the proposed UrbanVerse-Gen pipeline. For structure-from-motion, we adopt MASt3R (Leroy et al., 2024) with a ViT-Large backbone to estimate camera intrinsics, metric depth, and camera poses. For 2D object semantic parsing, we use the state-of-the-art open-vocabulary detector YOLO-Worldv2 XL (Cheng et al., 2024) to obtain on-ground object bounding boxes. Subsequently, SAM 2.1 Large (Ravi et al., 2024) refines these detections by generating pixel-level semantic instance masks conditioned on the YOLO-Worldv2 predictions. To match objects with database assets, we employ CLIP ViT-L/14 (Radford et al., 2021) for textual semantic similarity, and DINOv2 ViT-B/32 (Oquab et al., 2023) to measure visual similarity between input object masks and the thumbnails of 3D assets in our database.

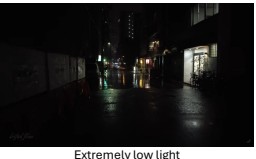 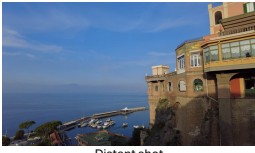 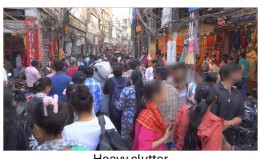 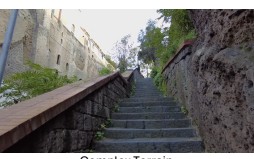

Extremely low light      Distant shot      Heavy clutter      Complex Terrain

Figure 33: **Typical challenging input video conditions.**

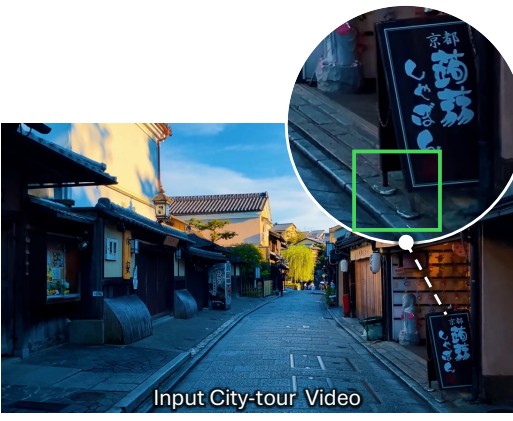 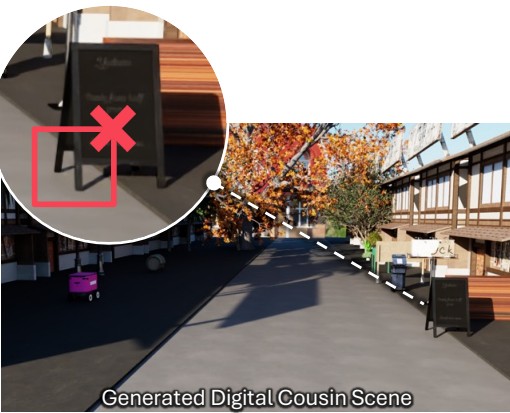

Input City-tour Video          Generated Digital Cousin Scene

Figure 34: **Example failure cases from UrbanVerse-Gen.** The sidewalk billboard that is originally placed on the sidewalk in the input video is shifted toward the road due to inaccurate depth and pose drift

## F  FAILURE CASE ANALYSIS

**Challenging Input Conditions.** UrbanVerse-Gen is generally robust across diverse city-tour videos, but certain input conditions can still degrade reconstruction quality. As the examples shown in Fig. 33, extremely low light, distant shot, heavy clutter, or complex terrain may affect depth estimation and geometric consistency. To prevent clearly unrecoverable clips from entering the pipeline, we screen every 10th frame with GPT–4.1 during large-scale generation and automatically filter out problematic segments.

**Depth and Pose Drift.** Fast camera motion or unstable handheld recordings may introduce depth or pose drift, occasionally causing misplaced objects (*e.g.*, a sidewalk billboard shifted toward the roadway). Multi-view aggregation substantially reduces such errors, though they remain the most common failure mode. As illustrated in Fig. 34, a sidewalk billboard shifted toward the road due to inaccurate depth and pose drift. on the sideIncorporating spatial constraints between objects and their plausible placement areas is a promising direction for further improving stability.

**Imperfect Asset Retrieval.** Visually complex or rare objects may not always retrieve a perfect appearance match from UrbanVerse-100K. Our three-stage matching strategy—semantic matching, geometry filtering, and appearance selection—ensures that the retrieved asset maintains correct category, affordance, and collision geometry. As a result, appearance mismatches do not harm physical interaction and often serve as benign domain randomization.

**Mis-segmentation Under Heavy Occlusion.** Severe occlusion by pedestrians or vehicles can lead to incomplete masks from open-vocabulary detectors. However, because our layout extraction fuses multi-view evidence rather than relying on single-frame segmentation, the reconstructed scene geometry typically remains stable.

**Impact on Downstream Learning.** Despite these failure modes, we observe minimal negative impact on downstream policy learning. Multi-view fusion and strict geometry filtering provide stable scene layouts even when some frames are noisy, and both simulator evaluations and real-world sim-to-real results confirm that policies trained in UrbanVerse generalize reliably. This indicates that UrbanVerse provides sufficient scene fidelity for large-scale robot training.

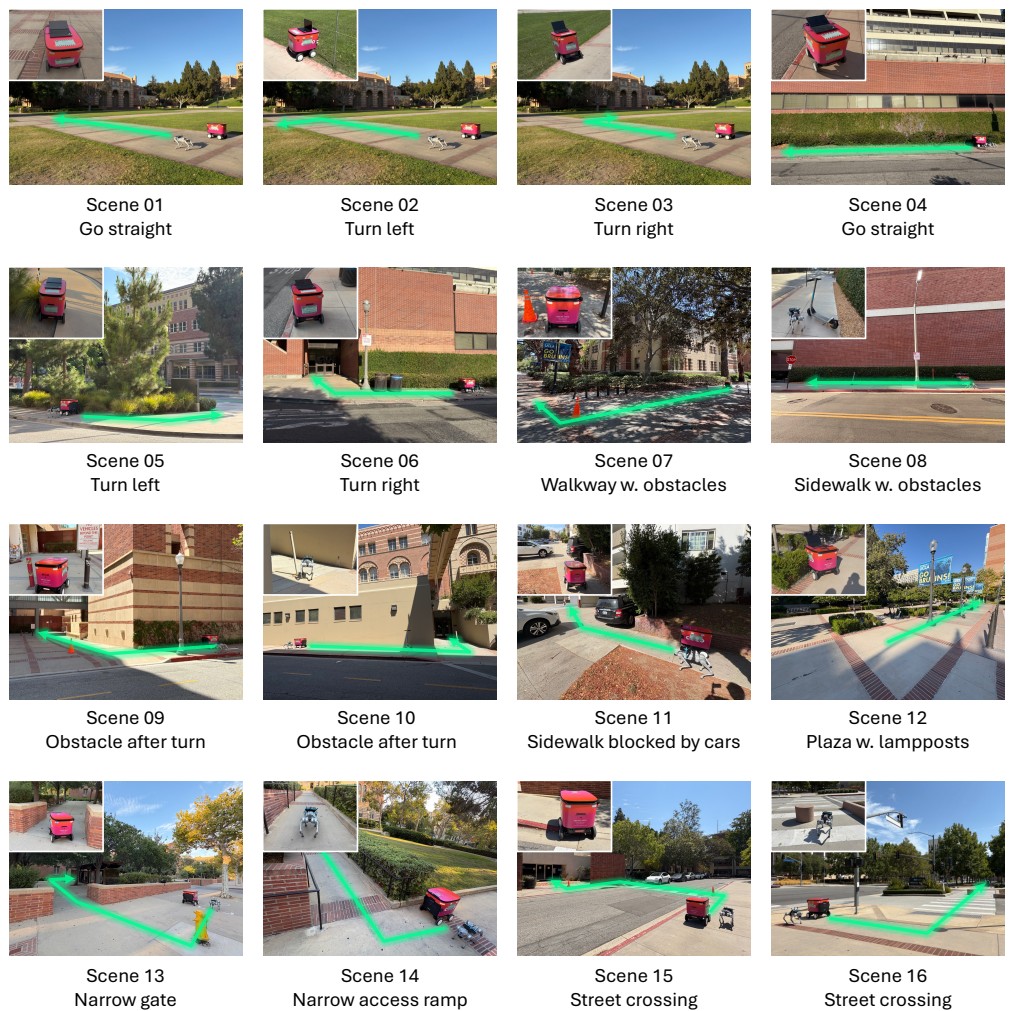

Figure 35: **Real-world testing scenarios.** We evaluate zero-shot sim-to-real policy transfer across 16 diverse urban scenes and two embodiments (Coco wheeled delivery robot and Unitree Go2 quadruped), spanning challenges such as straight paths (Scenes 01, 04) and turns (Scenes 02, 03, 05, 06) on open ground or sidewalks, walkway and sidewalk obstacles (Scenes 07, 08), obstacles appearing after turns (Scenes 09, 10), sidewalk blockage by cars (Scene 11), structural elements including lampposts and narrow access points (Scenes 12–14), and street crossings (Scenes 15, 16). Key challenges for each scene are highlighted in Red at the upper-right corner.

## G  REAL-WORLD TESTING SCENE SELECTION

To rigorously evaluate zero-shot sim-to-real transfer, we carefully select 16 real-world testing scenes that expose diverse challenges for two tested robot embodiments: the Coco wheeled delivery robot and the Unitree Go2 quadruped, as shown in Fig. 35. These scenarios assess each robot's ability to follow trajectories on open ground and sidewalks, negotiate turns both with and without obstacles, and cope with safety-critical events such as obstacles appearing after sharp turns, sidewalks blocked by parked cars, and walkways cluttered with objects. Structural challenges such as narrow gates, access ramps, lampposts, and urban street crossings are also included to test mobility and accessibility. Concretely, the set spans straight paths (Scenes 01, 04), turns (Scenes 02, 03, 05, 06), walkways and sidewalks with obstacles (Scenes 07, 08), obstacles appearing after turns (Scenes 09, 10), sidewalk blockage by cars (Scene 11), lampposts and plazas (Scene 12), narrow gates and ramps (Scenes 13, 14), and street crossings (Scenes 15, 16). By covering both trajectory-following and obstacle-navigation tasks across two distinct embodiments, these scenes provide a comprehensive benchmark for evaluating policy robustness and generalization in realistic urban environments.

## H   ADDITIONAL EXPERIMENTS ON TRAINING SCENE HORIZON LENGTH

In this section, we further study how the spatial horizon of training scenes affects policy learning. Specifically, we compare *Half-Length UrbanVerse scenes*, where each scene is truncated to roughly half of its original spatial extent (∼100 m), with *Full-Length UrbanVerse scenes* that retain the complete layout (∼200 m). Both settings use the full set of 160 UrbanVerse training scenes (320 truncated scenes in the half-length case). Policies are trained on a wheeled robot and evaluated on the ten CraftBench test scenes.

As shown in Tab. 7, full-length scenes lead to clear improvements across all metrics, raising Success Rate by +9.6, reducing Collision Time by –5.0, and increasing Route Completion by +11.6. We attribute this to two factors. First, longer scenes expose the agent to richer spatial structures, denser object configurations, and longer-range dependencies that better match the complexity of real-world navigation and the artist-designed CraftBench layouts. Second, longer episodes provide PPO with more diverse transitions and more challenging decision points per rollout, improving both collision avoidance and global route planning.

| Training Scene Length | SR ↑ | CT ↓ | RC ↑ |
|---|---|---|---|
| Half Length (∼100m) | 32.3 | 40.5 | 50.8 |
| Full Length (∼200m) | **41.9** | **35.5** | **62.4** |

Table 7: **Effect of training scene horizon length.** Policies are trained on all 160 UrbanVerse scenes; the half-length setting truncates each scene to ∼100 m, while the full-length setting uses the complete ∼200 m layouts. Evaluation is conducted on the ten CraftBench test scenes using a wheeled robot.

These results highlight that increasing scene horizon—and thereby spatial complexity—is an effective way to improve generalization in UrbanVerse-trained policies.

## I   ADDITIONAL RESULTS ON CUSTOMIZABLE REAL-TO-SIM-TO-REAL POLICY LEARNING AND TRANSFER

In certain practical deployments, the robot's target operational environment is known beforehand. We therefore investigate whether sim-to-real policy transfer can be further improved through **customized policy fine-tuning** within *digital cousin* simulation scenes of the target environments, generated by our UrbanVerse-Gen real-to-sim pipeline—forming *a fully automated, end-to-end real-to-sim-to-real policy learning framework*. To this end, we focus on the `Scene 12` real-world test scenario (see the site image in Fig. 35) as the target operational environment. This scene is particularly challenging, as the agent must navigate through a dense arrangement of lampposts to reach the goal while avoiding randomly parked e-scooters, bollards, and shrubs along the sides of the path. In this environment, our original PPO-UrbanVerse policy on the Coco wheeled robot failed entirely, achieving a success rate of 0%, as reported in Tab. 9. To construct the corresponding digital cousin scenes for fine-tuning, we recorded a short handheld RGB video of the target environment and used UrbanVerse-Gen to generate eight customized simulation scenes tailored to it. We then *fine-tuned* the PPO-UrbanVerse model (pre-trained on 160 UrbanVerse scenes) on these *customized* scenes (denoted as *UrbanVerse + Target*) and compared it against two baselines: the original pretrained policy without fine-tuning (denoted as *UrbanVerse*) and a policy trained solely on the target scenes (denoted as *Target Only*).

As shown in Tab. 8, only the policy pretrained on the 160-scene UrbanVerse library and subsequently fine-tuned on the 8 customized digital cousin scenes succeeded in completing the task, effectively navigating the challenging sequence of lampposts and e-scooters—demonstrating rapid *real-to-sim-to-real* transfer. Interestingly, although both the pretrained and target-only policies failed to reach the goal, PPO-UrbanVerse still outperformed the latter in route completion (RC) and distance-to-goal (DTG). This suggests that the

| Training Scenes | SR ↑ | CT ↓ | RC ↑ | DTG ↓ |
|---|---|---|---|---|
| UrbanVerse | 0.0 | 100.0 | 72.9 | 5.3 |
| Target Only | 0.0 | 100.0 | 23.6 | 20.0 |
| **UrbanVerse + Target** | **80.0** | **20.0** | **96.5** | **0.3** |

Table 8: **Results of real-to-sim-to-real customized policy learning and transfer.** Performance comparison across different training setups in the `Scene 12` real-world test scenario. All results are evaluated on the 🚌 wheeled robot.

broader and more diverse training distribution provided by UrbanVerse acts as an implicit regularizer, preventing overfitting to a small number of customized scenes and mitigating the sim-to-real gap.

| | Scene 01 | | | | Scene 02 | | | | Scene 03 | | | | Scene 04 | | | |
|---|---|---|---|---|---|---|---|---|---|---|---|---|---|---|---|---|
| **Route Length** | 22.1 m | | | | 25.8 m | | | | 22.9 m | | | | 23.1 m | | | |
| **Method** | SR↑ | CT↓ | RC↑ | DTG↓ | SR↑ | CT↓ | RC↑ | DTG↓ | SR↑ | CT↓ | RC↑ | DTG↓ | SR↑ | CT↓ | RC↑ | DTG↓ |
| NoMad | 100.0 | 0.0 | 93.3 | 1.4 | 100.0 | 0.0 | 99.6 | 0.3 | 100.0 | 0.0 | 100.0 | 1.1 | 100.0 | 0.0 | 95.1 | 1.3 |
| CityWalker | 100.0 | 0.0 | 93.3 | 1.5 | 100.0 | 0.0 | 97.5 | 3.8 | 100.0 | 0.0 | 96.3 | 0.8 | 66.7 | 33.3 | 74.3 | 5.5 |
| S2E | 100.0 | 0.0 | 90.2 | 2.0 | 100.0 | 0.0 | 95.9 | 2.1 | 66.7 | 66.7 | 89.4 | 2.9 | 100.0 | 0.0 | 90.5 | 2.0 |
| PPO-UrbanSim | 100.0 | 0.0 | 90.9 | 2.0 | 100.0 | 0.0 | 65.0 | 6.0 | 100.0 | 0.0 | 87.1 | 4.3 | 0.0 | 100.0 | 67.3 | 6.0 |
| PPO-UrbanVerse | **100.0** | **0.0** | 90.9 | 2.0 | **100.0** | **0.0** | 95.1 | 2.1 | **100.0** | **0.0** | 93.1 | 2.0 | **100.0** | **0.0** | 90.7 | 2.0 |

| | Scene 05 | | | | Scene 06 | | | | Scene 07 | | | | Scene 08 | | | |
|---|---|---|---|---|---|---|---|---|---|---|---|---|---|---|---|---|
| **Route Length** | 13.9 m | | | | 13.7 m | | | | 24.8 m | | | | 26.2 m | | | |
| **Method** | SR↑ | CT↓ | RC↑ | DTG↓ | SR↑ | CT↓ | RC↑ | DTG↓ | SR↑ | CT↓ | RC↑ | DTG↓ | SR↑ | CT↓ | RC↑ | DTG↓ |
| NoMad | 0.0 | 100.0 | 53.2 | 5.4 | 0.0 | 100.0 | 69.3 | 4.9 | 0.0 | 100.0 | 30.0 | 15.0 | 33.3 | 66.7 | 63.6 | 10.0 |
| CityWalker | 0.0 | 100.0 | 38.3 | 6.5 | 0.0 | 100.0 | 31.1 | 6.8 | 0.0 | 100.0 | 36.5 | 14.8 | 0.0 | 100.0 | 14.8 | 22.1 |
| S2E | 0.0 | 100.0 | 9.5 | 9.0 | 100.0 | 0.0 | 82.7 | 2.0 | 66.7 | 33.3 | 69.8 | 7.0 | 66.7 | 33.3 | 66.5 | 8.0 |
| PPO-UrbanSim | 0.0 | 100.0 | 32.5 | 6.8 | 0.0 | 100.0 | 17.3 | 7.9 | 0.0 | 100.0 | 23.5 | 16.3 | 0.0 | 100.0 | 12.5 | 22.8 |
| PPO-UrbanVerse | **100.0** | **0.0** | **85.6** | **2.1** | **100.0** | **0.0** | 80.4 | 2.1 | **100.0** | **0.0** | **94.2** | **2.0** | **100.0** | **0.0** | **88.1** | **3.1** |

| | Scene 09 | | | | Scene 10 | | | | Scene 11 | | | | Scene 12 | | | |
|---|---|---|---|---|---|---|---|---|---|---|---|---|---|---|---|---|
| **Route Length** | 36.0 m | | | | 18.9 m | | | | 22.6 m | | | | 35.7 m | | | |
| **Method** | SR↑ | CT↓ | RC↑ | DTG↓ | SR↑ | CT↓ | RC↑ | DTG↓ | SR↑ | CT↓ | RC↑ | DTG↓ | SR↑ | CT↓ | RC↑ | DTG↓ |
| NoMad | 0.0 | 100.0 | 6.9 | 28.4 | 0.0 | 100.0 | 56.3 | 7.9 | 0.0 | 100.0 | 52.5 | 10.1 | 100.0 | 0.0 | 94.3 | 2.0 |
| CityWalker | 0.0 | 100.0 | 0.0 | 30.7 | 0.0 | 100.0 | 19.1 | 9.3 | 33.3 | 66.7 | 66.6 | 9.1 | 0.0 | 100.0 | 15.1 | 17.2 |
| S2E | 0.0 | 100.0 | 12.8 | 26.9 | 0.0 | 100.0 | 62.2 | 5.5 | 100.0 | 0.0 | 90.4 | 2.0 | 0.0 | 100.0 | 25.7 | 12.8 |
| PPO-UrbanSim | 0.0 | 100.0 | 20.9 | 11.8 | 0.0 | 100.0 | 8.4 | 10.6 | 0.0 | 100.0 | 82.9 | 3.5 | 0.0 | 100.0 | 10.0 | 15.0 |
| PPO-UrbanVerse | **33.3** | **66.7** | 70.2 | 9.1 | **33.3** | **66.7** | **79.4** | **4.0** | 33.3 | 66.7 | 52.4 | 9.7 | 0.0 | 100.0 | 72.9 | 5.3 |

| | Scene 13 | | | | Scene 14 | | | | Scene 15 | | | | Scene 16 | | | |
|---|---|---|---|---|---|---|---|---|---|---|---|---|---|---|---|---|
| **Route Length** | 28.0 m | | | | 21.8 m | | | | 25.9 m | | | | 31.7 m | | | |
| **Method** | SR↑ | CT↓ | RC↑ | DTG↓ | SR↑ | CT↓ | RC↑ | DTG↓ | SR↑ | CT↓ | RC↑ | DTG↓ | SR↑ | CT↓ | RC↑ | DTG↓ |
| NoMad | 0.0 | 100.0 | 9.2 | 18.1 | 0.0 | 100.0 | 10.0 | 17.9 | 0.0 | 100.0 | 68.1 | 7.7 | 0.0 | 100.0 | 17.4 | 24.9 |
| CityWalker | 0.0 | 100.0 | 10.0 | 20.5 | 0.0 | 100.0 | 6.9 | 17.9 | 0.0 | 100.0 | 74.4 | 4.4 | 0.0 | 100.0 | 8.3 | 24.9 |
| S2E | 0.0 | 100.0 | 11.2 | 18.2 | 33.3 | 66.7 | 35.1 | 12.4 | 0.0 | 100.0 | 72.6 | 5.4 | 33.3 | 66.7 | 49.0 | 13.5 |
| PPO-UrbanSim | 0.0 | 100.0 | 10.7 | 17.6 | 0.0 | 100.0 | 9.2 | 17.4 | 0.0 | 100.0 | 5.0 | 14.9 | 0.0 | 100.0 | 10.5 | 25.2 |
| PPO-UrbanVerse | **100.0** | **0.0** | **92.7** | **2.1** | **66.7** | **33.3** | **79.6** | **3.9** | **100.0** | **0.0** | **85.8** | **2.1** | **66.7** | **33.3** | **83.9** | **4.6** |

Table 9: **Expanded real-world results of Coco wheeled robot on each scene. Best performance is colored in Blue .**

## J  ADDITIONAL ZERO-SHOT SIM-TO-REAL TRANSFER RESULTS

We provide per-scene real-world experimental results for both robot embodiments. Specifically, Tab. 9 reports results for the Coco wheeled delivery robot, while Tab. 10 presents results for the Unitree Go2 quadruped. All experiments are conducted three times for each method.

## K  SYSTEM COMPUTATIONAL ANALYSIS

We provide a detailed analysis of the computational cost and scalability of UrbanVerse, covering: (i) real-to-sim scene generation with UrbanVerse-Gen, (ii) large-scale object annotation in UrbanVerse-100K, and (iii) a comparison with the procedural workflow of UrbanSim. Results are summarized in TablesTab. 11, Tab. 12, Tab. 13, and Tab. 14.

**UrbanVerse-Gen Scene Generation.** As shown in Tab. 11, the computational cost of UrbanVerse-Gen scales almost linearly with input video length. The dominant runtime component is MASt3R-based 3D reconstruction, while GPT–4.1 queries are kept lightweight by sampling every third frame for object categorization. Short clips (10–40 s) require 4–14 multimodal LLM calls and 12–114 s of processing, and longer clips (80–180 s) require 27–60 calls and 289–1135 s on a single NVIDIA H100 GPU. Using 4 H100 GPUs, the system generates 160 fully interactive scenes from 180 s city-walk videos in 1.26 hours (Tab. 12), demonstrating strong practical scalability.

|  | Scene 01 | | | | Scene 02 | | | | Scene 03 | | | | Scene 04 | | | |
|---|---|---|---|---|---|---|---|---|---|---|---|---|---|---|---|---|
| Route Length | 22.1 m | | | | 25.8 m | | | | 22.9 m | | | | 23.1 m | | | |
| Method | SR↑ | CT↓ | RC↑ | DTG↓ | SR↑ | CT↓ | RC↑ | DTG↓ | SR↑ | CT↓ | RC↑ | DTG↓ | SR↑ | CT↓ | RC↑ | DTG↓ |
| NoMad | 100.0 | 0.0 | 94.5 | 5.0 | 100.0 | 0.0 | 97.5 | 0.8 | 100.0 | 0.0 | 97.1 | 0.9 | 100.0 | 0.0 | 80.1 | 4.7 |
| CityWalker | 100.0 | 0.0 | 94.5 | 5.0 | 100.0 | 0.0 | 96.1 | 1.0 | 100.0 | 0.0 | 98.0 | 0.8 | 100.0 | 0.0 | 90.0 | 5.3 |
| S2E | 100.0 | 0.0 | 88.9 | 2.0 | 100.0 | 0.0 | 93.8 | 2.1 | 100.0 | 0.0 | 92.1 | 2.0 | 100.0 | 0.0 | 67.1 | 6.1 |
| PPO-UrbanSim | 100.0 | 0.0 | 92.3 | 2.0 | 100.0 | 0.0 | 45.6 | 11.4 | 100.0 | 0.0 | 68.3 | 6.0 | 0.0 | 100.0 | 1.5 | 20.1 |
| PPO-UrbanVerse | 100.0 | 0.0 | 88.7 | 2.1 | 100.0 | 0.0 | 93.9 | 2.0 | 100.0 | 0.0 | 95.2 | 2.1 | 100.0 | 0.0 | 90.3 | 2.0 |

|  | Scene 05 | | | | Scene 06 | | | | Scene 07 | | | | Scene 08 | | | |
|---|---|---|---|---|---|---|---|---|---|---|---|---|---|---|---|---|
| Route Length | 13.9 m | | | | 13.7 m | | | | 24.8 m | | | | 26.2 m | | | |
| Method | SR↑ | CT↓ | RC↑ | DTG↓ | SR↑ | CT↓ | RC↑ | DTG↓ | SR↑ | CT↓ | RC↑ | DTG↓ | SR↑ | CT↓ | RC↑ | DTG↓ |
| NoMad | 100.0 | 0.0 | 90.7 | 6.8 | 100.0 | 0.0 | 83.1 | 2.8 | 0.0 | 100.0 | 22.4 | 18.0 | 0.0 | 100.0 | 44.2 | 13.1 |
| CityWalker | 100.0 | 0.0 | 93.0 | 2.4 | 0.0 | 100.0 | 30.8 | 6.0 | 0.0 | 100.0 | 9.1 | 20.7 | 0.0 | 100.0 | 2.7 | 22.8 |
| S2E | 0.0 | 100.0 | 78.5 | 3.3 | 100.0 | 0.0 | 79.0 | 2.1 | 66.7 | 33.3 | 94.6 | 2.1 | 100.0 | 0.0 | 87.0 | 3.1 |
| PPO-UrbanSim | 0.0 | 100.0 | 15.9 | 8.8 | 0.0 | 100.0 | 65.8 | 3.3 | 0.0 | 100.0 | 30.8 | 15.1 | 0.0 | 100.0 | 14.4 | 20.2 |
| PPO-UrbanVerse | 100.0 | 0.0 | 88.0 | 2.0 | 100.0 | 0.0 | 78.0 | 2.1 | 100.0 | 0.0 | 92.3 | 2.1 | 100.0 | 0.0 | 91.4 | 2.0 |

|  | Scene 09 | | | | Scene 10 | | | | Scene 11 | | | | Scene 12 | | | |
|---|---|---|---|---|---|---|---|---|---|---|---|---|---|---|---|---|
| Route Length | 36.0 m | | | | 18.9 m | | | | 22.6 m | | | | 35.7 m | | | |
| Method | SR↑ | CT↓ | RC↑ | DTG↓ | SR↑ | CT↓ | RC↑ | DTG↓ | SR↑ | CT↓ | RC↑ | DTG↓ | SR↑ | CT↓ | RC↑ | DTG↓ |
| NoMad | 0.0 | 100.0 | 74.4 | 10.1 | 0.0 | 100.0 | 83.8 | 13.4 | 0.0 | 100.0 | 36.8 | 13.2 | 0.0 | 100.0 | 18.4 | 18.2 |
| CityWalker | 0.0 | 100.0 | 37.1 | 19.1 | 0.0 | 100.0 | 5.2 | 20.3 | 0.0 | 100.0 | 23.1 | 14.9 | 0.0 | 100.0 | 25.8 | 16.6 |
| S2E | 33.3 | 66.7 | 57.4 | 13.6 | 100.0 | 0.0 | 96.2 | 2.1 | 66.7 | 33.3 | 63.3 | 2.7 | 0.0 | 100.0 | 53.0 | 10.5 |
| PPO-UrbanSim | 0.0 | 100.0 | 11.7 | 26.7 | 0.0 | 100.0 | 0.0 | 21.6 | 0.0 | 100.0 | 58.0 | 4.6 | 0.0 | 100.0 | 12.8 | 22.1 |
| PPO-UrbanVerse | 100.0 | 0.0 | 93.5 | 2.1 | 100.0 | 0.0 | 96.4 | 2.0 | 66.7 | 33.3 | 65.1 | 2.3 | 100.0 | 0.0 | 89.8 | 2.3 |

|  | Scene 13 | | | | Scene 14 | | | | Scene 15 | | | | Scene 16 | | | |
|---|---|---|---|---|---|---|---|---|---|---|---|---|---|---|---|---|
| Route Length | 28.0 m | | | | 21.8 m | | | | 25.9 m | | | | 31.7 m | | | |
| Method | SR↑ | CT↓ | RC↑ | DTG↓ | SR↑ | CT↓ | RC↑ | DTG↓ | SR↑ | CT↓ | RC↑ | DTG↓ | SR↑ | CT↓ | RC↑ | DTG↓ |
| NoMad | 0.0 | 100.0 | 1.5 | 20.8 | 0.0 | 100.0 | 33.6 | 13.9 | 0.0 | 100.0 | 0.0 | 26.4 | 0.0 | 100.0 | 12.2 | 32.4 |
| CityWalker | 0.0 | 100.0 | 1.5 | 20.8 | 0.0 | 100.0 | 20.4 | 16.6 | 0.0 | 100.0 | 48.5 | 10.4 | 0.0 | 100.0 | 5.8 | 32.3 |
| S2E | 0.0 | 100.0 | 69.3 | 7.9 | 33.3 | 66.7 | 33.9 | 13.8 | 0.0 | 100.0 | 53.6 | 9.9 | 33.3 | 66.7 | 34.3 | 17.6 |
| PPO-UrbanSim | 0.0 | 100.0 | 23.4 | 15.7 | 0.0 | 100.0 | 10.0 | 21.2 | 0.0 | 100.0 | 8.0 | 14.5 | 0.0 | 100.0 | 7.4 | 32.7 |
| PPO-UrbanVerse | 100.0 | 0.0 | 94.9 | 2.0 | 66.7 | 33.3 | 85.0 | 3.1 | 33.3 | 66.7 | 81.4 | 3.2 | 66.7 | 33.3 | 58.8 | 6.0 |

Table 10: **Expanded real-world results of GO2 quadruped robot on each scene.** Best performance is colored in Blue.

| Video Duration (sec) | Video Length (# frames) | LLM Calls | Scene Generation Wall Time (sec) |
|---|---|---|---|
| 10 | 10 | 4 | 12.38 |
| 40 | 40 | 14 | 114.46 |
| 80 | 80 | 27 | 289.80 |
| 180 | 180 | 60 | 1135.20 |

Table 11: **UrbanVerse-Gen real-to-sim scene generation time with varying input video lengths on an NVIDIA H100 GPU.**

| Setting | # City-tour Videos | # Cousin Scenes / Layout | # Unique Layouts | # Total Scenes | Wall Time |
|---|---|---|---|---|---|
| Single layout with 5 digital cousins | 1 | 5 | 1 | 5 | 18.92 min |
| 160 Scenes (1 × NVIDIA H100) | 32 | 5 | 32 | 160 | 10.08 hrs |
| 160 Scenes (4 × NVIDIA H100, default) | 32 | 5 | 32 | 160 | 1.26 hrs |

Table 12: **Overall computational time of UrbanVerse scene generation.**

| # Objects | GPT-4.1 Call Counts | API Wall Clock Time | API Cost |
|---|---|---|---|
| 1 | 1 | 0.0003 hrs | $0.018 |
| 2 | 2 | 0.0005 hrs | $0.029 |
| ... | ... | ... | ... |
| 102,444 | 102,444 | 65.5 hrs | $1,334 |
| **Average** | 1 / object | 2.3 sec / object | $0.013 / object |

Table 13: **Annotation cost and runtime statistics of the UrbanVerse-100K annotation pipeline.**

| Aspect | UrbanSim | UrbanVerse |
|---|---|---|
| How to Add a New 3D Asset | Manual annotation (metric resizing, mass setup) | Fully automated annotation pipeline |
| Time to Add a New 3D Asset | ∼600 sec | ∼2.3 sec |
| How to Add a New Scene | Manual creation of scene templates | Automatic scene generation via UrbanVerse-Gen |
| Time to Create a New Scene (∼200m) | ∼240 min | ∼18.9 min (NVIDIA H100) |
| Rendering Efficiency (RGB, Single Env, L40S) | ∼94 FPS | ∼94 FPS |

Table 14: **Comparison of UrbanSim and UrbanVerse for asset creation, scene generation, and rendering efficiency.**

**UrbanVerse-100K Asset Annotation.** The UrbanVerse-100K annotation pipeline also scales efficiently. As summarized in Tab. 13, each 3D asset is annotated with exactly one GPT–4.1 call, averaging 2.3 s and $0.013 per object. Annotating the full dataset of 102,444 assets requires 65.5 hours of API wall time and $1,334 in total cost, providing an economical path to large-scale semantic and physical labeling without manual intervention.

**Comparison with UrbanSim.** UrbanSim relies on procedural templates that are hand-designed by developers and require manual asset annotation. While procedural sampling itself incurs little computational cost, this workflow limits realism and diversity. As shown in Tab. 14, UrbanVerse automates both scene creation and asset annotation, reducing scene generation time from ∼240 min (UrbanSim) to 18.9 min, and reducing per-object annotation time from ∼600 s to 2.3 s. Both systems achieve similar rendering throughput in Isaac Sim (94 FPS), but UrbanVerse provides orders-of-magnitude higher scalability and produces scenes grounded in real-world distributions.

## L  EXPERIMENT SETUP DETAILS

### L.1  DEFINITION OF URBAN NAVIGATION EVALUATION METRICS

In this section, we formally define the evaluation metrics used in our urban navigation experiments.

**Success Rate (SR; %).** The percentage of episodes in which the robot successfully reaches the goal without collision, averaged across scenes and runs. Higher values indicate better navigation performance.

**Collision Time (CT; %).** The fraction of total episode time during which the robot is in contact with any obstacle. Lower values indicate safer navigation.

**Route Completion (RC; %).** The percentage of the planned evaluation route completed before termination (goal reached, fatal collision, off-road deviation, or timeout). Higher values reflect more reliable progress.

**Distance to Goal (DTG; m).** The final Euclidean distance from the robot to the goal at episode termination. Lower values indicate more precise goal reaching.

### L.2 DETAILS OF TRAINING AND EVALUATION SETUP

For training and evaluation, agents are initialized within the annotated traversable regions in Ur-banVerse scenes. The goal point is randomly sampled at a distance between 10 m and 30 m from the starting pose. Each episode runs for a fixed horizon (60s@5Hz), during which the policy must navigate toward the goal while avoiding collisions and staying within the traversable area. Rollouts are collected with a specified horizon length (*i.e.*, 32 steps) for training updates. For both training and evaluation, an episode is terminated either when the robot collides with any object in the scene or when the maximum time horizon is reached.

### L.3 DETAILS OF POLICY LEARNING

**Model architecture.** For RL training, we adopt an actor–critic architecture with continuous action space, trained using PPO (Schulman et al., 2017). Each observation is the relative position of the goal point and an RGB frame of size $135 \times 240$ with three channels. The convolutional encoder consists of three layers with depths [16, 32, 64], each followed by ReLU activation. The goal point is encoded using a MLP. No special regularization is applied. The encoder output is passed through a MLP with three hidden layers of size 128 and ELU activations. Actor and critic share the same backbone. The output distribution uses Gaussian parameters with unconstrained $\mu$ and $\sigma$, initialized respectively with the default initializer and a constant (0.0).

**Reward functions.** The reward is designed to encourage goal-reaching while penalizing unsafe behaviors. Specifically:

$$R = R_A + R_C + R_P + R_V \tag{1}$$

- Arrived reward $R_A$: A large positive reward (+2000) is given when the agent successfully reaches the goal.
- Collision penalty $R_C$: A penalty (-200) is applied if the agent collides with any obstacle.
- Position tracking $R_P$: A shaping reward based on the error between the commanded and actual position, with two scales: coarse (std = 5.0, weight = 10) $R_{P,c}$ and fine (std = 1.0, weight = 50) $R_{P,f}$.
- Velocity reward $R_V$: A reward (weight = 10) encourages matching the target velocity command, defined as the cosine similarity between the current velocity and target velocity between the robot position and target position.

This combination balances sparse terminal signals (arrival, collision) with dense shaping terms (tracking error, velocity), stabilizing training and guiding exploration.

**Optimization and training hyperparameters.** We provide the detailed hyperparameters in Tab. 15.

### L.4 ROBOT PLATFORM CONFIGURATIONS IN SIMULATION

Our simulated platforms are implemented in NVIDIA IsaacSim (Xu et al., 2022; Mittal et al., 2025), a GPU-accelerated environment that provides physics-accurate interactions and photorealistic rendering. Each robot model is instantiated from its official URDF specification and equipped with RGB sensing for visual input.

Locomotion is governed by a layered control architecture composed of a low-level joint controller and a high-level policy optimized through reinforcement learning. Policies are trained with Proximal Policy Optimization (PPO) (Schulman et al., 2017) under curriculum learning and terrain randomization. The reward design promotes balance, velocity tracking, and energy-efficient movement, while penalizing collisions and falls. To enhance sim-to-real transfer, domain randomization is applied across textures, friction parameters, and mass properties. This unified framework is applied consistently across all robot embodiments, supporting scalable and embodiment-aware training.

**Wheeled robot.** The wheeled platform is modeled as a differential-drive robot controlled via a kinematic formulation (Polack et al., 2017). Linear and angular velocities $(v, \omega)$ are generated from waypoints using an ideal PD controller (Sridhar et al., 2024). These commands are propagated in

| Parameter | Value |
|---|---|
| Learning rate | $1 \times 10^{-4}$ (adaptive schedule) |
| Discount factor $\gamma$ | 0.99 |
| GAE parameter $\tau$ | 0.95 |
| PPO clipping $\epsilon$ | 0.2 |
| KL threshold | 0.01 |
| Entropy coefficient | 0.002 |
| Critic loss coefficient | 1.0 |
| Gradient norm clipping | 1.0 |
| Horizon length | 32 |
| Minibatch size | 512 |
| Mini-epochs | 5 |
| Bounds loss coefficient | 0.01 |
| Training epochs | 1500 |
| Device | Single GPU (L40S), mixed precision |

Table 15: Optimization and training hyperparameters.

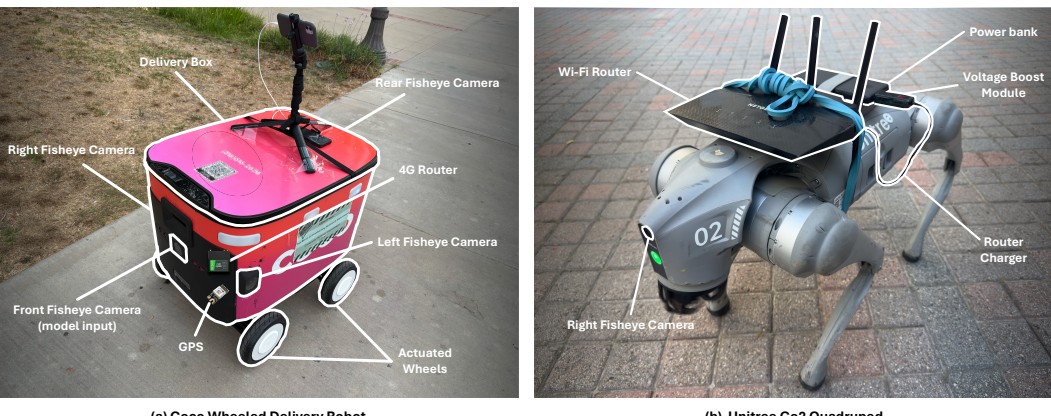

(a) Coco Wheeled Delivery Robot          (b) Unitree Go2 Quadruped

Figure 36: **Robot platform configurations used in real-world experiments: (a) Coco wheeled delivery robot and (b) Unitree Go2 quadruped.**

IsaacSim's rigid-body physics engine, where wheel-ground frictional contact governs the realized motion.

**Unitree Go2 quadruped.** The quadruped embodiment (Unitree Go2) is designed for agile locomotion in unstructured environments. Control actions are expressed as $(v_x, v_y, \omega)$, again produced from waypoints through an ideal PD controller (Sridhar et al., 2024). The locomotion module itself is a compact MLP trained on IsaacSim's standard quadruped training setups (Xu et al., 2022; Mittal et al., 2025), enabling stable gait generation across diverse terrains.

### L.5    ROBOT PLATFORM CONFIGURATIONS IN REAL-WORLD

For real-world experiments, we deploy two distinct robot platforms: the Coco wheeled delivery robot and the Unitree Go2 quadruped, as shown in Fig. 35. These platforms represent complementary embodiments for urban navigation—a compact, wheeled sidewalk delivery robot and a legged quadruped capable of traversing uneven terrain.

**Coco wheeled robot.** As shown in Fig. 36 (a), the real Coco robot is equipped with four actuated wheels, differential-drive odometry, a GPS unit, and multiple fisheye cameras (front, rear, left, and right) that provide panoramic perception. Its sensing and communication stack includes a 4G router for remote teleoperation connectivity and a delivery box as payload. The front fisheye camera serves

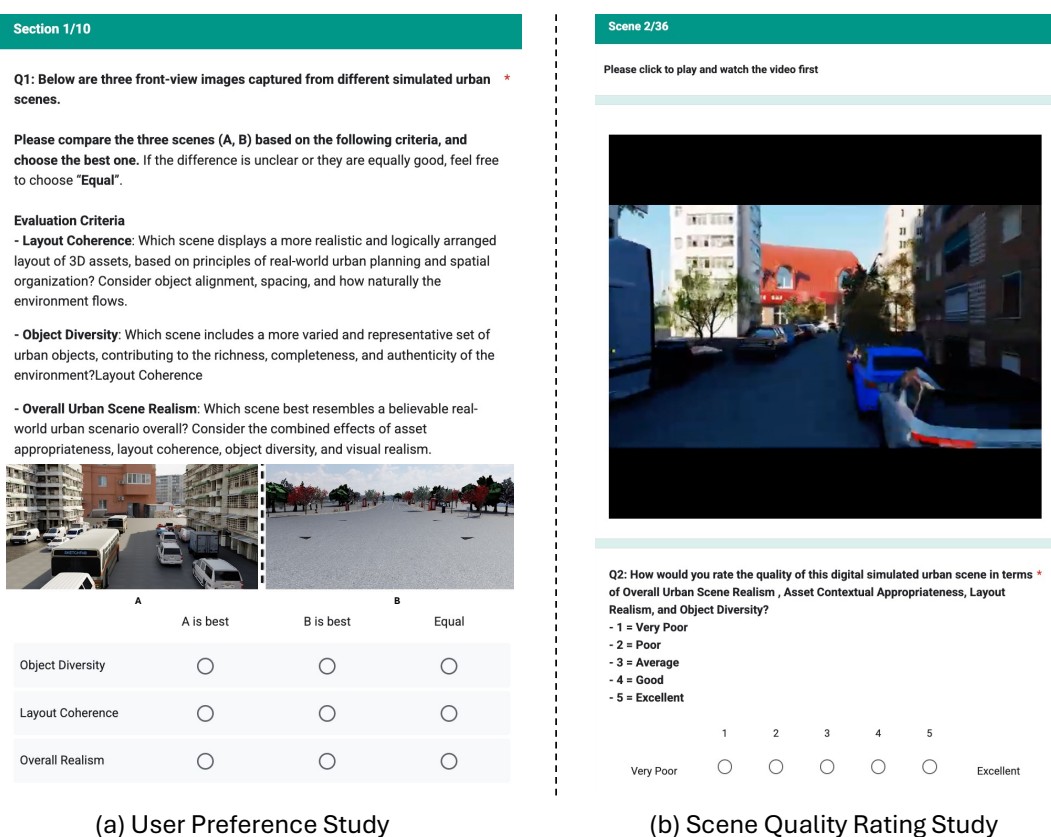

(a) User Preference Study  (b) Scene Quality Rating Study

Figure 37: **User study interface example.**

as the primary input to our policy models. The robot is controlled via the same kinematic model used in simulation, where linear and angular velocities $(v, \omega)$ are computed from waypoints using an ideal PD controller. Odometry is used for real-time position estimation and to continuously update the target position during navigation.

**Unitree Go2 quadruped.** As shown in Fig. 36 (b), the Unitree Go2 offers a contrasting embodiment with articulated legs and onboard compute support. It is equipped with a fisheye perception camera, a Wi-Fi router for connectivity, and an extended power system consisting of a power bank, voltage boost module, and router charger to support sustained experiments. Low-level locomotion is handled by the controller provided by Unitree. Instead of executing joint-level commands from a trained policy, we interface with the Go2 through its built-in velocity control API, sending high-level commands $(v_x, v_y, \omega)$ that leverage its native gait generation and stability modules. LiDAR-based odometry is used for real-time position estimation and to continuously update the target position during navigation.

Together, these two platforms enable us to examine sim-to-real transfer across distinct locomotion modalities and hardware configurations, thereby providing a broader evaluation of embodied navigation in diverse urban settings.

## M  HUMAN EVALUATION DETAILS

To better understand how human judgments align with the realism of generated scenes, we designed two complementary user studies using a custom-built online interface, as illustrated in Fig. 37.

**Comparative User Preference Study.** For the first study, participants were presented with pairs of static overview images randomly sampled from our dataset. Each pair consisted of one scene generated by UrbanVerse and one procedurally generated (PG) scene using UrbanSim (Wu et al., 2025b) built from the same UrbanVerse-100K assets. As shown in Fig. 37(a), participants compared

the two scenes side-by-side and were asked to select which scene performed better (or "Equal") across three criteria: (1) *Object Diversity* — the richness and representativeness of included urban objects; (2) *Layout Coherence* — the realism and logical arrangement of objects based on real-world urban design principles; and (3) *Overall Realism* — the degree to which the scene resembled a plausible real-world street environment. Responses were collected through a simple three-choice interface (A better, B better, Equal) for each criterion.

**Scene Quality Rating Study.** In the second study, participants evaluated scene quality through immersive video walkthroughs. Each trial displayed a 360° simulated flythrough of a given scene, as illustrated in Fig. 37(b). Participants were then asked to assign a quality score (1–5) reflecting overall realism, asset contextual appropriateness, layout realism, and object diversity. The Likert-style rating scale ranged from 1 = *Very Poor* to 5 = *Excellent*. This setup allowed participants to assess not only static composition but also temporal and spatial coherence as the camera moved through the scene.

**Study Deployment.** Both studies were conducted with 32 undergraduate participants. To ensure fairness, all scenes and videos were shuffled randomly across participants, and instructions clarified the evaluation criteria prior to the study. This interface design ensured consistent, criterion-driven human judgments across both comparative and rating-based evaluations.

