# OpenReview forum: "UrbanVerse: Scaling Urban Simulation by Watching City-Tour Videos"
_ICLR.cc/2026/Conference — ICLR 2026 Poster_

### Official Review · Reviewer_eUF4 · 2025-10-20

**Soundness:** 3
**Presentation:** 3
**Contribution:** 3
**Rating:** 8
**Confidence:** 4

**Summary:**

The paper provides a new dataset of 100k assets for urban scene simulators, including physical properties, along with ~300 PBR materials for ground surfaces, and ~300 lighting skyboxes. The process for data filtering and annotation is documented as well.

The paper also proposes a real2sim process to convert ego-centric urban navigation videos into interactive simulators. This is done by processing videos to extract a scene graph, match the text and geometry of the scene objects to assets in the above database, and then place assets in corresponding locations in the scene.

Experiments demonstrate the quality of this approach compared to a baseline model. Human preference scores for scenes (in diversity, coherence, and quality) favor the model over a baseline, nearing a ceiling set of scores from a set of human-crafted scenes. RL policy learning shows power law scaling both in the number of distinct scenes (from videos) and variations of scenes per video. Real-world evaluations add further rigor to the success of the sim2real transfer, including an extended navigation task with minimal human intervention for success.

**Strengths:**

# originality
- Combines a variety of techniques to produce high quality scenes. The novelty is not so much the problem or any individual component of the generation process, but the combination is a non-trivial addition given the strong results.

# quality
- Experiments show very strong sim2real results.
- Provides a massive and usable asset dataset that has been cleaned up and validated.

# clarity
- The data preparation process is thoroughly explained, as is the generation process. The plan to open-source will further facilitate reproduction, adoption, and extension.
- The narrative and steps were easy to follow.

# significance
- sim2real for urban scene navigation is directly valuable to the autonomous driving community. The assets will be of general value to downstream work on improved generation methods, with clear improvement over existing options and baselines (KITTI).

**Weaknesses:**

# originality
- No single part of the approach differs from established practices. This is fine given the power of the contribution.

# quality
- It would help to provide data on the scaling and costs of the different methods. For example: how many LLM/VLM calls are needed? how much wall clock time to process a scene? how do these vary in the size of the scene or length of the video?

# clarity
- The sim2real evaluation metrics would benefit from definitions (see below).

**Questions:**

# questions
- How does UrbanVerse scale?
	- How many VLM/LLM calls are needed?
	- How much wall time / compute?
	- How does UrbanVerse compare to UrbanSim in these dimensions?
- How does this scale with input video length or scene size (in meters, number of entities/assets, and so on)?
	- Is there any ablation data on training based on different video lengths or scene dimensions?
- lines 281-281: How sensitive are the outcomes to assigned parameters? How are they assigned?
- line 476: What were the two interventions needed?
- Table 2: Define the evaluation metrics. Some were not clear to me:
	- How does success rate differ to route completion? When can you complete the route without succeeding?
	- CT: Should the arrow be arrow down or up?
	- "Collision times" is ambiguous. It's not clear if this means "(mean) time to collision" vs "number of times there was a collision", for example.
	- The upward arrow suggests lower is better, implying the latter. But on that metric, PPO-UrbanVerse does quite poorly in Table 2 and for wheeled in Table 3. But well for quadruped? It's a bit confusing how to interpret this.


# suggestions
- Table 1: Separate "Ast." as a clearly marked final column as it is evaluated on different data (and thus not quite continuous with the other metrics).
- Figure 7: Would be nice to increase the scale to see where improvements asymptotically saturate. Given the layouts are limited to 32 environments it would suffice to increase the number of training cousins to show where success rate asymptotes (near 100%?).

---

> ### Author Response · Authors · 2025-11-25
> **Author Response to Reviewer eUF4 (1/5)**
>
> Thank you for carefully reviewing our paper and for your valuable feedback, as well as for assigning an initial score of 8. We appreciate your positive comments, especially your recognition that our scene-generation pipeline is *“a non-trivial addition”*, that our experiments show *“very strong sim2real results”*, and that the work is also *“directly valuable to the autonomous driving community”* and *“of general value to downstream work”*. We also thank you for acknowledging our open-source plan. This is exactly what we aimed for: we hope UrbanVerse’s assets, scenes, and pipelines benefit not only sidewalk autonomy but also autonomous driving, 3D scene generation, and related fields. To support this broader impact, all components are modular and can be used independently.
>
> We address your questions and incorporate your suggestions point by point below, and would greatly appreciate it if you could let us know whether our responses satisfactorily resolve your concerns.
>
> ------------------------------------------------------------------------------------
>
> > **Question 1:** It would help to provide data on the scaling and costs of the different methods. How does UrbanVerse scale:
> > - How many VLM/LLM calls are needed?
> > - How much wall time / compute?
> > - How does UrbanVerse compare to UrbanSim in these dimensions?
>
> **Answer 1:** Thank you for the helpful suggestion. In response, we provide a detailed computational and scaling analysis for both the UrbanVerse-Gen scene generation pipeline and the UrbanVerse-100K annotation process, along with a comparison to UrbanSim. These results are summarized below in **`Table R1`**, **`Table R2`**, **`Table R3`**, and **`Table R4`**. *We have also included this computational analysis in **`Appendix L`** of the revised paper.* We detail each analysis below.
>
> **(1) UrbanVerse-Gen Scene Generation Process:** As shown in **`Table R1`**, the computational cost and multimodal LLM (GPT-4.1) calls of UrbanVerse-Gen scale nearly linearly with video length. The main computation comes from the 3D reconstruction step (MAST3R), while GPT-4.1 is used efficiently by querying only every third frame to obtain object categories. Short clips (10–40 seconds) require 4–14 LLM calls and 12–114 seconds of processing, and longer clips (80–180 seconds) require 27–60 calls and 289–1135 seconds on an NVIDIA H100 GPU. In our experiments, as summarized in **`Table R2`**, with city-tour clips averaging 180 seconds, we generate 160 fully interactive scenes in 1.26 hours using 4 H100 GPUs.
>
> **(2) UrbanVerse-100K Annotation Process:** Further, as shown in **`Table R3`**, the UrbanVerse-100K annotation pipeline scales efficiently: each asset requires exactly **one** LLM call (GPT-4.1), averaging only 2.3 seconds and _$0.013_ per object. Annotating the full, massive 102,530 object dataset takes 65.5 hours of wall-clock time and _$1,334_ in API cost, demonstrating that our large-scale, physics- and semantics-rich asset annotation is both practical and economical.
>
> **(3) Comparison with UrbanSim:** UrbanSim uses procedural generation, where developers hand-design scene templates (e.g., placing objects along sidewalks at fixed intervals) and sample assets accordingly. Although this requires no model calls during scene generation (sampling), PG produces unrealistic layouts and still requires **manual scene creation** and **manual asset annotation**. In contrast, as compared in **`Table R4`**, **UrbanVerse automates both processes**, reducing scene creation from ~240 minutes to ~18.9 minutes and asset annotation from ~600 seconds to ~2.3 seconds per object, while achieving the same rendering efficiency (94 FPS; both uses IsaacSim simulation engine). This makes UrbanVerse far more scalable and better aligned with real-world distributions.

---

> ### Author Response · Authors · 2025-11-25
> **Author Response to Reviewer eUF4 (2/5)**
>
> ----------------------------------------------------------------------------------------------------
>
> **`Table R1`:** UrbanVerse-Gen real-to-sim scene generation time with varying input video lengths on an NVIDIA H100 GPU
>
> | **Video Duration (sec)** | **Video Length (# frames)** | **LLM Calls** | **Scene Generation Wall Time (sec)** |
> |--------------------------|-----------------------------|---------------|--------------------------------------|
> | 10                       | 10                          | 4             | 12.38                                |
> | 40                       | 40                          | 14            | 114.46                               |
> | 80                       | 80                          | 27            | 289.80                               |
> | 180                      | 180                         | 60            | 1135.20                              |
>
>
>
> ----------------------------------------------------------------------------------------------------
>
>
>
>
> **`Table R2`:** Overall computational time of UrbanVerse simulation scenes
>
> | **Setting**                          | **# Input City-tour Videos** | **# Cousin Scenes per Layout** | **# Unique Scene Layouts** | **# Total Simulation Scenes** | **Wall Time** |
> |--------------------------------------|-------------------------------|--------------------------------|-----------------------------|-------------------------------|---------------|
> | Single layout with 5 digital cousins | 1                             | 5                              | 1                           | 5                             | 18.92 min     |
> | 160 Scenes (1 × NVIDIA H100)         | 32                            | 5                              | 32                          | 160                           | 10.08 hrs     |
> | 160 Scenes (4 × NVIDIA H100, default) | 32                           | 5                              | 32                          | 160                           | 1.26 hrs      |
>
>
> ----------------------------------------------------------------------------------------------------
>
>
>
> **`Table R3`:** Annotation cost and runtime statistics of the UrbanVerse-100K annotation pipeline
>
> | **# Objects** | **GPT-4.1 Call Counts** | **API Wall Clock Time** | **API Cost** |
> |---------------|--------------------------|---------------------------|--------------|
> | 1             | 1                        | 0.0003 hrs               | $0.018       |
> | 2             | 2                        | 0.0005 hrs               | $0.029       |
> | ...           | ...                      | ...                      | ...          |
> | 102,530       | 102,530                  | 65.5 hrs                 | $1,334       |
> | **Average**   | 1 / object               | 2.3 sec / object         | $0.013 / object |
>
>
> ----------------------------------------------------------------------------------------------------
>
>
>
> **`Table R4`:** Comparison between UrbanSim and UrbanVerse across asset creation, scene generation, and rendering efficiency.
>
> | **Aspect** | **UrbanSim** | **UrbanVerse** |
> |-----------|--------------|----------------|
> | How to Add a New 3D Asset | Manual annotation (e.g., resizing to metric scale, setting mass) | Fully automated annotation pipeline |
> | Time to Add a New 3D Asset | ~600 seconds | ~2.3 seconds |
> | How to Add a New Scene | Manual creation of scene templates | Automatic scene generation via UrbanVerse-Gen |
> | Time to Create a New Scene (~200m) | ~240 minutes | ~18.9 minutes (NVIDIA H100 GPU) |
> | Rendering Efficiency (RGB, Single Env, NVIDIA L40S GPU) | ~94 FPS | ~94 FPS |
>
> ----------------------------------------------------------------------------------------------------

---

> > ### Comment · Reviewer_eUF4 · 2025-11-28
> >
> > Minor question: What hardware is used for the times reported for UrbanSim?

---

> ### Author Response · Authors · 2025-11-25
> **Author Response to Reviewer eUF4 (3/5)**
>
> > **Question 2:** How does this scale with input video length or scene size (in meters, number of entities/assets, and so on)? Is there any ablation data on training based on different video lengths or scene dimensions?
>
> **Answer 2:** Excellent question and idea. Following your suggestion, we conducted a control experiment on training scene horizon length by comparing **(i) Half-length UrbanVerse scenes**, where each scene is truncated to roughly half its original length (_~100 m_), and **(ii) Full-length UrbanVerse scenes**, which use the complete layouts (_~200 m_). Both policies are trained on all 160 UrbanVerse scenes (*320 truncated scenes in the half-length case*) using a wheeled robot and evaluated on the ten CraftBench test scenes. As the results shown in **`Table R5`** below, full-length scenes yield substantially better performance, improving Success Rate (+9.6), reducing Collision Times (–5.0), and increasing Route Completion (+11.6). *We have included this new experiment in **`Appendix H`** of the revised paper.*
>
> We attribute this improvement to two factors. First, longer scene horizons expose the agent to a wider range of spatial layouts, object configurations, and long-range dependencies—matching the complexity of real-world navigation and the realistic, artist-designed CraftBench scenes used for evaluation. This increases distributional diversity and encourages more robust policy learning. Second, longer scenes naturally create longer training trajectories, enabling PPO to observe more varied transitions, more obstacle interactions, and more challenging decision points per episode. This richer signal improves both collision avoidance and global route planning.
>
> Overall, these results show that scaling scene length—and therefore spatial complexity—is an important factor in UrbanVerse’s ability to improve policy generalization. Thank you again for the insightful suggestion.
>
> **`Table R5`:** *Influence of UrbanVerse scene horizon length on policy performance.* Both policies are trained on all 160 UrbanVerse scenes. In the Half-Length setting, each training scene is truncated to half its original length; the Full-Length setting uses the complete scenes. All policies are trained on a wheeled robot and evaluated on the ten CraftBench test scenes. We report the average Success Rate (SR), Collision Times (CT), and Route Completion (RC).
>
> | **Training Scene Length** | **SR ↑** | **CT ↓** | **RC ↑** |
> |---------------------------|---------:|---------:|---------:|
> | Half Length (~100m)       | 32.3     | 40.5     | 50.8     |
> | Full Length (~200m)       | **41.9** | **35.5** | **62.4** |
>
>
> ------------------------------------------------------------------------------------
>
> > **Question 3:** lines 281-281: How sensitive are the outcomes to assigned parameters? How are they assigned?
>
> **Answer 3:** UrbanVerse uses NVIDIA IsaacSim (PhysX 5) as its physics simulation engine. All semantic and physical parameters are assigned directly from our UrbanVerse-100K annotations through IsaacSim APIs, including category labels, mass, friction, and affordance tags for each asset. Our simulation are fully interactive (not static meshes) and thus requires plausible physical parameters to function correctly, without them, rigid-body simulation cannot run. For this reason, we could not perform a separate sensitivity study. In IsaacSim, incorrect physical values can immediately destabilize the simulation (e.g., an object with an unrealistically low mass may bounce or drift uncontrollably, while an object with an extremely large mass may sink into the ground or cause numerical instability). To this end, we densely annotated physical parameters across the entire UrbanVerse-100K database. Our goal in this work is to ensure that all assets are physics-ready and that the generated scenes remain stable and fully interactive.
>
> ------------------------------------------------------------------------------------
>
> > **Question 4:** line 476: What were the two interventions needed? (337m long-horizon experiments)
>
> **Answer 4:** For the 337 m long-horizon real-world experiment, we deployed PPO-UrbanVerse policy on a wheeled delivery robot using a human–AI shared-autonomy teleoperation system that allows safety interventions when needed. As shown in the experiment recording video ([**Click Here: Long-horizon Experiment Video** (anonymous)](https://anonymoususeruseanonymousname.github.io/realworld_long_horizon.html)): *i)* the first intervention occurred due to *temporary GPS drift* (at 0:17), and *ii)* the second occurred when some pedestrians intentionally approached the robot out of curiosity (at 1:05), so the operator stepped in for safety. Aside from these two interventions, our PPO-UrbanVerse policy successfully navigated the full 337 m route autonomously, further highlighting both the effectiveness of our approach and the practical challenges of real-world deployment.

---

> ### Author Response · Authors · 2025-11-25
> **Author Response to Reviewer eUF4 (4/5)**
>
> > **Question 5:** The sim2real evaluation metrics would benefit from definitions (see below); Table 3: Define the evaluation metrics.
> > - How does success rate differ to route completion? When can you complete the route without succeeding?
> > - CT: Should the arrow be arrow down or up?
> > - “Collision times’’ is ambiguous. It's not clear if this means “(mean) time to collision’’ vs “number of times there was a collision”, for example.
> > - The upward arrow suggests lower is better, implying the latter. But on that metric, PPO-UrbanVerse does quite poorly in Table 3 and for wheeled in Table 4. But well for quadruped? It's a bit confusing how to interpret this.
>
>
> **Answer 5:** Thank you for the suggestion. *We have added clear definitions of all evaluation metrics in **`Appendix J.1`** in the revised paper, and directed the reader to this section in the main paper.* Formally, they are:
>
> - **Success Rate (SR; %)**: Percentage of episodes in which the robot successfully reaches the goal without any collision, averaged across scenes and runs (↑ higher is better).
> - **Collision Time (CT; %)**: Percentage of total episode time during which the robot is in contact with any obstacle (↓ lower is better).
> - **Route Completion (RC; %)**: Percentage of the planned (testing) route distance completed before termination (goal reached, fatal collision, going off-road, or time limit) (↑ higher is better).
> - **Distance to Goal (DTG; m)**: The final Euclidean distance in meters from the robot to the goal when the episode terminated or ends (↓ lower is better).
>
> Next, based these definitions, we clarify your sub-questions in the following.
>
> **(1) Difference between Success Rate and Route Completion:** Success Rate measures whether the robot reaches the goal without any collision. Route Completion measures what percentage of the route the robot completes before the episode ends (goal reached, fatal collision, going off-road, or time limit). Intuitively, a robot can complete most of the route but still *not* succeed if it fatally collides and goes off-road (e.g., falls off the curb) near the end goal point or stops early for safety. In other words, Route Completion can be viewed as a softer measure of Success Rate.
>
> **(2) Arrow direction for Collision Time:** Collision Time measures the percentage of time the robot is in contact with obstacles. Lower is always better, so it should have a down arrow. We have corrected this.
>
> **(3) Meaning of Collision Time:** Collision Time does not count the number of collisions. It measures the percentage of episode time during which the robot is colliding with any object.
>
> **(4) Clarification on Collision Time (CT) in Table 3 and Table 4:** Collision Time (CT) is a “lower is better’’ metric (↓). You are correct that PPO-UrbanVerse does *not* achieve the best CT in the simulation evaluation (CraftBench) in **`Table 3`**; *the boldface was a typo, and we have corrected it in the revised manuscript*. Even so, PPO-UrbanVerse still achieves the best Success Rate (41.9%) and Route Completion (62.4%). In **`Table 3`**, our method achieved higher Success Rate but worse Collision Time because it can reliably reach the goal while occasionally making brief, non-fatal contacts with obstacles that increase CT without preventing success. In contrast, in the real-world evaluation (**`Table 4`**), PPO-UrbanVerse actually achieves the *lowest* Collision Time for both the wheeled robot (22.9) and the quadruped (10.4), outperforming all baselines.

---

> ### Author Response · Authors · 2025-11-25
> **Author Response to Reviewer eUF4 (5/5)**
>
> > **Suggestion 1:** Table 2: Separate ``Ast.'' as a clearly marked final column as it is evaluated on different data (and thus not quite continuous with the other metrics).
>
> **Action 1:** Thank you for the helpful suggestion. We have updated **`Table 2`** accordingly in the revised paper, by moving **Ast.** to a clearly separated final column to distinguish it from the other metrics.
>
> ------------------------------------------------------------------------------------
>
> > **Suggestion 2:** Figure 9: Would be nice to increase the scale to see where improvements asymptotically saturate. Given the layouts are limited to 32 environments it would suffice to increase the number of training cousins to show where success rate asymptotes (near 100%?).
>
>
> **Action 2:** Great idea. We were curious about this as well. Following your idea, we **generated another 160 new UrbanVerse simulation environments** using UrbanVerse-Gen from 32 new city-tour videos (32 new layouts X 5 digital cousins each). Combined with the original 160 scenes, we now have **320 scenes built from 64 unique layouts**. We are currently running the larger-scale RL training on these 320 scenes. The training takes a bit longer because of GPU server queueing. *We expect to finish this experiment later next week and will come back to update and share the results. Thank you for your understanding.*
>
> *Update - The experiment is now complete:* As shown in **`Table R6`** below, we observe that scaling from 160 to 320 UrbanVerse scenes continues to improve the PPO-UrbanVerse policy, with *no clear sign of saturation* yet. These results confirm that further scaling up UrbanVerse scenes continues to yield measurable gains. UrbanVerse is an ongoing long-term project, and we will keep expanding and scaling UrbanVerse scenes to further investigate where performance eventually saturates.
>
> **`Table R6`**: Scaling the PPO navigation policy with additional **UrbanVerse** scenes (160 → 320). We report performance tested on CraftBench.
>
> | # Training UrbanVerse Scenes                | SR ↑ | CT ↓ | RC ↑ |
> |---------------------------------------------|------|------|------|
> | 160 Scenes (32 layouts × 5 cousins)         | 41.9 | 35.5 | 62.4 |
> | 320 Scenes (64 layouts × 5 cousins)         | **47.4** | **22.1** | **71.9** |
>
>
> ------------------------------------------------------------------------------------
>
> *We hope our responses have clarified the scaling costs and efficiency of UrbanVerse, our evaluation metrics definitions, the ablation study on scene length you suggested, and the updates we made to the tables. We appreciate your positive and constructive feedback. If our answers fully address your questions, we would be grateful if you would consider updating your score. Thank you!*

---

> > ### Comment · Reviewer_eUF4 · 2025-11-28
> >
> > Thank you for the thorough responses. These reflect my current scoring and view of the work well.
> >
> > I very much appreciate the detailed and specific clarifications and additional experiments and videos.

---

> ### Author Response · Authors · 2025-11-28
> **Author Response to Reviewer eUF4's New Minor Question: What hardware is used for the times reported for UrbanSim?**
>
> > **New Minor question:** What hardware is used for the times reported for UrbanSim?
>
> **Answer:** Happy to answer and further clarify this new question!
>
> The timing numbers in **`Table R4`** (Response (4/5)) were measured on different machines for different compared dimensions, because UrbanSim’s tooling requires different configurations. For each compared dimension, *UrbanSim and UrbanVerse are always measured on the same machine* to ensure fairness. Below, we detail the exact hardware used, what each UrbanSim timing corresponds to and measured.
>
> **1. Time to Add a New 3D Asset (Row 3 of Table R4)**
> - **Hardware:** RTX 4080 desktop
> - **Reason:** This process is ***manual*** for UrbanSim, so timing is not tied to machine performance. This manual annotation tool requires graphic interface.
> - **What is measured for UrbanSim:** The average time required for the full manual annotation workflow using UrbanSim’s asset annotation tool, which includes the following steps:
>   *Step 1:* Import the raw 3D asset ( in its relative scale + coordinate system) into an calibrated metric scale.
>   *Step 2:* Manually adjust the asset's height/width/length until they look reasonable relative to the scene; align its facing direction (e.g., +Y); recenter to origin.
>   *Step 3:* Export the final asset.
> - **Measured Time:** This whole workflow consumes ~600 seconds ( ~10 minutes )
>
> **2. Time to Create a New Scene (Row 5 of Table R4)**
> - **Hardware:** RTX 4080 desktop
> - **Reason:** Also a ***manual*** process for UrbanSim, so hardware does not influence the measurement. This process requires graphic interface of IsaacSim.
> - **What is measured:** The average time consumed to handcraft a new procedural scene template, which involves the following steps:
>   *Step 1:* Design a new ground-plane layout (e.g., sidewalk/road/building zone ratios) and find corresponding PBR materials and convert it to MLD format to texture each component.
>   *Step 2:* Write/code a new configuration file defining which object categories can appear in which zones and at what placement intervals.
>   *Step 3:* Export this template for the later procedural sampling.
> - **Measured Time:** This whole workflow consumes ~240 minutes
>
> **3. Rendering Efficiency (Row 6 of Table R4)**
> - **Hardware:** **L40S GPU** for both UrbanSim and UrbanVerse
> - **Reason:** Both methods run on IsaacSim, so rendering efficiency is directly comparable under identical hardware under a headless mode.
> - **What is measured:** Scene rendering efficiency
> - **Measured Time:** ~94 FPS on average.

---

> ### Author Response · Authors · 2025-11-28
> **Author Response to Reviewer eUF4: Thank You for Reaffirming the Initial Score 8 After the Rebuttal**
>
> Dear Reviewer eUF4,
>
> Thank you for the positive feedback and for taking the time to review our paper and rebuttal thoroughly.
>
> We are glad that the clarifications, additional experiments, and demo videos provided during the rebuttal *addressed your concerns* and *further reinforced your initial positive score of 8*.
>
> Thank you as well for the suggestions that helped strengthen the revised version of UrbanVerse.
>
> Best regards,
>
> Authors of Submission #9637

---

### Official Review · Reviewer_ND3o · 2025-10-26

**Soundness:** 3
**Presentation:** 3
**Contribution:** 1
**Rating:** 2
**Confidence:** 5

**Summary:**

This paper presents UrbanVerse, a data-driven real-to-sim system that automatically converts city-tour videos into physics-aware, interactive simulation environments for training urban embodied AI agents. The system comprises two main components: (1) UrbanVerse-100K,  a curated repository of 102,530 high-quality metric-scale urban object assets, and (2) UrbanVerse-Gen, an automated pipeline that extracts scene layouts from real-world videos and instantiates simulation scenes by retrieving matched assets. The authors construct 160 training scenes from city-tour videos spanning 24 countries and demonstrate that policies trained in these environments follow scaling power laws and achieve strong sim-to-real transfer performance in mapless urban navigation tasks, including a 337-meter real-world deployment with only two human interventions.

**Strengths:**

1. UrbanVerse presents an effective framework that retrieves similar 3D assets through video reference and employs a mapping mechanism to construct street scenes that closely approximate real-world environments. This framework theoretically allow the system to recreate any scenario captured in city-tour videos.
2. The author establishes a comprehensive evaluation framework with six reasonable metrics (Cat, Ast, Cov, Lay, Loc, and Div) that assess different aspects of reconstruction fidelity. Combined with human evaluation, this hybrid evaluation thoroughly validates the effectiveness of the proposed framework.
3. The navigation experiments prove that scene diversity significantly enhances the generalization capability of navigation agents, which highlights the urbanverse's potential capability.
4. The long-distance real-world deployment provides convincing evidence of UrbanVerse's excellent sim-to-real transfer capabilities.

**Weaknesses:**

1. UrbanVerse currently exhibits limited interactivity between agents and the environment. After reading the paper, I assume that the urbanverse only supports collision response now. The lack of dynamic environment interactions (pedestrians, moving cars, etc.) prevents UrbanVerse from serving as a general-purpose platform for diverse urban embodied AI research, limiting its utility to primarily perception-based navigation scenarios.

2. The paper does not convincingly articulate why researchers should adopt UrbanVerse over these established alternatives beyond improved visual realism. Without demonstrating substantial functional advantages—such as superior sim-to-real transfer, computational efficiency, or unique interaction capabilities—the contribution appears incremental. The claim that UrbanVerse addresses limitations of procedural generation is weakened by the fact that existing simulators already achieve effective sim-to-real transfer for navigation tasks.

3. Without clear documentation of how the community can leverage this work—whether through asset libraries, simulation APIs, or integration tools—the practical impact remains limited. The contribution of  this urban simulation is undermined by the absence of concrete plans for open-source release, developer documentation, or community engagement strategies.

**Questions:**

q1: Please supplement the functional innovations of UrbanVerse compared to other simulators (CARLA, MetaUrban, UrbanSim), including: supported embodied intelligence tasks, rendering efficiency comparison, supported agent types, multi-agent interaction, and environmental dynamic disturbances.
q2:Please describe the user-side pipeline of UrbanVerse. Based on the current content in the article, it is difficult to determine UrbanVerse's potential contributions to the community.

---

> ### Author Response · Authors · 2025-11-25
> **Author Response to Reviewer ND3o (1/6)**
>
> Thank you for the constructive and valuable feedback. We appreciate the positive comments on our real-to-sim scene generation framework being *"effective"*, as well as the acknowledgement that our navigation experiments demonstrate how scene diversity *"significantly enhances"* agent generalization and provide *"convincing evidence"* of UrbanVerse’s strong sim-to-real transfer.
>
> We address your concerns and incorporate your suggestions point by point below, and would greatly appreciate it if you could let us know whether our responses satisfactorily resolve your questions.
>
> ------------------------------------------------------------------------------------
>
> > **Question 1:** UrbanVerse currently exhibits limited interactivity between agents and the environment. After reading the paper, I assume that the urbanverse only supports collision response now. The lack of dynamic environment interactions (pedestrians, moving cars, etc.) prevents UrbanVerse from serving as a general-purpose platform for diverse urban embodied AI research, limiting its utility to primarily perception-based navigation scenarios.
>
>
> **Answer 1:** Thank you for raising this concern. These aspects are important to urban simulation, and UrbanVerse already supports them comprehensively. We would like to first clarify that **UrbanVerse scenes are fully interactive in IsaacSim simulation engine and can be populated with diverse dynamic agents**, with rigid-body physics, semantic labels, articulated dynamics, frictional contact, real-time scene editing, and multi-agent interaction. The system supports more than collision-only interactions.
>
> To address your concerns clearly, we first provide several demonstration videos from our *anonymous* UrbanVerse website:
>
> - **[Click Here: UrbanVerse Scenes Populated with Dynamic Agents (anonymous)](https://anonymoususeruseanonymousname.github.io/scene_with_dynas.html)** — two UrbanVerse scenes populated with diverse dynamic agents, including multiple robots (wheeled, quadruped, humanoid), moving pedestrians, cars, wheelchair users, and e-scooter riders.
> - **[Click Here: Mobile Manipulation in UrbanVerse Scenes (anonymous)](https://anonymoususeruseanonymousname.github.io/mobile_manipulator.html)** — two mobile manipulation tasks operated inside UrbanVerse scenes: (1) a wheeled delivery robot with a Franka Panda arm, and (2) a Boston Dynamics Spot quadruped with an arm.
> - **[Click Here: VR Interaction Demo (anonymous)](https://anonymoususeruseanonymousname.github.io/vr_interaction.html)** — a VR-based interaction demo where users can enter an UrbanVerse scene, grab and move objects, and edit the layout physically in real time.
>
> Next, we further clarify how UrbanVerse addresses the specific questions below.
>
> ------------------------------------------------------------------------------------
>
> > **Question 1.1:** Lack of dynamic environment interactions (pedestrians, cars, etc.).
>
> **Answer 1.1:** UrbanVerse indeed *supports interactive dynamic agents generation through a GPU-accelerated multi-agent motion planner based on ORCA* [1] algorithm, enabling cars, pedestrians, and other dynamic agents to move realistically and interact with both static objects and the embodied agent on the fly during training or testing. Here, we provide a video demo of two UrbanVerse scenes populated with diverse dynamic agents, including multiple robots (wheeled, quadruped, humanoid), moving pedestrians, cars, wheelchair users, and e-scooter riders: **[Click Here: UrbanVerse Scenes Populated with Dynamic Agents (anonymous)](https://anonymoususeruseanonymousname.github.io/scene_with_dynas.html)**. Following your suggestion, *we have now added a method description of interactive dynamic-agent population in the revised main paper (Line 307–313) and visualized dynamic-agent scenes in* ***`Figure 5`***.
>
> Specifically, given an UbanVerse scene, we first generate a 2D occupancy map that identifies obstacles, roadways (for cars or scooters), and traversable regions (for pedestrians and mobile machines) using objects' collision surface the semantic annotation. We then sample start–goal pairs for each dynamic agent and compute initial collision-free trajectories using ORCA. During simulation, agents continuously adjust their velocities and positions based on proximity and relative motion, resulting in realistic, collision-aware interactions with both the static environment and dynamic agents. This enables UrbanVerse scenes to support moving cars, pedestrians, scooters, and other dynamic objects, thereby providing dynamic environmental interactions to the embodied agent on the fly. This mechanism is generic and works for any UrbanVerse scene.
>
> [1] Van Den Berg, J., Guy, S. J., Lin, M., \& Manocha, D. Reciprocal n-body collision avoidance. In Robotics Research: The 14th International Symposium ISRR, 2011.

---

> ### Author Response · Authors · 2025-11-25
> **Author Response to Reviewer ND3o (2/6)**
>
> ------------------------------------------------------------------------------------
>
> > **Question 1.2:** Limited utility to primarily perception-based navigation scenarios.
>
> **Answer 1.2:** UrbanVerse is *not* restricted to navigation. In our paper, we focused on urban navigation as a case study because it is a task that can demonstrate the importance of UrbanVerse’s object and layout diversity, as well as the distributional realism of its reconstructed scenes. However, the utility of UrbanVerse extends well beyond perception-based navigation. As shown in the dynamic-agent demo above, UrbanVerse supports interactive multi-agent simulation with moving objects. In addition, as all assets are semantics- and physics-ready (e.g., category labels, mass, friction, and affordances), UranVese enables a broad range of embodied tasks, including semantic-goal navigation, object search, or mobile manipulation. To facilitate this, we provide two mobile manipulation examples: (1) a wheeled delivery robot equipped with a Franka Panda arm, and (2) a Boston Dynamics Spot quadruped with an arm—both operating in UrbanVerse scenes: **[Click Here: Mobile Manipulation in UrbanVerse Scenes (anonymous)](https://anonymoususeruseanonymousname.github.io/mobile_manipulator.html)**. Further, *we have added a description of the tasks supported by UrbanVerse in the revised main paper (Line 334–339) and visualized example tasks in* ***`Figure 6`***.
>
>
> ------------------------------------------------------------------------------------
>
> > **Question 1.3:** Only supports collision response.
>
> **Answer 1.3:** In UrbanVerse scenes, every object is instantiated with complete geometry, semantic information (e.g., category labels and descriptions), physical properties (e.g., mass, friction), and affordance (e.g., openable, graspable) using the rich annotation from the UrbanVerse-1K database. As a result, UrbanVerse supports more than rigid-body collision: agents can push, stack, grasp, and interact with articulated objects, and can also leverage semantic and affordance cues to act in an object-centric, structured manner rather than relying solely on geometry. To illustrate this, we provide a VR-based interaction demo where users can enter an UrbanVerse scene, grab and move objects, and edit the layout physically and in real time: **[Click Here: VR Interaction Demo (anonymous)](https://anonymoususeruseanonymousname.github.io/vr_interaction.html)**.

---

> ### Author Response · Authors · 2025-11-25
> **Author Response to Reviewer ND3o (3/6)**
>
> > **Question 2:** Without clear documentation of how the community can leverage this work—whether through asset libraries, simulation APIs, or integration tools—the practical impact remains limited. The contribution of this urban simulation is undermined by the absence of concrete plans for open-source release, developer documentation, or community engagement strategies.
>
>
> **Answer 2:** We appreciate your concern regarding the developer documentation and open-source release, and community engagement of UrbanVerse project. We provide them and fully address these points below.
>
>
> **(1) “Developer documentation and community engagement strategies”:** We do have extensive developer documentation for UrbanVerse project. *To directly address this concern, we provide a comprehensive fully anonymous UrbanVerse documentation in the website:* **[Click Here: UrbanVerse Documentation (anonymous)](https://anonymousrepohasanonymousname.github.io)**, which includes:
>
> 1. **Open-source plan:** specific release dates, platforms, and data formats.
> 2. **User-side pipeline:** clear instructions on how to use UrbanVerse via APIs or through the built-in asset and scene libraries.
> 3. **Installation, gallery, and quickstart guide:** step-by-step setup instructions with illustrative examples to help users quickly get started.
> 4. **Robot Learning Guide:** reinforcement learning and imitation learning workflows built on UrbanVerse APIs.
> 5. **Developer Guide:** data collection, interfaces, real-world deployment, and instructions for adding new assets or robots, with concrete examples.
> 6. **API Reference:** detailed documentation of all UrbanVerse modules and functions.
> 7. **UrbanVerse Community:** guidelines for community engagement, coding standard, issues, contributions, and support.
>
> *Note: All external URLs in the documentation website have been replaced with anonymous placeholders to preserve anonymity.*
>
> We agree that clear documentation is essential for community adoption. *However, the double-blind submission policy prohibits non-anonymous external links, and a practical documentation site necessarily contains many such links for installation and environment setup. For this reason, we did not include the full documentation in the original submission* and planned to release it together with the codebase after the review period.
>
> **(2) “Concrete plans for open-source release”:** As stated in **`Line 107`** of our original submission, *“all assets, scenes, and code of UrbanVerse will be open-sourced”*, a commitment also *acknowledged* by `Reviewer eUF4 and `Reviewer Nwq702. We fully stand by this commitment. Once the double-blind review period concludes, we will release all resources (licensed under **CC BY 4.0** for broad commercial and non-commercial use). In **`Table R1`** below, we provide a concrete breakdown of our open-source release plan:
>
> ---
>
> **`Table R1`:** UrbanVerse project open-source release plan
>
> | **Content** | **Platform** | **Format** | **Release Month** |
> |-------------|--------------|------------|--------------------|
> | UrbanVerse-100K Asset Database | Hugging Face | `.glb`, `.json`, `.jpg`, `.mdl`, `.hdr` | January 2026 |
> | 160 UrbanVerse Scenes | Hugging Face | `.usd` | January 2026 |
> | CraftBench Scenes | Hugging Face | `.usd` | January 2026 |
> | UrbanVerse-Gen Pipeline | GitHub | `.py` | February 2026 |
> | RL Training Scripts and Checkpoints | GitHub | `.py`, `.pt` | February 2026 |
> | UrbanVerse-100K Annotation Tool | GitHub | `.py` | January 2026 |
> | Documentation & Tutorials | GitHub | `.html` | January 2026 |
>
> ---
>
> We hope these solid materials and clarifications make clear that UrbanVerse is supported by extensive documentation, concrete open-source plans, and clear community engagement, and that its contribution as a scalable, realistic, and fully interactive urban simulation platform is well-founded, rather than undermined.
>
> *Please let us know if there are additional aspects you would like to included in the documentation; we would be happy to further improve it. Thank you!*

---

> ### Author Response · Authors · 2025-11-25
> **Author Response to Reviewer ND3o (4/6)**
>
> > **Question 3:** Please supplement the functional innovations of UrbanVerse compared to other simulators (CARLA, MetaUrban, UrbanSim), including: supported embodied intelligence tasks, rendering efficiency comparison, supported agent types, multi-agent interaction, and environmental dynamic disturbances.
>
>
> **Answer 3:** Thanks for your suggestion. We have now *added a feature comparison that includes the functional aspects you highlighted between UrbanVerse and existing urban simulators (CARLA, MetaUrban, and UrbanSim) in* **`Table 1`** *of the revised paper.* We also provide the expanded comparison and discussion below.
>
> As the function comparison shown in **`Table R2`** below, UrbanVerse introduces three core innovations that substantially reduce the sim-to-real gap:
>
> - **(1)** A novel **automatic real-to-sim, data-driven** scene generation mechanism (UrbanVerse-Gen) that greatly improves scene layout realism and diversity at scale.
> - **(2)** **An order-of-magnitude larger, high-quality asset library (UrbanVerse-100K)** with far more object categories and instances, as well as richer ground materials and sky illumination maps.
> - **(3)** **Enabled object physical parameters** through rich physical-attribute annotations.
>
> Building on these innovations, UrbanVerse further unlocks several capabilities that prior simulators do not support:
>
> - **(1)** Broader embodied-task coverage, including mobile manipulation via physics-enabled assets, and semantics-heavy tasks such as open-vocabulary perception and embodied VQA, enabled by rich category coverage and annotations (descriptions, affordances, etc.).
> - **(2)** Support for a wider range of robot types, including mobile manipulators and heterogeneous multi-robot settings.
> - **(3)** Additional environmental disturbance factors beyond dynamic agents, such as variable illumination directions, color shifts, and ground-appearance variations.
>
> In addition, users can automatically annotate and add new assets using the UrbanVerse-100K annotation toolkit, or generate new scene layouts using the UrbanVerse-Gen pipeline, **all without manual authoring**.
>
> ---
>
> **`Table R2`**: Function comparison of UrbanVerse with existing urban simulation platforms
>
> | **Feature** | **CARLA** | **MetaUrban** | **UrbanSim** | **UrbanVerse (Ours)** |
> |-------------|-----------|---------------|--------------|------------------------|
> | **Physics Engine** | Unreal 5 | Panda3D | IsaacSim | IsaacSim |
> | **Scene Generation Mechanism** | Hand-crafted | Procedural generation | Procedural generation | **Automatic real-to-sim, data-driven** |
> | **Scene Layout Realism** | Realistic | Unrealistic | Unrealistic | **Realistic** |
> | **Scene Layout Diversity** | 15 fixed scenes | 7 fixed templates | 6 fixed templates | **160 built-in scenes + unlimited (+∞) user-generated scenes** |
> | **# Object Categories** | 106 | 39 | 39 | **667** |
> | **# Object Assets** | 935 | 10,000 | 15,000 | **102,530** |
> | **# Ground Materials** | 30 | 5 | 8 | **288** |
> | **# Sky Maps** | 5 | 1 | 1 | **306** |
> | **Physical Parameters Enabled** | ✔️ | ✘ | ✘ | ✔️ |
> | **Supported Embodied Tasks** | Autonomous driving | Sidewalk navigation | Autonomous driving, Sidewalk navigation, Social navigation | Autonomous driving, Sidewalk navigation, Social navigation, **Mobile Manipulation**, **Open-vocabulary Perception**, **Embodied VQA** |
> | **Supported Robot Types** | 1 type (vehicle) | 1 type (wheeled robot) | 10 types (wheeled/quadruped/wheeled-legged/humanoid) | **20 types** (wheeled/quadruped/wheeled-legged/humanoid/**mobile manipulators**) |
> | **Environmental Dynamic Disturbances** | Weather, Dynamic objects | Dynamic objects | Dynamic objects | Dynamic objects, **sky backgrounds**, **illumination directions and colors**, **ground appearances** |
> | **Multi-agent Interaction Support** | ✘ | ✔️ | ✔️ | **✔️** |
> | **Rendering Efficiency (RGB, L40S GPU)** | ~13 FPS | ~52 FPS | ~94 FPS | **~94 FPS** |
> | **Supported Training Paradigms** | RL, IL, VLA | RL, IL | RL, IL | **RL, IL, VLA** |
> | **Adding New Assets** | Manual | Manual | Manual | **Automatic via our open-source code/API** |
> | **Adding New Scene Layouts** | Manual | Manual | Manual | **Automatic via our open-source code/API** |

---

> ### Author Response · Authors · 2025-11-25
> **Author Response to Reviewer ND3o (5/6)**
>
> > **Question 4:** The paper does not convincingly articulate why researchers should adopt UrbanVerse over these established alternatives beyond improved visual realism. Without demonstrating substantial functional advantages—such as superior sim-to-real transfer, computational efficiency, or unique interaction capabilities—the contribution appears incremental. The claim that UrbanVerse addresses limitations of procedural generation is weakened by the fact that existing simulators already achieve effective sim-to-real transfer for navigation tasks.
>
>
> **Answer 4:** In the manuscript, we discussed UrbanVerse’s core advantages over existing simulators, namely its *improved scene diversity and layout realism*, in Section 1 and Section 2, demonstrated its superior sim-to-real performance over procedural baselines (UrbanSim) across 16 real-world scenarios in **`Table 4`**, and now provided a detailed comparison with related simulators in **`Table 1`**. To address your concern directly, we further clarify and summarize below the key functional advantages of UrbanVerse over existing alternatives:
>
> **(1) Superior scene layout fidelity and realism at scale:** The core advantage of UrbanVerse over existing simulators is its improved scene **layout fidelity** and realism at scale—specifically, accurately reflecting what objects appear in real streets and where they are placed. Such realistic layout distributions are essential for robust and generalizable robot training in urban environments.
>
> Simulators based on hand-crafted scenes (e.g., CARLA) are not scalable (15 scenes in total), while procedural-generation (PG) simulators produce template-driven layouts that deviate substantially from real-world distributions. UrbanVerse addresses these limitations with a novel, **automatic, data-driven** scene generation pipeline that reconstructs metric-accurate layouts directly from real-world city-tour videos, faithfully capturing object distributions and spatial layouts. This enables both high layout realism and scalability.
>
> To further illustrate the gap between UrbanVerse and PG-based simulators in layout fidelity, we provide a side-by-side walkthrough comparison between UrbanVerse and UrbanSim (PG) scenes: **[Click Here: Scene Comparison (UrbanVerse vs. UrbanSim) (anonymous)](https://anonymoususeruseanonymousname.github.io/scene_comparison.html)**. As shown in the comparison, PG scenes typically place objects along sidewalks or roads at fixed intervals dictated by templates, which fails to reflect the variability and complexity observed in real-world streets.
>
>
> **(2) Scaled asset diversity:** UrbanVerse-100K provides **102,530 physics-ready assets across 667 categories**, substantially exceeding existing alternative simulators, such as CARLA (106 categories, 935 assets), MetaUrban (39 categories, 10k assets), and UrbanSim (39 categories, 15k assets). This scale and category breadth offer far better coverage of real-world object diversity, improving robustness in downstream navigation and interaction tasks.
>
> **(3) Enabled physically plausible interactivity:** Unlike existing simulators where most objects are static props, every UrbanVerse asset includes semantic labels, physical parameters, and affordances. As a result, objects respond physically plausibly to robot actions (e.g., a garbage bag rolling when pushed) and enable tasks beyond navigation, such as the mobile manipulation examples demonstrated in the previous response.
>
> **(4) Significantly improved sim-to-real transfer performance:** As demonstrated in our paper (**`Table`** 4, Section 4.3) and also noted in your comments *“convincing evidence of UrbanVerse’s excellent sim-to-real transfer”*, UrbanVerse provides substantially stronger sim-to-real performance than PG-based simulators. Across 16 tested real-world scenarios, UrbanVerse improves success rates by up to **+70.9%** over policies trained in UrbanSim PG scenes, driven by its higher scene diversity and more realistic layout distributions that better bridge the sim-to-real gap. To further illustrate this, we provide a side-by-side real-world sim-to-real transfer navigation experiments comparison between policies trained in UrbanVerse and UrbanSim (PG): **[Click Here: Real-World Navigation Comparison (anonymous)](https://anonymoususeruseanonymousname.github.io/realworld_sidebyside.html)**.
>
> Moreover, as shown in our experiments, existing *procedural-generation simulators* such as UrbanSim *fail* to scale-up policy performance (**`Figure 9 (a)`**, Section 4.2) and *cannot* achieve effective real-world transfer (**`Table 4`**, Section 4.3), attaining **only an 18.8%** success rate (SR) in navigation.
>
> *Given the advantages outlined above and the limitations that UrbanVerse directly addresses, we believe UrbanVerse's contribution is substantially more than incremental.*

---

> ### Author Response · Authors · 2025-11-25
> **Author Response to Reviewer ND3o (6/6)**
>
> > **Question 5:** Please describe the user-side pipeline of UrbanVerse. Based on the current content in the article, it is difficult to determine UrbanVerse's potential contributions to the community.
>
> **Answer 5:**
> Following your suggestion, we now provide and illustrate a detailed user-side pipeline of UrbanVerse in both our documentation website: **[UrbanVerse User-side Pipeline (anonymous)](https://anonymousrepohasanonymousname.github.io/source/user_side_pipeline/index.html)**, and *have included it in* ***`Appendix B (Figure 12)`*** *of the revised paper.*
>
> At a high level, UrbanVerse provides a complete end-to-end workflow:
> - **(1) Generate your own scenes:** Users can input YouTube city-walk videos, phone-recorded clips, or existing datasets into UrbanVerse, and use the UrbanVerse-Gen APIs to automatically generate and export fully interactive simulation scenes in IsaacSim.
> - **(2) Use built-in repositories:** Users can directly load our ready-to-use repositories, including **160** real-to-sim UrbanVerse scenes and **10** artist-crafted CraftBench scenes.
> - **(3) Run downstream tasks:** All scenes support reinforcement learning, imitation learning, multimodal dataset collection, closed-loop evaluation, and zero-shot sim-to-real deployment through our open-sourced APIs.
>
> This unified pipeline makes UrbanVerse easy to adopt, either as a plug-and-play simulation environment or as a scalable real-to-sim scene generation toolkit for the community.
>
> ------------------------------------------------------------------------------------
>
> *We hope our responses have clarified UrbanVerse’s interactive features, dynamic agent support, its functional and real-to-sim transfer advantages over existing simulators, and our comprehensive user-side pipeline and documentation. If these answers address your concerns, we would greatly appreciate it if you could consider updating your score. Thank you!*

---

### Official Review · Reviewer_9ozp · 2025-11-01

**Soundness:** 3
**Presentation:** 3
**Contribution:** 3
**Rating:** 6
**Confidence:** 5

**Summary:**

The paper introduces UrbanVerse, a real-to-sim pipeline that converts crowd-sourced city-tour videos into physics-aware, interactive urban scenes in Isaac Sim. It builds UrbanVerse-100K (100k+ metric-scaled, attribute-annotated assets) and UrbanVerse-Gen to lift semantics, layout, ground, and sky from uncalibrated videos, producing 160 training scenes plus a 20-scene benchmark; KITTI-360 reconstruction metrics, human studies, and scaling experiments show improved scene realism and power-law gains in mapless navigation. Policies trained on these scenes outperform baselines in simulation and zero-shot real-world tests (up to ~90% success on a quadruped) and complete a 337 m route with two interventions.

**Strengths:**

1. The work shows extensive efforts for data curation to ensure a diverse and realistic dataset.
2. The paper shows human evaluation on scene quality. This is the most reliable comparison in the absense of a ground truth scene layout. The result shows scenes in UrbanVerse generally have better quality than UrbanSim.
3. Real world deployment experiments shows good sim-to-real performance of the RL-trained models on the UrbanVerse environment.

**Weaknesses:**

1. The work develops digital cousin from online videos, but does not provide valid reasons why previous 3DGS-based digital twin methods fail for the purpose. This motivation of the work is not fully consolidated.
2. The data curation pipeline utilized LLM to annotate the physical attributes of the objects. However, this is not verified by human and prone to error. The authors should show evidence that these annotations are correct and convincing.
3. The groud fitting step assumes a single ground plane. This fails to model uneven surfaces and curbs that are vital for the success of sidewalk navigation. These sides walk features can also be different for wheeled or legged robots.
4. The number of the scenes in the dataset is limited to 160. This is quite small considering the large number of city-touring videos available online. The limitation can be verified in Fig. 7 where the navigation performance grows linearly with the nubmer of digital cousins. It is unknown where is the margin for the scaling law on the number of environments.

**Questions:**

In Tab. 1, why VGGT has lower quality reconstruction than MASt3R? This seems contridictory to common knowledge that VGGT generally outperformes MASt3R in different scenes.

---

> ### Author Response · Authors · 2025-11-25
> **Author Response to Reviewer 9ozp (1/4)**
>
> Thank you for carefully reviewing our submission and for your constructive feedback. We appreciate your positive comments regarding our “extensive efforts for data curation to ensure a diverse and realistic dataset”, our human scene-quality evaluation showing that UrbanVerse scenes “have better quality than UrbanSim”, and the “good sim-to-real performance” demonstrated in our real-world deployment experiments.
>
> Below, we address your concerns and incorporate your suggestions point by point, and would be grateful if you could let us know whether our responses satisfactorily resolve them.
>
> ------------------------------------
>
> > **Question 1:** The work develops digital cousin from online videos, but does not provide valid reasons why previous 3DGS-based digital twin methods fail for the purpose. This motivation of the work is not fully consolidated.
>
> **Answer 1:** Although we discussed the shortcomings of 3DGS-based approaches (e.g., Vid2Sim [1], OmniRe [2]) in `Section 2 (Lines 143–145)`, we agree that the motivation can be made clearer. To clairfy and address this, we provide and have added:
> - **`Table R1`** below -- A detailed feature comparison between UrbanVerse and 3DGS-based digital twin methods.
> - A more detailed discussion comparing 3DGS-based approaches and UrbanVerse in **`Section 2 (Lines#145–148)`** of the revised *main* paper.
>
> *Note: Here, the 3DGS-based approaches we compare refer specifically to those that use RGB-only inputs to ensure a fair comparison*
>
> Next, we discuss how these differences motivate the design of UrbanVerse and why 3DGS alone is insufficient for interactive, physics-grounded urban simulation. In brief, as summarized in **`Table R1`** below, 3DGS-based digital twins cannot create fully interactive, physics-ready *digital cousin* simulation scenes from city-tour videos due to several key limitations:
>
> 1. **One-to-one reconstruction only:** 3DGS produces a single digital twin per video. It cannot generate multiple diversified “digital cousins’’ from one video, whereas UrbanVerse supports one-to-many scene generation.
> 2. **No object-level decomposition:** 3DGS outputs a single fused radiance field, not individual objects. Without object instances, it is impossible to edit objects or randomize object appearances. UrbanVerse provides full object-level simulations that support all of these operations.
> 3. **No semantic or physical attributes:** 3DGS digital-twin meshes have no category labels or physical properties. This prevents physics-based interaction, such as realistic collision behavior. UrbanVerse assigns all such attributes to every object using its rich annotation.
> 4. **Only a single “good-path’’ trajectory:** 3DGS scenes only support replaying the original camera trajectory for robot learning. They cannot support multiple trajectories (e.g., going backwards) because the geometry is incomplete. UrbanVerse supports unlimited trajectory sampling for robot exploration or multi-agent interaction.
> 5. **No environmental diversification:** 3DGS scenes are fully static and cannot change lighting, sky, asset appearance, or ground appearance. UrbanVerse can vary all these factors to improve policy generalization.
> 6. **Limited multimodal outputs:** 3DGS mainly provides RGB/depth/normal rendering. UrbanVerse also provides semantics, LiDAR, and VQA pairs for downstream training.
>
> Overall, while 3DGS methods are excellent for photorealistic reconstruction, they lack the object structure, physics, and environmental diversification required for interactive urban simulation. UrbanVerse is designed to fill this gap, and we clarify this motivation more clearly in the revised manuscript.
>
> ------
>
> **`Table R1`:** Comparison between 3DGS-based Digital Twins and UrbanVerse
>
> | **Feature** | **3DGS Digital Twin** | **UrbanVerse (Ours)** |
> |-------------|------------------------|-------------------------|
> | **Real-to-Sim Paradigm** | One-to-One | **One-to-Many** |
> | **Object-Level Decomposition** | ✘ | ✔️ |
> | **Interactivity (Assets, Physics, Agents)** | ✘ | ✔️ |
> | **Geometry Completeness** | ✘ (incomplete) | **✔️ (complete meshes)** |
> | **Trajectory Sampling per Scene** | 1 trajectory (good-path only) | **+∞ sampled trajectories** |
> | **Semantics Enabled** | ✘ | ✔️ |
> | **Physical Parameters (Mass, Friction)** | ✘ | ✔️ |
> | **Scene Editing & Diversification** | ✘ | ✔️ |
> | **Physically-Based Lighting** | ✘ | ✔️ |
> | **Reflective Materials** | ✘ | ✔️ |
> | **Environmental Disturbances** | None | **Cousin assets, sky background, illumination, ground appearance** |
> | **Data Collection Modalities** | RGB, Depth, Normal | RGB, Depth, Normal, **Semantics, LiDAR, VQA Pairs** |
>
> ------
>
> [1] Xie, Ziyang, et al. Vid2sim: Realistic and interactive simulation from video for urban navigation. In CVPR, 2025.
>
> [2] Chen, Ziyu, et al. Omnire: Omni urban scene reconstruction. In ICLR, 2025.

---

> ### Author Response · Authors · 2025-11-25
> **Author Response to Reviewer 9ozp (2/4)**
>
> > **Question 2:** The data curation pipeline utilized LLM to annotate the physical attributes of the objects. However, this is not verified by human and prone to error. The authors should show evidence that these annotations are correct and convincing.
>
>
> **Answer 2:** We agree that validating LLM-based physical annotations is important. Direct human verification, however, is *infeasible* because even human annotators cannot accurately provide true dimensions or mass for objects such as cars, cones, or machines without expert knowledge. Prior simulators (e.g., MetaUrban, UrbanSim) rely mainly on visual resizing and relative-scale checks during their manual annotation. Following this anchor-based visual calibration practice, our original submission included qualitative validation by placing hundreds of assets together (`Figure 2` in the main paper). *Here, we further provide an walkthrough video of hundreds of our annotated UrbanVerse-1K assets placed one scene to illustrate their realistic, physically plausible relative scales:* **[Click Here: UrbanVerse-1K Object Gallery Walkthrough (anonymous)](https://anonymoususeruseanonymousname.github.io/urbanverse_1k.html)**.
>
> To further address your concern, in **`Table R2`** below (in next response post), we did our best efforts to conduct a quantitative evaluation on 17 categories comprising 1,335 objects for which reliable ground-truth specifications or commonly agreed-upon real-world dimensions are publicly available (e.g., Tesla Cybertruck, vending machines, traffic cones, laptops). For each category, we compared our annotations against collected ground truth using Mean Absolute Percentage Error (MAPE), denoted as $\text{MAPE} = \frac{100\%}{N} \sum_{i=1}^{N} \left| \frac{\text{Annotation}_i - \text{GT}_i}{\text{GT}_i} \right|$, evaluated over height, length, width, and mass. *We have also included this quantative evaluation in* ***`Appendix C.4`*** *of the revised paper.*
>
> As shown in **`Table R2`** below (in next response post), our geometric annotations (height/length/width) are highly accurate, typically within **1–8% MAPE**, and often around **1–3%** for rigid objects such as cars, hydrants, and balls. Mass is naturally harder to estimate due to material variation, but still remains within a reasonable average error of **19.58%**.
>
> Overall, these results demonstrate that our automatic annotation pipeline produces accurate and reliable physical attributes, making it suitable for large-scale simulation without manual labeling.

---

> ### Author Response · Authors · 2025-11-25
> **Author Response to Reviewer 9ozp (3/4)**
>
> **`Table R2`:** Evaluation of annotated physical attributes. We report the MAPE and standard deviation against ground-truth attribute values across 17 object categories for Height (MAPE-H), Length (MAPE-L), Width (MAPE-W), and Mass (MAPE-M).
>
> | **Category**                 | **# Object** | **MAPE-H (%)**        | **MAPE-L (%)**        | **MAPE-W (%)**        | **MAPE-M (%)**         |
> |------------------------------|--------------|------------------------|------------------------|------------------------|-------------------------|
> | **Average**                  | --           | **5.88 ± 3.85**        | **6.14 ± 3.86**        | **7.00 ± 4.11**        | **19.58 ± 12.11**       |
> | Lamborghini Huracan STO      | 12           | 0.49 ± 0.30            | 0.66 ± 0.40            | 0.51 ± 0.30            | 2.02 ± 1.30             |
> | McLaren 600LT Spider         | 4            | 0.59 ± 0.40            | 0.78 ± 0.50            | 0.62 ± 0.40            | 2.47 ± 1.50             |
> | Tesla Cybertruck             | 7            | 1.97 ± 1.40            | 2.50 ± 1.70            | 2.02 ± 1.40            | 9.05 ± 6.50             |
> | Land Rover Defender          | 2            | 2.99 ± 2.00            | 3.72 ± 2.80            | 1.99 ± 1.40            | 12.56 ± 7.50            |
> | Electric Scooter             | 68           | 5.97 ± 4.00            | 9.38 ± 7.00            | 12.32 ± 6.00           | 29.96 ± 19.00           |
> | Bicycle                      | 118          | 7.03 ± 5.00            | 5.05 ± 3.00            | 7.75 ± 5.00            | 17.78 ± 13.00           |
> | Vending Machine              | 153          | 7.94 ± 5.00            | 9.00 ± 5.00            | 7.18 ± 5.00            | 19.60 ± 10.00           |
> | Street Cabinet               | 43           | 20.00 ± 13.00          | 23.00 ± 13.00          | 22.00 ± 12.00          | 14.86 ± 10.00           |
> | Parking Meter                | 16           | 7.94 ± 5.00            | 10.71 ± 7.00           | 6.56 ± 5.00            | 22.94 ± 14.00           |
> | Fire Hydrant                 | 160          | 1.47 ± 1.00            | 1.53 ± 1.00            | 1.53 ± 1.00            | 7.14 ± 4.00             |
> | Traffic Cone                 | 126          | 12.57 ± 9.00           | 20.40 ± 13.00          | 20.40 ± 13.00          | 51.07 ± 25.00           |
> | Jersey Barrier               | 95           | 5.06 ± 3.00            | 7.07 ± 5.00            | 25.83 ± 13.00          | 75.00 ± 50.00           |
> | Egg                          | 188          | 1.09 ± 0.70            | 1.03 ± 0.70            | 1.03 ± 0.70            | 12.45 ± 8.00            |
> | Cigarette                    | 28           | 1.00 ± 0.60            | 1.00 ± 0.60            | 1.00 ± 0.60            | 15.10 ± 9.00            |
> | Laptop                       | 171          | 20.46 ± 13.00          | 5.06 ± 3.00            | 4.84 ± 3.00            | 25.02 ± 17.00           |
> | Football                     | 99           | 1.50 ± 1.00            | 1.50 ± 1.00            | 1.50 ± 1.00            | 6.83 ± 4.00             |
> | Basketball                   | 45           | 1.92 ± 1.00            | 1.92 ± 1.00            | 1.92 ± 1.00            | 8.97 ± 6.00             |

---

> ### Author Response · Authors · 2025-11-25
> **Author Response to Reviewer 9ozp (4/4)**
>
> > **Question 3:** The ground fitting step assumes a single ground plane. This fails to model uneven surfaces and curbs that are vital for the success of sidewalk navigation. These sides walk features can also be different for wheeled or legged robots.
>
> **Answer 3:** Actually, UrbanVerse does **not assume a single** ground plane. Instead, UrbanVerse **explicitly model two types** of ground planes (as described in `Line#299-301` of the original submission): a **road plane** and a **sidewalk plane**. These planes are fitted **separately**, and when a sidewalk is present, **the sidewalk plane is elevated by 15 cm relative to the road plane** to realistically model curb height. We invite the reviewer to view our scene walkthrough demo, which shows the modeled curbs: **[Click Here: UrbanVerse Scene Walkthrough (anonymous)](https://anonymoususeruseanonymousname.github.io/urbanverse_scenes.html)**
>
> Indeed, as you noted, curb modeling is indeed important for urban sidewalk navigation. In our real-world experiments, both the wheeled and legged robots *successfully traverse curbs and move between street and sidewalk* using our PPO-UrbanVerse policy, thanks to this explicit curb modeling. We provide two real-world experiment recording videos here, one with a wheeled robot and one with a legged robot, **showing successful curb traversal:** **[Click Here: Real-World Street-Crossing with Curbs (anonymous)](https://anonymoususeruseanonymousname.github.io/realworld_cross_curbs.html)**
>
> ------------------------------------
>
> > **Question 4:** The number of the scenes in the dataset is limited to 160. It is unknown where is the margin for the scaling law on the number of environments.
>
> **Answer 4:** We are actually continuously expanding UrbanVerse scenes for this project. Following your suggestion, *we have already expanded the dataset by generating another 160 new UrbanVerse scenes* from 32 new city-tour videos (32 new layouts, each with 5 digital cousins). Together with the original 160 scenes, we now have *320* scenes built from 64 unique layouts. We are currently training PPO policies on this expanded set to examine how performance scales. The experiment is running and requires additional time due to server queueing. We will include results later next week and will update them as soon as they are available.
>
> *Update - The experiment is now complete:* As shown in **`Table R3`**, performance continues to increase when doubling the number of environments, with *no clear indication of saturation* at this scale. These results suggest that the policy continues to benefit from additional environments. UrbanVerse is an ongoing long-term project, and we will keep expanding and scaling its scene repository to more thoroughly explore where the performance curve begins to plateau.
>
> **`Table R3`**: Scaling the PPO navigation policy with additional **UrbanVerse** scenes (160 → 320). We report performance tested on CraftBench.
>
> | # Training UrbanVerse Scenes                | SR ↑ | CT ↓ | RC ↑ |
> |---------------------------------------------|------|------|------|
> | 160 Scenes (32 layouts × 5 cousins)         | 41.9 | 35.5 | 62.4 |
> | 320 Scenes (64 layouts × 5 cousins)         | **47.4** | **22.1** | **71.9** |
>
> ------------------------------------
>
> > **Question 5:** In Tab. 1, why VGGT has lower quality reconstruction than MASt3R? This seems contradictory to common knowledge that VGGT generally outperforms MASt3R in different scenes.
>
> **Answer 5:** Great question. The main reason is that VGGT is a pure feedforward model and does *not perform any global post-optimization to make its depth consistent with camera poses.* As a result, VGGT is fast and produces good *relative* depth in prior literature, *but the depth is often not well aligned with the estimated camera poses in a consistent metric scale across the entire video sequence.*
>
> MASt3R, on the other hand, uses *a sparse global optimization step that refines depth and ensures cross-frame consistency with camera poses*. This makes its depth better aligned with the camera poses throughout the entire sequence. Recent work ViPE [3] also discusses this in the first paragprah of Section 3.3, highlighting that post-optimization is essential for video-based 3D geometric perception.
>
> In our work, this global pose–depth consistency is especially important for UrbanVerse-Gen, since accurate 3D layout extraction and object parsing rely on both precise depth and consistent alignment across all frames. Better global alignment directly improves 3D lifting and multi-frame object instance fusion, which explains why MASt3R performs better than VGGT in our evaluation on the KITTI-360 dataset.
>
> [3] Huang, Jiahui, et al. *ViPE: Video Pose Engine for 3D Geometric Perception.* NVIDIA Research Whitepapers, 2025.
>
> ------------------------------------
>
> *We hope our responses have addressed your questions and concerns. If so, we would sincerely appreciate your consideration in updating the score. Thank you!*

---

### Official Review · Reviewer_Nwq7 · 2025-11-02

**Soundness:** 3
**Presentation:** 3
**Contribution:** 3
**Rating:** 6
**Confidence:** 3

**Summary:**

The paper introduces UrbanVerse, a data-driven system that automatically converts real-world city-tour videos into interactive, physics-aware urban simulations. The framework comprises:

- UrbanVerse-100K, a large-scale annotated database of 100k+ metric-scale 3D urban assets with semantic and physical attributes.

- UrbanVerse-Gen, an automated pipeline that reconstructs scene layouts and lighting from uncalibrated videos and instantiates them into simulation-ready environments in Isaac Sim.

The authors demonstrate that policies trained on these automatically generated scenes follow scaling laws and transfer effectively to real-world navigation tasks—achieving up to +30% zero-shot sim-to-real improvement over prior baselines.

**Strengths:**

- Novel real-to-sim paradigm: The approach of transforming crowd-sourced city-tour videos into physically interactive simulations is innovative and highly scalable.

- Large, well-annotated asset database: The UrbanVerse-100K dataset provides a valuable and reusable resource with semantic and physical labels, addressing a major limitation of existing simulators.

- Strong quantitative and qualitative results: The system shows high reconstruction fidelity (93% semantic accuracy, 1.4 m localization error) and impressive sim-to-real performance across multiple robot platforms.

- Comprehensive experiments: The paper evaluates reconstruction quality, scaling effects (power-law validation), and real-world zero-shot transfer, providing a well-rounded empirical validation.

- Open-sourcing commitment: Promising for community impact, given the open release of code, assets, and simulation scenes.

**Weaknesses:**

- **Overly complex pipeline with high compounding error:**
    The overall data annotation and reconstruction pipeline involves numerous components—open-vocabulary detection, depth estimation, 3D lifting, GPT-4 labeling, geometry matching, appearance retrieval, and simulation instantiation. Each step introduces noise, and the compounded uncertainty across so many modules raises serious concerns about consistency and accuracy. While the qualitative examples look convincing, they may reflect cherry-picked successes rather than representative results at scale.

- **Lack of failure case analysis:**
Following the previous, the paper only shows successful examples but omits clear failure cases or error breakdowns. Understanding when and why the system fails—e.g., mis-segmentation, inaccurate depth estimation, wrong asset retrieval—would be critical to assessing robustness and diagnosing limitations. Including a few representative failure visualizations would make the results more transparent and credible.

- **Questionable large-scale data quality:**
    The pipeline heavily relies on noisy, crowd-sourced internet videos, often with unstable camera motion, motion blur, or occlusions. Given such low-quality inputs and a long chain of automated processing, it is doubtful that most of the reconstructed scenes achieve the same level of fidelity as the few presented cases. The paper lacks quantitative or statistical analysis of **data noise, annotation error rates, or reconstruction failure modes**, which undermines confidence in the dataset’s overall reliability.

- **Limited scope of validation:**
    Despite the title’s emphasis on “urban simulation,” the evaluation focuses only on **navigation** tasks, without exploring manipulation, traffic reasoning, or social interaction scenarios. It remains unclear whether UrbanVerse scenes are truly general-purpose or mainly tuned for simple navigation.

- **System integration over algorithmic innovation:**
    The paper’s novelty lies mainly in combining existing large vision models (CLIP, SAM2, GPT-4.1, MASt3R, etc.) into a pipeline, rather than introducing new algorithms or learning frameworks. The contribution is thus primarily engineering-based.

- **Insufficient runtime and scalability analysis:**
    The pipeline’s cost—both in compute and manual verification—is not clearly reported. Given the dependence on heavy models and large-scale video inputs, the claimed scalability may be impractical without significant resources.

**Questions:**

- Can UrbanVerse handle dynamic agents (e.g., pedestrians, vehicles in motion), or is it restricted to static geometry?

- How does scene reconstruction cope with occlusions and low-light conditions in real videos?

- Would the system generalize to non-street urban contexts (e.g., parks, campuses, or indoor–outdoor transitions)?

---

> ### Author Response · Authors · 2025-11-25
> **Author Response to Reviewer Nwq7 (1/6)**
>
> Thank you for reviewing our submission and for your constructive feedback. We appreciate your positive comments on UrbanVerse’s scalable real-to-sim paradigm, the value of our UrbanVerse-100K asset database, the strong scene generation quality and sim-to-real results across multiple robots, and our plan to open-source the entire system.
>
> Below, we address your concerns and incorporate your suggestions point by point, and we would be grateful if you could let us know whether our responses satisfactorily resolve them.
>
> ------------------------------------
>
> > **Question 1:** Overly complex pipeline with high compounding error. The overall data annotation and reconstruction pipeline involves numerous components—open-vocabulary detection, depth estimation, 3D lifting, GPT-4 labeling, geometry matching, appearance retrieval, and simulation instantiation. Each step introduces noise, and the compounded uncertainty across so many modules raises serious concerns about consistency and accuracy. While the qualitative examples look convincing, they may reflect cherry-picked successes rather than representative results at scale.
>
> **Answer 1:** Thank you for raising this point. To address your concern about larger-scale qualitative evidence and to demonstrate that our results are not cherry-picked, we provide:
> - **(1)** detailed walkthrough demos of **24 additional simulation scenes** sampled from the full set of 160 UrbanVerse-generated scenes: **[Click Here: More UrbanVerse Scene Examples (anonymous)](https://anonymoususeruseanonymousname.github.io/urbanverse_scenes.html)**
> - **(2)** a substantially larger collection of **70 qualitative scene examples** in **`Appendix D.1 (Figure 20)`** of the revised paper.
>
> We included only a few examples in the original submission due to space constraints, and we are happy to provide further examples upon request.
>
> Regarding the concern about compounding errors across modules, **`Table 2`** in the original manuscript presents a comprehensive quantitative evaluation covering all major stages of our UrbanVerse-Gen pipeline:
>
> - **Category Recovery (Cat.)** and **mAP25** measure GPT-4 labeling and open-vocabulary detection accuracy.
> - **Distance (Dist.)**, **Orientation (Ori.)**, and **Scale** measure 3D lifting accuracy (via estimated depth and poses) for layout extraction.
> - **Asset Recovery (Ast.)** evaluates geometry matching and appearance retrieval.
>
> Across these six metrics, UrbanVerse-Gen maintains strong accuracy and consistency: **93.1%** of object categories are correctly recovered; reconstructed objects have only **1.4 m** position deviation, **19.8°** orientation error, and **0.8 m³** scale error; and asset retrieval achieves **75.1%** accuracy. These results show that errors do not accumulate in a way that harms overall scene quality.
>
> Finally, the strong real-world sim-to-real transfer results in **`Table 4`** provide end-to-end validation: policies trained on our automatically generated scenes transfer reliably to the real world, indicating that the generated environments are sufficiently accurate and effective for robot learning.
>
> ------------------------------------
>
> > **Question 2:** Can UrbanVerse handle dynamic agents (e.g., pedestrians, vehicles in motion), or is it restricted to static geometry?
>
> **Answer 2:** Yes. UrbanVerse *fully supports* dynamic agents. To directly address and clarify this concern, we provide and have added:
> - **[Click Here: UrbanVerse Scenes Populated with Dynamic Agents (anonymous)](https://anonymoususeruseanonymousname.github.io/scene_with_dynas.html)** -- Video demos of two UrbanVerse scenes populated with diverse dynamic agents, including multiple robots (wheeled, quadruped, humanoid), moving pedestrians, cars, wheelchair users, and e-scooter riders: .
> -  A method description of Interactive Dynamic-agent Population in the revised *main* paper (**`Line#307–313`**) and visualized dynamic-agent scenes in **`Figure 5`**.
>
> UrbanVerse uses a **GPU-accelerated multi-agent motion planner** based on ORCA [1] algorithm to populate scenes with agents that move realistically and interact with both the environment and the robot during training and testing. Specifically, for any UrbanVerse scene, we generate a 2D occupancy map from object geometry and semantics, sample start–goal pairs for each agent, and compute initial collision-free paths using ORCA. During simulation, agents continuously adjust their velocities based on nearby obstacles and nearby agents, producing smooth, collision-aware motion. This mechanism is scene-agnostic and supports diverse moving agents across all generated environments.
>
> [1] Van Den Berg, J., Guy, S. J., Lin, M., & Manocha, D. *Reciprocal n-body collision avoidance*. In Robotics Research: The 14th International Symposium (ISRR), 2011.

---

> ### Author Response · Authors · 2025-11-25
> **Author Response to Reviewer Nwq7 (2/6)**
>
> > **Question 3:** Limited scope of validation. Despite the title’s emphasis on “urban simulation,” the evaluation focuses only on navigation tasks, without exploring manipulation, traffic reasoning, or social interaction scenarios. It remains unclear whether UrbanVerse scenes are truly general-purpose or mainly tuned for simple navigation.
>
> **Answer 3:** UrbanVerse is *not* limited to navigation task. To demonstrate this and address this concern, we provide and have added:
> - **[Click Here: Mobile Manipulation in UrbanVerse Scenes (anonymous)](https://anonymoususeruseanonymousname.github.io/mobile_manipulator.html)** -- Two mobile manipulation task demo sthat include *(1)* a wheeled delivery robot equipped with a Franka Panda arm, and *(2)* a Boston Dynamics Spot quadruped with an arm, both operating in UrbanVerse scenes.
> - A description of the tasks supported by UrbanVerse in the revised *main* paper (**`Line#334–339`**) and visualized example tasks in **`Figure 6`**.
>
> In the paper, we chose urban navigation as the main case study because it most clearly highlights the strengths of UrbanVerse's realistic layout diversity and fidelity, realistic asset distributions, and scalable real-to-sim generation. However, UrbanVerse is designed to support much broader embodied tasks. All assets come with semantic labels, physical parameters (mass, friction), and affordance tags, and the simulator supports dynamic agents and multi-agent interaction. This enables tasks beyond simple navigation, such as semantic-goal navigation, object search, social interaction, and mobile manipulation. Looking forward in the future, we are also planning to add a lightweight abstraction layer to better support traffic-reasoning scenarios since UrbanVerse already naturally supports autonomous-driving environments.
>
> ------------------------------------
>
> > **Question 4:** Questionable large-scale data quality.
>
> **Answer 4:** Thank you for raising this concern. To show that our results are not limited to a few present examples, in Answer 1 above, we have provided walkthrough demos of 24 additional simulation scenes sampled from the full set of 160 UrbanVerse scenes here (**[Click Here: More UrbanVerse Scene Examples (anonymous)](https://anonymoususeruseanonymousname.github.io/urbanverse_scenes.html)** ). To further validate UrbanVerse-100K's assets quality, we provide:
> - **[Click Here: UrbanVerse-1K Object Gallery Walkthrough (anonymous)](https://anonymoususeruseanonymousname.github.io/urbanverse_1k.html)** -- An walkthrough video of hundreds of our annotated UrbanVerse-1K assets placed one scene to illustrate their realistic, physically plausible relative scales.
>
> Beyond qualitative evidence, in **`Table R1`** below (in next response post), we also further conducted a quantitative validation of our assets annotation. We evaluated 17 categories (1,335 objects total) for which reliable ground-truth specifications or widely agreed real-world dimensions are available (e.g., Tesla Cybertruck, vending machines, traffic cones, laptops). For each category, we compared our annotations against collected ground truth using Mean Absolute Percentage Error (MAPE), denoted as $\text{MAPE} = \frac{100\%}{N} \sum_{i=1}^{N} \left| \frac{\text{Annotation}_i - \text{GT}_i}{\text{GT}_i} \right|$, evaluated over height, length, width, and mass. *We have also included this quantative evaluation in* ***`Appendix C.4`*** *of the revised paper.*
>
> As shown in **`Table R1`** below (in next response post), geometric attributes are highly accurate, typically within 1-8% MAPE and often around 1-3% for rigid objects such as cars, hydrants, and balls. Mass estimation is naturally more variable due to material differences, but remains within a reasonable error margin (average 19.58%). These results indicate that our automatic annotation pipeline is also reliable enough to support simulation at scale without manual labeling.

---

> ### Author Response · Authors · 2025-11-25
> **Author Response to Reviewer Nwq7 (3/6)**
>
> **`Table R1`:** Evaluation of annotated physical attributes. We report the MAPE and standard deviation against ground-truth attribute values across 17 object categories for Height (MAPE-H), Length (MAPE-L), Width (MAPE-W), and Mass (MAPE-M).
>
> | **Category**                 | **# Object** | **MAPE-H (%)**        | **MAPE-L (%)**        | **MAPE-W (%)**        | **MAPE-M (%)**         |
> |------------------------------|--------------|------------------------|------------------------|------------------------|-------------------------|
> | **Average**                  | --           | **5.88 ± 3.85**        | **6.14 ± 3.86**        | **7.00 ± 4.11**        | **19.58 ± 12.11**       |
> | Lamborghini Huracan STO      | 12           | 0.49 ± 0.30            | 0.66 ± 0.40            | 0.51 ± 0.30            | 2.02 ± 1.30             |
> | McLaren 600LT Spider         | 4            | 0.59 ± 0.40            | 0.78 ± 0.50            | 0.62 ± 0.40            | 2.47 ± 1.50             |
> | Tesla Cybertruck             | 7            | 1.97 ± 1.40            | 2.50 ± 1.70            | 2.02 ± 1.40            | 9.05 ± 6.50             |
> | Land Rover Defender          | 2            | 2.99 ± 2.00            | 3.72 ± 2.80            | 1.99 ± 1.40            | 12.56 ± 7.50            |
> | Electric Scooter             | 68           | 5.97 ± 4.00            | 9.38 ± 7.00            | 12.32 ± 6.00           | 29.96 ± 19.00           |
> | Bicycle                      | 118          | 7.03 ± 5.00            | 5.05 ± 3.00            | 7.75 ± 5.00            | 17.78 ± 13.00           |
> | Vending Machine              | 153          | 7.94 ± 5.00            | 9.00 ± 5.00            | 7.18 ± 5.00            | 19.60 ± 10.00           |
> | Street Cabinet               | 43           | 20.00 ± 13.00          | 23.00 ± 13.00          | 22.00 ± 12.00          | 14.86 ± 10.00           |
> | Parking Meter                | 16           | 7.94 ± 5.00            | 10.71 ± 7.00           | 6.56 ± 5.00            | 22.94 ± 14.00           |
> | Fire Hydrant                 | 160          | 1.47 ± 1.00            | 1.53 ± 1.00            | 1.53 ± 1.00            | 7.14 ± 4.00             |
> | Traffic Cone                 | 126          | 12.57 ± 9.00           | 20.40 ± 13.00          | 20.40 ± 13.00          | 51.07 ± 25.00           |
> | Jersey Barrier               | 95           | 5.06 ± 3.00            | 7.07 ± 5.00            | 25.83 ± 13.00          | 75.00 ± 50.00           |
> | Egg                          | 188          | 1.09 ± 0.70            | 1.03 ± 0.70            | 1.03 ± 0.70            | 12.45 ± 8.00            |
> | Cigarette                    | 28           | 1.00 ± 0.60            | 1.00 ± 0.60            | 1.00 ± 0.60            | 15.10 ± 9.00            |
> | Laptop                       | 171          | 20.46 ± 13.00          | 5.06 ± 3.00            | 4.84 ± 3.00            | 25.02 ± 17.00           |
> | Football                     | 99           | 1.50 ± 1.00            | 1.50 ± 1.00            | 1.50 ± 1.00            | 6.83 ± 4.00             |
> | Basketball                   | 45           | 1.92 ± 1.00            | 1.92 ± 1.00            | 1.92 ± 1.00            | 8.97 ± 6.00             |

---

> ### Author Response · Authors · 2025-11-25
> **Author Response to Reviewer Nwq7 (4/6)**
>
> > **Question 5:** Insufficient runtime and scalability analysis.
>
> **Answer 5:** Thank you for the helpful suggestion. In response, we now provide a clear and detailed runtime and scalability analysis for both the UrbanVerse-Gen scene generation pipeline and the UrbanVerse-100K annotation process. These results are summarized below in **`Table R2`**, **`Table R3`**, and **`Table R4`**. *We have also included this computational analysis in* ***`Appendix L`*** *of the revised paper.* Next, we detail each analysis below.
>
> **(1) UrbanVerse-Gen Scene Generation Process:** As shown in **`Table R2`**, the computational cost and GPT-4.1 calls scale almost linearly with video length. The dominant cost comes from the 3D reconstruction step (MASt3R). Short clips (10-40 s) require only **4-14 LLM calls** and **12-114 s** of processing, while longer clips (80–180 s) require **27-60 LLM calls** and **289-1135 s** on an NVIDIA H100 GPU. In our experiments, using 180 s city-tour clips, we generate **160 fully interactive scenes in 1.26 hours** on **4× H100 GPUs**, as summarized in **`Table R3`**.
>
> **(2) UrbanVerse-100K Annotation Process:** As shown in **`Table R4`**, the annotation pipeline scales efficiently and predictably: each asset requires exactly **one** GPT-4.1 call, averaging **2.3 s** and **$ 0.013** per object. Annotating the full set of **102,530 assets** takes **65.5 hours** of API wall-clock time and **$ 1,334** in total cost. This demonstrates that our automatic annotation pipeline is both practical and economical, and significantly more efficient than manual annotation.
>
> Overall, these analyses confirm that **UrbanVerse scales well in both computation and cost**, even when applied to large video collections and large asset corpora.
>
> ------
> **`Table R2`:** UrbanVerse-Gen real-to-sim scene generation time with varying input video lengths on an NVIDIA H100 GPU
>
> | **Video Duration (sec)** | **Video Length (# frames)** | **LLM Calls** | **Scene Generation Wall Time (sec)** |
> |--------------------------|-----------------------------|---------------|--------------------------------------|
> | 10                       | 10                          | 4             | 12.38                                |
> | 40                       | 40                          | 14            | 114.46                               |
> | 80                       | 80                          | 27            | 289.80                               |
> | 180                      | 180                         | 60            | 1135.20                              |
>
> ------
>
> **`Table R3`:** Overall computational time of UrbanVerse simulation scenes
>
> | **Setting**                          | **# Input City-tour Videos** | **# Cousin Scenes per Layout** | **# Unique Scene Layouts** | **# Total Simulation Scenes** | **Wall Time** |
> |--------------------------------------|-------------------------------|--------------------------------|-----------------------------|-------------------------------|---------------|
> | Single layout with 5 digital cousins | 1                             | 5                              | 1                           | 5                             | 18.92 min     |
> | 160 Scenes (1 × NVIDIA H100)         | 32                            | 5                              | 32                          | 160                           | 10.08 hrs     |
> | 160 Scenes (4 × NVIDIA H100, default) | 32                           | 5                              | 32                          | 160                           | 1.26 hrs      |
>
> ------
>
> **`Table R4`:** Annotation cost and runtime statistics of the UrbanVerse-100K annotation pipeline
>
> | **# Objects** | **GPT-4.1 Call Counts** | **API Wall Clock Time** | **API Cost** |
> |---------------|--------------------------|---------------------------|--------------|
> | 1             | 1                        | 0.0003 hrs               | $0.018       |
> | 2             | 2                        | 0.0005 hrs               | $0.029       |
> | ...           | ...                      | ...                      | ...          |
> | 102,530       | 102,530                  | 65.5 hrs                 | $1,334       |
> | **Average**   | 1 / object               | 2.3 sec / object         | $0.013 / object |

---

> ### Author Response · Authors · 2025-11-25
> **Author Response to Reviewer Nwq7 (5/6)**
>
> > **Question 6:** System integration over algorithmic innovation. The paper’s novelty lies mainly in combining existing large vision models (CLIP, SAM2, GPT-4.1, MASt3R, etc.) into a pipeline, rather than introducing new algorithms or learning frameworks. The contribution is thus primarily engineering-based.
>
> **Answer 6:** Thank you for the comment. While UrbanVerse does integrate several strong vision and language models, its contribution goes far beyond simple system assembly. The primary goal of this work is to **scale up urban robot learning** and **significantly reduce the sim-to-real gap**. To achieve this, UrbanVerse introduces **four core innovations** that are not addressed by prior simulators such as CARLA, MetaUrban, or UrbanSim:
>
> - **(1) A novel automatic real-to-sim, data-driven scene generation mechanism (UrbanVerse-Gen)** that revolutionizes the prior unrealistic procedural scene generation paradigm, with scenes grounded in real-world layouts and large-scale diversity.
> - **(2) An order-of-magnitude larger, high-quality asset library (UrbanVerse-100K)** with 667 categories and 102,530 instances, along with rich ground materials and sky illumination maps (prior simulators offer up to <106 categories and <15,000 assets).
> - **(3) Full physical interactivity** via rich object-level semantic and physical attributes, enabling physically-plausible simulation and robot–object interaction.
> - **(4) Empirical evidence of strong scaling behavior and generalization**, including power-law improvements and state-of-the-art sim-to-real transfer across robot platforms.
>
> ------------------------------------
>
> > **Question 7:** Would the system generalize to non-street urban contexts (e.g., parks, campuses, or indoor–outdoor transitions)?
>
> **Answer 7:** UrbanVerse is currently designed for street-level urban environment simulations. Generalizing to non-street contexts such as parks, campuses, or indoor–outdoor transitions would require specialized designs, such as complicated terrain modeling. We view this as a natural and exciting future direction, and plan to extend UrbanVerse to broader outdoor settings by incorporating terrain maps and indoor-outdoor transition accesses. *Following your suggestion, we have added a dedicated Limitation section in* ***`Section 5 (Lines 535–539)`*** *of the revised main paper to clearly and transparently discuss these limitations.*
>
> However, interestingly, in the real-world experiments, our PPO-UrbanVerse navigation policy **actually shows strong generalization to non-street scenarios**. During the experiments, we observe our PPO-UrbanVerse policy **successfully operates in campus areas and even handles accessibility ramps and sloped walkways.** We invite the reviewer to view these behaviors in our real-world sim-to-real transfer experiment recordings: **[Click Here: Real-World Demos in Diverse Contexts (anonymous)](https://anonymoususeruseanonymousname.github.io/realworld_sidebyside.html)**.

---

> ### Author Response · Authors · 2025-11-25
> **Author Response to Reviewer Nwq7 (6/6)**
>
> > **Question 8:** Lack of failure case analysis; How does scene reconstruction cope with occlusions and low-light conditions in real videos?
>
> **Answer 8:** Thank you for raising this point. *Following your suggestion, we now include a dedicated Failure Case Analysis section in* ***`Appendix G`*** *of the revised paper, together with a discussion in the Limitation section (* ***`Lines 535–539`*** *) to make our limitation transparent.* Below, we summarize our main error sources and how UrbanVerse handles them.
>
> **(1) Challenging input conditions:** Certain video conditions can degrade 3D reconstruction quality, such as extremely low light, distant shots, heavy clutter, or complex terrain. In our large-scale experiments, we automatically filter out such problematic clips by sending every 10th frame to GPT-4.1 for quality screening. This avoids feeding clearly unrecoverable video clips into the pipeline.
>
> **(2) Depth/pose drift:** Fast motion or unstable handheld cameras may introduce depth and pose noise, occasionally causing misplacement of objects (e.g., a advertising board on the sidewalk might be shifted slightly on the road). While multi-view alignment reduces this, such drift remains our primary failure mode. We plan to incorporate hard constraints between objects and its plausible placement area to further alleviate these errors, using object-area spatial relationship estimation.
>
> **(3) Imperfect asset retrieval:** For visually complex or rare objects, retrieval may return an asset with slightly different appearance. However, our three-stage matching (semantic matching → geometry filtering → appearance selection) is designed to ensure that retrieved assets retain correct category and affordance (semantic matching), and collision geometry (geometry filtering). As a result, even when appearance differs, the physical behavior and reaction remains correct during the simulation, and the visual variation effectively acts as benign domain randomization.
>
> **(4) Mis-segmentation under heavy occlusion:** Severe occlusion by pedestrians or vehicles may lead to incomplete masks. However, our layout extraction remains stable because it aggregates multi-view evidence rather than relying on a single frame.
>
> Despite these potential failure modes, they may not significantly impact downstream policy learning because: *(i)* Multi-view aggregation and strict geometry filtering produce stable scene-level layouts even when some frames are noisy; and *(ii)* Our end-to-end policy performance evaluations (`Table 3`) and real-world sim-to-real transfer results (`Table 4`) show that policies trained in UrbanVerse scenes scales effectively and generalize reliably to the real world. This indicates that the overall scene fidelity of UrbanVerse is sufficient for effective robot training.
>
> ------------------------------------
>
> *We hope our responses have clarified the robustness of our pipeline, the large-scale data quality, UrbanVerse’s support for dynamic agents and broader embodied tasks, and the practical scalability of both scene generation and asset annotation. If our answers satisfactorily address your concerns, we would appreciate your consideration in updating the score. Thank you!*

---

### Author Response · Authors · 2025-11-30
**Overall Summary (3/3): Review and Rebuttal**

---

> **Q5:** Functional comparison with existing simulators (asked by Reviewer `ND3o`)

To address this, we conducted a systematic comparison with CARLA, MetaUrban, and UrbanSim in the rebuttal and the **Table 1** of the *main* revised paper.

---

> **Q6:** Developer Documentation, User-side Pipeline, Community Engagement Strategires, and Open-source Clarity (asked by Reviewer `ND3o`)

To address these questions, we provided all requested materials directly in the rebuttal, summarized below:

- **[Click Here: UrbanVerse Developer Documentation](https://anonymousrepohasanonymousname.github.io)**
- **[Click Here: UrbanVerse User-side Pipeline](https://anonymousrepohasanonymousname.github.io/source/user_side_pipeline/index.html)** (Also added to **Appendix B**)
- **[Click Here: UrbanVerse Community Engagement Guide](https://anonymousrepohasanonymousname.github.io/source/community/index.html)**
- **Detailed Open-Source Plan (see `Table R1` below)**


---


**`Table R1`:** UrbanVerse Open-Source Release Plan

| **Content** | **Platform** | **Format** | **Release Month** |
|-------------|--------------|------------|--------------------|
| UrbanVerse-100K Asset Database | Hugging Face | `.glb`, `.json`, `.jpg`, `.mdl`, `.hdr` | January 2026 |
| 160 UrbanVerse Scenes | Hugging Face | `.usd` | January 2026 |
| CraftBench Scenes | Hugging Face | `.usd` | January 2026 |
| UrbanVerse-Gen Pipeline | GitHub | `.py` | February 2026 |
| RL Training Scripts and Checkpoints | GitHub | `.py`, `.pt` | February 2026 |
| UrbanVerse-100K Annotation Tool | GitHub | `.py` | January 2026 |
| Documentation & Tutorials | GitHub | `.html` | January 2026 |


---

*We again thank the Area Chairs and Reviewers for their time and effort. Finally, we sincerely hope the Area Chairs will consider UrbanVerse’s contributions to the community, along with the detailed rebuttal responses and the improved revised manuscript that addresses all reviewer concerns.*

Best regards,

Authors of Submission #9637

---

### Author Response · Authors · 2025-11-30
**Overall Summary (2/3): Review and Rebuttal**

---

## ***Review Summary, Updates and Improvements from the Rebuttal***

---

**Initial Reviews and Rebuttal Exchanges**: **Our work, UrbanVerse, received *initial* ratings of 8 (`eUF4`), 6 (`Nwq7`), 6 (`9ozp`), and 2 (`ND3o`).** After we posted our detailed rebuttal responses, reviewer `eUF4` [replied](https://openreview.net/forum?id=HE6j2jtjII&noteId=ojTBWmrMWN) and *reaffirmed their positive score of* ***8***. The remaining reviewers did not have time to respond our rebuttal before the reviewer-response window closed.

---

**Reviewer Consensus on Key Strengths:** From the initial reviews, we are pleased that the core contributions of **UrbanVerse** were *consistently* recognized. *All Reviewers* (`Nwq7` / `9ozp` / `ND3o` / `eUF4`) highlighted the “[impressive](https://openreview.net/forum?id=HE6j2jtjII&noteId=VDTuXZZGIE)”, “[excellent](https://openreview.net/forum?id=HE6j2jtjII&noteId=xyvTff1lNf)”, and “[very strong](https://openreview.net/forum?id=HE6j2jtjII&noteId=adpAIYpoix)” sim-to-real performance of navigation policies trained in our UrbanVerse scenes. Most Reviewers (`Nwq7` / `ND3o` / `eUF4`) further recognized our real-to-sim UrbanVerse-Gen scene generation pipeline as “[novel](https://openreview.net/forum?id=HE6j2jtjII&noteId=VDTuXZZGIE)”, “[highly scalable](https://openreview.net/forum?id=HE6j2jtjII&noteId=VDTuXZZGIE)”, “[effective](https://openreview.net/forum?id=HE6j2jtjII&noteId=xyvTff1lNf)”, and “[non-trivial](https://openreview.net/forum?id=HE6j2jtjII&noteId=adpAIYpoix)”. In addition, multiple Reviewers (`Nwq7` / `9ozp` / `eUF4`) acknowledged our “[extensive efforts](https://openreview.net/forum?id=HE6j2jtjII&noteId=wQU5HOKth5)” in building the “[massive and usable](https://openreview.net/forum?id=HE6j2jtjII&noteId=adpAIYpoix)” UrbanVerse-100K asset database, noting that it both “[addresses major limitations of existing simulators](https://openreview.net/forum?id=HE6j2jtjII&noteId=VDTuXZZGIE)” and is “[directly valuable to the autonomous driving community](https://openreview.net/forum?id=HE6j2jtjII&noteId=adpAIYpoix)”.

---

**Key Concerns and How We Addressed Them:** Alongside the positive feedback on **UrbanVerse**, reviewers raised several questions regarding additional functional capabilities, computational costs, annotation quality, and developer documentation. We addressed each point in detailed individual responses and updated the manuscript accordingly.  Below, we concisely summarize the main concerns and how we addressed them:

---

> **Q1:** Can UrbanVerse handle interactive dynamic agents? (asked by `Nwq7` / `ND3o`)

Yes. UrbanVerse fully supports dynamic agents and multi-agent interactions. To directly address this concern, we provided:
- Demo videos (**[Click Here: UrbanVerse Scenes Populated with Dynamic Agents)](https://anonymoususeruseanonymousname.github.io/scene_with_dynas.html)**) of UrbanVerse scenes populated with diverse dynamic agents, including robots, pedestrians, cars, wheelchair users, and e-scooter riders
- A newly added description of *Interactive Dynamic-agent Population* in the *main* paper (**Line#307–313**) and visual examples in **Figure 5**.

---

> **Q2:** Can UrbanVerse support diverse embodied AI tasks? (asked by Reviewers `Nwq7` / `ND3o`)

Yes. UrbanVerse scenes are fully interactive with semantic and physical annotations, enabling a wide range of embodied tasks such as mobile manipulation, multimodal data collection, VR interaction, and more. To directly address this concern, we provided:
- Mobile manipulation demos (**[Click Here: Mobile Manipulation in UrbanVerse Scenes](https://anonymoususeruseanonymousname.github.io/mobile_manipulator.html)**) operated in UrbanVerse scenes across two robot platforms.
- A VR interaction demo (**[Click Here: VR Interaction Demo)](https://anonymoususeruseanonymousname.github.io/vr_interaction.html)**).
- A newly added description on supported tasks in the *main* paper (**Line#307–313**) and visual examples in **Figure 5**.

---

> **Q3:** Computational cost analysis (asked by Reviewers `Nwq7` / `ND3o` / `eUF4`)

To address this concern, we conducted a comprehensive analysis of computational cost and scalability of UrbanVerse system, provided in the rebuttal and incorporated into the revised paper in **Appendix L**.

---

> **Q4:** Evaluation of UrbanVerse-1K annotation quality (asked by Reviewers `Nwq7` / `9ozp`)

To address this, we provided and conducted:
- A new quantitative evaluation on the annotated attributes againts their collected ground-truth in both the rebuttal and **Appendix C.4**.
- A walkthrough video (**[Click Here: UrbanVerse-1K Object Gallery Walkthrough)](https://anonymoususeruseanonymousname.github.io/urbanverse_1k.html)**) showing hundreds of UrbanVerse-1K assets in a unified scene to validate their realistic, physically plausible dimensions in true metric-scale.

---

---

### Author Response · Authors · 2025-11-30
**Overall Summary (1/3): Paper and Contribution**

We sincerely thank  Area Chairs and the Reviewers for their time, careful evaluations, and constructive feedback. Their comments helped us substantially strengthen the revised version of **UrbanVerse**, now posted. We especially appreciate the tremendous efforts of the newly assigned Area Chairs in carefully reviewing our rebuttal and revised manuscript.

In the following overall summary, we provide a concise overview of the paper’s core contributions, summarizes the reviewers’ feedback and main concerns, and highlights how each point was addressed and incorporated into the revised manuscript.

---

## ***Paper and Contribution Summary***

---

This paper introduces **UrbanVerse**, a data-driven real-to-sim system that automatically converts crowd-sourced city-tour videos into physics-aware, interactive urban simulation scenes in IsaacSim for embodied-AI policy learning. By moving from traditional procedural or manually authored pipelines to a *data-driven* paradigm, UrbanVerse addresses two long-standing limitations of prior simulators: *(i)* procedural methods (e.g., MetaUrban, UrbanSim) fail to match real-world distribution fidelity, and *(ii)* manual authoring (e.g., CARLA) does not scale. As a result, UrbanVerse produces scenes that more faithfully capture real-world street layouts and object distributions, enabling scalable urban robot learning and reducing the sim-to-real gap. Experiments show that UrbanVerse scenes preserve real-world semantics and layouts, achieving human-rated realism comparable to manually crafted scenes. For urban navigation, policies trained in UrbanVerse exhibit clear scaling laws, strong generalization, and superior sim-to-real performance over prior methods.


In summary, UrbanVerse delivers four key contributions:
1. **UrbanVerse-Gen:** A scalable, novel automatic real-to-sim pipeline that converts city-tour videos into physics-aware urban simulation scenes, replacing traditional template-based (procedural) and manual workflows.
2. **UrbanVerse-100K:** An order-of-magnitude larger and higher-quality asset library containing **102,530** metric-scale 3D objects across **667 categories**, **288** ground texture materials, and **306** photorealistic sky illumination maps, each annotated with rich semantic and physical attributes (e.g., mass, friction) *(Prior simulators typically provide <106 categories and <15,000 assets.)*
3. **Reay-to-use Simulation scenes:** A collection of **160 (now expanded to 320)** fully interactive urban simulation scenes generated from city-tour videos across the world using UrbanVerse-Gen, plus **CraftBench**, a set of 10 high-fidelity, test-only environments crafted by professional 3D artists.
4. **Empirical validation of strong scaling behavior and generalization:** Mapless urban navigation policies trained in UrbanVerse: *(i)* exhibit scaling power laws, *(ii)* improve success rate by **+6.3%** in simulation and **+30.1%** in zero-shot sim-to-real transfer over prior methods, and *(iii)* complete a 337 m real-world mission on public streets with only two interventions.

As noted by Reviewers `Nwq7` and `eUF4`, all assets, scenes, and code of UrbanVerse will be *fully open-sourced* to accelerate urban embodied-AI research.

---

### Meta-Review · Area_Chair_z3iZ · 2026-01-07

**Summary:**

This paper presents UrbanVerse, a new dataset for training and evaluated embodied AI agents in urban environments. The proposed dataset is built using a repository of 100k+ annotated 3D assets. The environments themselves are then constructed via a pipeline that extracts metrics-scale 3D simulations from videos of urban scenes. The resulting dataset has over 160 scenes that run in IsaacSim.

While there were concerns raised in the initial review, the authors responded to them well.

As a note, the reference

	Vishnu Sashank Dorbala, James F Mullen, and Dinesh Manocha. Can an embodied agent find your "cat-shaped mug"? 1lm-based zero-shot object navigation. IEEE Robotics and Automation Letters, 9(5):4083-4090, 2023. 33, 34


contains a typo. The reference in the paper is **1**lm but should be **l**lm. I also the authors to double check their use of their reference. As far as I can tell, the cited work does not use IsaacSim like the usage of the citation implies.

**Reviewer Concerns:**

## Reviewer Nwq7

- The proposed pipeline is overly complex. I believe this concern was addressed.
- No failure case analysis. I believe this concern was addressed.
- A concern about dataset quality. I believe this concern was addressed.
- The validation of the dataset focuses on only navigation. I believe this concern was addressed.
- System integration over algorithmic innovation. I believe this concern was addressed.
- Runtime and scalability analysis. I believe this concern was addressed.

## Reviewer 9ozp

- Why are existing 3DGS-based methods insufficient? I believe this is addressed.
- Annotation errors. I believe this is addressed.
- Single-plane ground fitting step. I believe this has been largely addressed. The reviewer was likely looking for something more extensive than just a road plane and a sidewalk plane with a fixed height offset, but two is better than one.
- Limited number of scenes. I believe this has been addressed.

## Reviewer ND3o

- Limited amount of interactivity and dynamic actors within the environment. I believe this was a misunderstanding by the reviewer and has been addressed.
- Lack of differentiation with existing simulators. I believe this has been addressed.
- Unclear story of how the community can leverage this work. I believe this has been addressed.

## Reviewer eUF4

This reviewer brought up major concerns, but had a several questions that the authors responded to well.

**Reviewer Scores:**

Reviewer Nwq7 -- this reviewer may have increased their score to an 8.
Reviewer 9ozp -- this reviewer likely would have stayed with a 6.
Reviewer ND3o -- this reviewer would have increased their core to a 6.
Reviewer eUF4 -- this reviewer would have remained at an 8.

---

### Decision · Program_Chairs · 2026-01-26

Accept (Poster)